# Fast Local Search Algorithms for Clustering with Adaptive Sampling and Bandit Strategies

**Junyu Huang[1], Zhen Zhang[2,3], Beirong Cui[1], Jianxin Wang[1,4], Qilong Feng[1,*]**

[1]School of Computer Science and Engineering, Central South University
[2]School of Advanced Interdisciplinary Studies, Hunan University of Technology and Business
[3]Xiangjiang Laboratory, Changsha, China
[4]The Hunan Provincial Key Lab of Bioinformatics, Central South University,
Changsha 410083, China
`junyuhuangcsu@foxmail.com, csuzz@foxmail.com`
`234711098@csu.edu.cn, jxwang@mail.csu.edu.cn, csufeng@mail.csu.edu.cn`

## Abstract

Local search is a powerful clustering technique that provides high-quality solutions with theoretical guarantees. With distance-based sampling strategies, local search methods can achieve constant approximations for clustering with linear running time in data size. Despite their effectiveness, existing algorithms still face scalability issues as they require scanning the entire dataset for iterative center swaps. This typically leads to an $O(ndk)$ running time, where $n$ is the data size, $d$ is the dimension, $k$ is the number of clusters. To further improve the efficiency of local search algorithms, we propose new methods based on adaptive sampling and bandit strategies. Specifically, adaptive sampling can well approximate the distance-based sampling distribution without maintaining pairwise distances between data points and the centers, enabling fast and accurate sampling in sublinear time after an $\tilde{O}(nd)$ time preprocessing step. The bandit strategy models the best swap pair selection as a bandit problem, where a grouping strategy is proposed for fast identification of the optimal swap pair. With these techniques, our proposed algorithm can achieve constant approximation in expected running time $\tilde{O}(nd + k^4)$ under mild assumptions on optimal clusters and swap pair distributions. Our approach also extends naturally to the $k$-median objective, achieving constant approximation in expected running time $\tilde{O}(nd + \sqrt{n}k^3)$ without distributional assumptions. Empirical results demonstrate that our algorithm achieves up to 1000× speedup over existing local search methods on datasets with 100 million points, while delivering comparable clustering quality. Compared to coreset-based approaches, it provides up to around 80× speedup and consistently yields better clustering results.

## 1 Introduction

Clustering is a fundamental unsupervised learning problem with wide applications in machine learning. It aims to partition a dataset into clusters such that points within the same cluster can share high similarity. Among various formulations, $k$-means and $k$-median are widely studied for their practical effectiveness. This paper focuses on these two problems in $d$-dimensional Euclidean space.

Over the past few decades, numerous heuristic and approximation algorithms have been developed for the $k$-means and $k$-median problems. Among them, the Lloyd's algorithm [27] remains the most widely used in practice. However, a potential issue is that Lloyd's algorithm may require an

---

*Corresponding Author

unbounded number of iterations to reach a convergence [1]. To address this, Arthur and Vassilvitskii [1] proposed the $k$-means++ seeding method, which achieves an $O(\log k)$-approximation for both $k$-means and $k$-median in $O(ndk)$ time, where $n$ is the data size, $d$ is the dimension, and $k$ is the number of clusters. Intuitively, $k$-means++ selects $k$ centers sequentially, choosing each new center with probability proportional to the (squared) distances from data points to their nearest centers (also known as the $D^2$-Sampling (or $D$-Sampling) strategy [25]). To further enhance the clustering quality, local search methods are commonly employed to achieve constant approximations [23, 2, 25, 11, 21, 7]. Starting from a random initial solution, local search aims to find iterative improvements on clustering quality through center swaps between data points and the centers selected. Kanungo et al. [23] proposed the first local search algorithm for $k$-means, achieving a polynomial-time $(9 + \epsilon)$-approximation. Arya et al. [2] gave the first local search algorithm for $k$-median, achieving a polynomial-time $(3 + \epsilon)$-approximation. Recently, Cohen-Addad et al. [13] improved this result for $k$-median to a polynomial-time $(2.836 + \epsilon)$-approximation by introducing distance-based potential functions, which is the current best approximation result for $k$-median with local search.

As contemporary datasets scale to hundreds of millions of entries, even algorithms with modest polynomial complexity can become impractical. Hence, in $d$-dimensional Euclidean space, there has been growing interest in developing local search algorithms with linear runtime in the data size. In this setting, Lattanzi and Sohler [25] proposed the LS++ algorithm. Instead of exhaustive enumerations, LS++ selects exactly one point for swapping in via $D^2$-Sampling strategy. They proved that LS++ can achieve an $O(1)$-approximation for both $k$-means and $k$-median objectives in time $O(ndk^2 \log \log k)$, using $O(k \log \log k)$ local search steps. Choo et al. [11] further reduced the number of local search steps to $O(\epsilon k)$ for LS++, achieving an $O(1/\epsilon^3)$-approximation. Fan et al. [19] narrows the swap candidates to exactly 2 using a greedy strategy, achieving an $O(1)$-approximation in $O(ndk \log \log k)$ time under mild assumptions on average cluster sizes. While single-swap local search outperforms the seeding methods, there is still a gap on clustering quality guarantees compared to multi-swap strategies. To bridge this gap, recent studies have focused on sampling-based multi-swap methods. Huang et al. [21] proposed a $(50(1 + 1/t) + \epsilon)$-approximation algorithm for $k$-means with $O(ndk^{2t+1} \log(\epsilon^{-1} \log k))$ time, where $t$ is the swap size. Independently, Beretta et al. [7] gave a 10.46-approximation algorithm for $k$-means in $O(nd\text{poly}(k))$ time, which is the current best result for $k$-means with linear runtime in data size. These local search methods can naturally be extended to the general metric space to achieve $O(1)$-approximation and linear running time in the data size.

For clustering in general metric spaces, it has been proved that achieving any constant-factor approximation requires $\Omega(nk)$ time [29]. Therefore, a natural question arises: can we further improve this bound in $d$-dimensional Euclidean space? Along this line of research, several algorithms have been proposed. One approach is to incorporate the Euclidean spanners [20] with a near-linear time graph-based clustering algorithm [32], where an $O(c)$-approximation can be achieved in $O(n^{1+1/c})$ time for $k$-median. However, its extension to the $k$-means objective remains unclear. Recently, la Tour and Saulpic [24] proposed an $O(1)$-approximation framework with $\tilde{O}(nd + n^{1+1/c})$ [2] running time for $k$-clustering, using geometric hashing and ball-covering strategies for center selections. Alternatively, another direction focuses on coresets constructions [17, 5, 16, 14, 12], which aims to reduce the data sizes to $\tilde{O}(k/\epsilon)$ with preserved $(1 + \epsilon)$-approximation guarantees. While traditional coresets methods require $O(ndk)$ time, Draganov et al. [18] broke this barrier and gave an $\tilde{O}(nd)$ time coreset method. Combined with LS++, this yields an $O(1)$-approximation in $\tilde{O}(nd + k^3 d)$ time.

Although some existing algorithms can achieve faster than $O(ndk)$ running times for $k$-median and $k$-means problems, they appear to be less practical and may still encounter difficulties for handling real-world large-scale datasets. For the almost linear-time algorithms [24, 32], while they can offer compelling guarantees on clustering quality and running time, the practical implementations often involve complex data structures. Consequently, it remains unclear whether these methods can effectively scale to datasets with sizes over 100 million. As for coresets methods, even with sampling and spread compression strategies [18], our experiments (see Appendix E.2) show that these methods still face scalability issues. It can take over 2 hours of computations for datasets with 100 million points (while our algorithm can produce higher-quality solutions within just 2 minutes).

In this paper, we aim to bypass the coresets construction methods and develop practical clustering algorithms with theoretical guarantees for large-scale datasets. Given that local search is widely used in large-scale clustering applications, our primary focus is still on local search algorithms.

---

[2] The $\tilde{O}$ notation is used to hide polylogarithmic factors in $n$ and $d$.

Specifically, our goal is to design local search algorithms with running time $\tilde{O}(nd + n^\xi \text{poly}(k)d)$ for a fixed constant $\xi \in [0, 1)$. This improves upon the running time of previous local search algorithms in the common practical settings where $k$ is relatively small compared to data size $n$. In the following, we briefly remark on the challenges encountered in this setting. Firstly, it is a non-trivial task to break the $O(ndk)$ running time for sampling-based swap pair constructions. Existing linear-time local search algorithms typically rely on $D^2$-Sampling [25, 21, 19, 7] to avoid exhaustive enumerations. However, this approach requires maintaining distances between data points and their nearest centers, leading to an $O(nd)$ update time for each local search step. Since $\Omega(k)$ steps are usually required for theoretical convergence, the total complexity for the sampling process becomes $O(ndk)$. To accelerate the sampling process, several approximation methods have been proposed, including approximate $D^2$-Sampling [3, 4], approximate nearest neighbor search [15], and projection-based sampling [9]. However, these methods are primarily designed for finding initial seedings, where centers are sampled sequentially. It remains unclear whether they can effectively handle the setting for local search, in which centers may be frequently inserted and deleted. Secondly, identifying the best swap pair at each local search step is computationally expensive, as evaluating the clustering cost change for even a constant number of candidate swaps requires scanning the entire dataset, leading to an $O(ndk)$ complexity. To accelerate the best swap pair identification, recent approaches proposed to formulate swap pair selection as a bandit problem [34, 33, 6]. Nevertheless, these methods still require a search space that scales linearly with the data size $n$, resulting in a runtime complexity of $\tilde{O}(ndk)$. Moreover, their theoretical guarantees rely on restrictive assumptions, requiring the cost changes induced by each swap pair to follow a Gaussian distribution $\mathcal{N}(\gamma, 1)$ with a constant mean $\gamma$.

## 1.1 Our Contribution

The main objective in this paper is to develop faster local search algorithms for clustering while bypassing coresets construction methods. For $k$-means clustering, we first propose a sampling-based algorithm achieving a constant approximation in expected running time $\tilde{O}(nd + \sqrt{n}k^4)$, assuming a sub-Gaussian prior on cost changes of swap pairs. To accelerate the sampling process, we propose an adaptive sampling method that dynamically maintains a tree structure to approximate sampling distributions. The tree structure can be initialized in $\tilde{O}(nd)$ time and updated within sublinear time. Unlike previous tree embedding methods (i.e., HSTs [15, 24]), our approach dynamically maintains a weight distribution for each internal node, enabling an approximate $D^2$-Sampling process to be executed on the tree. For the best swap pair identification, we model the task as a bandit problem, regarding swap pairs as arms. Existing bandit methods usually require strict Gaussian assumptions for cost changes of each swap pair [6, 34, 33] with constant mean and deviations. To relax this assumption, we propose to combine sampling with a swap pair grouping strategy, which can eliminate unnecessary swap pairs assuming sub-Gaussian prior on cost changes of swap pairs with bounded deviations. With theses techniques, the best swap pair can be determined in subliner time $\tilde{O}(k^3)$.

To further enhance the scalability of local search algorithms, we propose a more practical local search algorithm that adapts Metropolis-Hastings strategy for approximating the $D^2$ sampling process. Metropolis-Hastings strategy generates candidate samples from a uniform proposal distribution and employs rejection sampling to determine transitions between samples. We show that our algorithm can effectively approximate the $D^2$-Sampling distribution under mild assumptions regarding the sizes and tail behavior of optimal clusters, which enables each sampling step to be executed in $\tilde{O}(n/k)$ time. By combining with bandit methods, a constant approximate solution can be obtained in expected time $\tilde{O}(nd + k^4)$. Experiments show that our algorithm can achieve clustering quality comparable to existing local search methods, with up to 1000× speedup on datasets containing 100 million points. Compared to coresets, it provides up to around 80× speedup with better qualities.

Finally, we extend our adaptive sampling approach to the $k$-median objective by introducing a weighted sampling strategy. This strategy can estimate the $k$-median costs for swap pairs using weighted samples drawn from the dynamic tree structure maintained. We show that clustering costs can be approximated within a $(1 \pm \epsilon)$ factor using $\tilde{O}(\sqrt{n}/\epsilon)$ samples. By incorporating swap pair grouping strategy, sub-optimal swap pairs can be effectively filtered in $\tilde{O}(\sqrt{n}k^2/\epsilon)$ time for each local search step. Thus, our approach can give a constant approximation for the $k$-median objective in expected $\tilde{O}(nd + \sqrt{n}k^3)$ time, without requiring any distribution assumptions on data points.

Putting all these together, we can get the following results.

**Theorem 1.** *With sub-Gaussian prior on cost changes of swap pairs, there exists an algorithm for $k$-means that can achieve constant approximation within expected running time $\tilde{O}(nd + \sqrt{n}k^4)$.*

**Theorem 2.** *With sub-Gaussian prior on cost changes of swap pairs and mild assumptions on optimal clusters, there is a constant approximation algorithm for $k$-means with expected runtime $\tilde{O}(nd + k^4)$.*

**Theorem 3.** *For the $k$-median problem, there exists an algorithm that can output a constant approximate solution in expected $\tilde{O}(nd + \sqrt{n}k^3)$ running time without any data assumptions.*

Table 1: Comparisons with existing local search methods ($n$: data size, $d$: dimension, $t$: swap size).

| Objective | Ref | Approximation | Assumption | Running Time |
|---|---|---|---|---|
| | [23] | $9 + \epsilon$ | - | $\epsilon^{-1}d\mathrm{poly}(n,k)$ |
| | [25] | $O(1)$ | - | $O(ndk^2 \log\log k)$ |
| | [11] | $O(1)$ | - | $O(ndk^2)$ |
| $k$-means | [19] | $O(1)$ | Average Cluster Sizes | $O(ndk \log\log k)$ |
| | [21] | $50(1 + 1/t) + \epsilon$ | - | $O(ndk^{2t+1} \log(\epsilon^{-1}\log k))$ |
| | [7] | $10.46$ | - | $O(nd\mathrm{poly}(k))$ |
| | This Paper | $O(1)$ | Sub-Gaussian Prior | $\tilde{O}(nd + \sqrt{n}k^4)$ |
| | | $O(1)$ | Sub-Gaussian Prior, Tailed Behavior and Sizes of Optimal Clusters | $\tilde{O}(nd + k^4)$ |
| | [2] | $3 + \epsilon$ | - | $\epsilon^{-1}d\mathrm{poly}(n,k)$ |
| | [25] | $O(1)$ | - | $O(ndk^2 \log\log k)$ |
| | [11] | $O(1)$ | - | $O(ndk^2)$ |
| $k$-median | [19] | $O(1)$ | Average Cluster Sizes | $O(ndk \log\log k)$ |
| | [21] | $O(1) \, (< 50(1 + 1/t) + \epsilon)$ | - | $O(ndk^{2t+1} \log(\epsilon^{-1}\log k))$ |
| | [7] | $O(1) \, (< 10.46)$ | - | $O(nd\mathrm{poly}(k))$ |
| | This Paper | $O(1)$ | - | $\tilde{O}(nd + \sqrt{n}k^3)$ |

Table 1 presents detailed comparisons with state-of-the-art results. For the $k$-means objective, our algorithm achieves constant approximation while reducing the joint dependence on $n$ and $k$, improving the efficiency when $k \ll n$. For the $k$-median objective, our method also ensures constant approximation with faster running time without relying on any distributional assumptions.

## 2 Preliminary

In this paper, we use $P \subset \mathbb{R}^d$ and $k$ to denote the set of the given dataset and the number of clusters to be opened, respectively. For any two data point $p, q \in \mathbb{R}^d$, let $\delta(p, q)$ and $\delta^2(p, q)$ denote their Euclidean and squared Euclidean distance, respectively. Given a set $C \subset \mathbb{R}^d$ of data points, denote $\delta(P, C) = \sum_{p \in P} \min_{c \in C} \delta(p, c)$ and $\delta^2(P, C) = \sum_{p \in P} \min_{c \in C} \delta^2(p, c)$ as the clustering cost of $P$ with respect to $C$ for the $k$-median and $k$-means objectives, respectively. We use $OPT$ to denote the optimal clustering cost of the given clustering instance. The goal of the $k$-median ($k$-means) clustering is to find a set $C \subset \mathbb{R}^d$ with size $k$ while minimizing $\delta(P, C)$ (or $\delta^2(P, C)$ for $k$-means).

The dimension of a given clustering instance can be reduced from $d$ to $O(\log n)$ via the standard Johnson-Lindenstrauss Transform, which incurs only constant factor approximation loss [22, 28]. Without loss of generality, we can assume that $d = O(\log n)$ through an $\tilde{O}(nd)$ time preprocessing step for dimensionality reduction.

## 3 The BanditLS Algorithm

In this section, we present the BanditLS algorithm, which integrates adaptive sampling and bandit-based strategies to accelerate local search for $k$-means. Our main objective is to design a local search algorithm with running time $\tilde{O}(nd + \sqrt{n}\mathrm{poly}(k))$, thereby reducing the combined impact of parameters $n$ and $k$ on the running time. This can improve the efficiency of previous local search methods, especially when $k$ is relatively small compared to the data size $n$. While coreset-based approaches can achieve $\tilde{O}(nd + k^3d)$ time, they often fail to scale well in practice due to time overheads. For datasets with sizes over 100 million, the constructions for coresets can require several hours of computation, making it less practical for large-scale scenarios (see Appendix E.2 for details). The key challenges here lie in bypassing the coresets construction methods while accelerating both the sampling process and identification of the best swap pairs during the local search swaps.

The formal description for the proposed BanditLS algorithm is given in Algorithm 1. We first outline the main idea behind. To eliminate the $O(ndk)$ overhead caused by explicitly maintaining pairwise distances for $D^2$-Sampling, we propose an adaptive sampling method. Specifically, we first embed the dataset into a tree structure in near-linear preprocessing time, aiming to discretize the candidate distances between points and centers from $k$ down to $O(\log^2(nd))$. Instead of computing pairwise distances, we then dynamically maintain weight distributions at internal nodes, capturing the distance distribution between data points and their centers. Based on traversals of the maintained weight distributions from root to the leaf, the $D^2$-Sampling process can be approximated within sublinear time. To efficiently select the best swap pair without exhaustive scans during the swaps, we formulate the task as a bandit problem. Unlike previous methods relying on strict Gaussian assumptions for all swap pairs, we propose a sampling-based grouping strategy that only requires assumptions for high-impact swap pairs. This approach can filter out ineffective swaps using only $\tilde{O}(k^2)$ samples, reducing the swap selection complexity from $O(ndk)$ to be independent of the data size $n$.

Starting from a random solution, BanditLS proceeds in three main stages: (1) Tree construction (steps 1–5) where the data is embedded into a bounded-height tree $\mathcal{T}'$ and centers are opened or closed by marking the corresponding leaf nodes; (2) Swap pair construction (steps 7–9) where an adaptive sampling strategy is used to approximate the $D^2$-Sampling distribution via dynamic weight distributions maintained, allowing efficient sampling without explicitly tracking nearest-center distances; (3) Swap pair identification and tree updates (steps 10–17), where the best swap pair is selected via a bandit model. To avoid restrictive assumptions, a grouping strategy is proposed to include virtual swap pairs with zero cost changes, enabling sublinear time identification of promising swap pairs under sub-Gaussian prior. These components together yield the improved runtime.

---

**Algorithm 1** BanditLS$(P, k, d, C, \eta, \sigma)$

---

**Input:** A $k$-means clustering instance $(P, k, d)$, a set $C \subset \mathbb{R}^d$ of random initial clustering centers, a parameter $\eta \in (0, 1)$, a parameter $\sigma > 0$ representing the prior knowledge on deviations.
**Output:** A set $C \subset \mathbb{R}^d$ of clustering centers.

 1:   $\mathcal{T} =$ Tree-Construction$(P, k, d)$. ▷ Tree-Construction is detailed in Algorithm 4 in Appendix A
 2:   $\mathcal{T}' =$ Tree-Conversion$(\mathcal{T})$.        ▷ Tree-Conversion is detailed in Algorithm 6 in Appendix A
 3:   Call TREE-INIT$(\mathcal{T}')$ to initialize the dynamic data structure.      ▷ Algorithm 7 in Appendix A
 4:   **for** $c \in C$ **do**
 5:      Call the TREE-OPEN$(\mathcal{T}', c)$ algorithm to mark the leaf node in $\mathcal{T}'$ associated with $c$ as active and update the tree structure $\mathcal{T}'$.    ▷ TREE-OPEN is detailed in Algorithm 8 in Appendix A
 6:   **for** $i = 1$ to $\tilde{O}(\sqrt{n}k)$ **do**
 7:      $x =$ Adaptive-Sampling$(\mathcal{T}', C)$.     ▷ Adaptive-Sampling is in Algorithm 10 in Appendix A
 8:      Set $s_x = \arg\min_{c \in C} \delta(x, c)$ and randomly sample a center $q' \in C \backslash \{s_x\}$.
 9:      $\mathcal{S}_1 = \{(x, s_x), (s_x, s_x)\}, \mathcal{S}_2 = \{(x, q'), (q', q')\}$.
10:      $o_1 =$ BanditCS$(P, k, d, C, \mathcal{S}_1, \eta, \sigma), o_2 =$ BanditCS$(P, k, d, C, \mathcal{S}_2, \eta, \sigma)$.      ▷ Algorithm 2
11:      **if** $|o_1| > 1$ and $|o_2| > 1$ **then**
12:         Continue.
13:      **else**
14:         Let $\mathcal{O} = \{o_i : |o_i| = 1, i \in \{1, 2\}\}$, and randomly choose an $o'$ from $\mathcal{O}$.
15:      $C = C \backslash \{v\} \cup \{u\}$, where $(u, v) \in o'$.
16:      Call the TREE-CLOSE$(\mathcal{T}', v)$ algorithm to deactivate the leaf node in $\mathcal{T}'$ associated with $v$ and update the tree structure $\mathcal{T}'$. ▷ TREE-CLOSE is detailed in Algorithm 9 in Appendix A
17:      Call the TREE-OPEN$(\mathcal{T}', u)$ algorithm to mark the leaf node in $\mathcal{T}'$ associated with $u$ as active and update the tree structure $\mathcal{T}'$.
18:   **return** $C$.

---

### 3.1 Tree Construction and Adaptive Sampling

In this subsection, we give the analysis for the proposed tree construction and adaptive sampling methods, which are the key components for accelerating the $D^2$-Sampling process. The core idea is to discretize the distances between data points and their centers by embedding them into a dynamically maintained tree structure. This reduces the number of distinct distance categories from $k$ to $O(\log^2(nd))$, corresponding directly to the tree's height. Thus, by traversing from root to the

leaves, the $D^2$-Sampling process can be approximated within $O(\log^2(nd))$ time. Due to space limit, we present the main ideas here and leave the detailed algorithmic descriptions in Appendix A.

Our approach begins by embedding the clustering instance into a tree structure $\mathcal{T}$ (see Algorithm 4 in Appendix A.1 for details) with height $H = O(\log(d\Delta))$, where $\Delta$ is the aspect ratio of the clustering instance[3]. Each node $v \in \mathcal{T}$ is assigned a level $le(v)$, where leaf nodes have level 1, and the levels increase by one at each node when moving upward toward the root (see step 7 of Algorithm 4). For any two points $p, q \in P$, let $v_{p,q}$ denote their lowest common ancestor in $\mathcal{T}$. The tree distance between $p$ and $q$ is defined as $\delta_\mathcal{T}(p, q) = 2\sqrt{d} \cdot (2^{le(v_{p,q})} - 2)$. Given a set $C$ of centers, define $\delta_\mathcal{T}(p, C) = \min_{c \in C} \delta_\mathcal{T}(p, c)$ as the tree distance from data point $p$ to the center in $C$ that shares the lowest common ancestor with $p$ in $\mathcal{T}$. If each internal node $v$ in the tree $\mathcal{T}$ has at most two children, let $l(v)$ and $r(v)$ denote its left and right children, respectively. If $v$ has only one child, the child of $v$ is treated as its left child for the ease of analysis. Given a node $v$ of the tree $\mathcal{T}$, let $\mathcal{B}_v$ denote the set of points corresponding to the leaf nodes that are descendants of node $v$. Denote $\delta_\mathcal{T}(\mathcal{B}_v, C) = \sum_{v' \in \mathcal{B}_v} \delta_\mathcal{T}(v', C)$. The following lemma shows that the tree structure $\mathcal{T}$ can well approximate the pairwise distances.

**Lemma 1.** (Cohen-Addad et al. [15]) $\delta(p, q) \leq \delta_\mathcal{T}(p, q)$ and $E[\delta_\mathcal{T}(p, q)] \leq \tilde{O}(d) \cdot \delta(p, q)$.

Based on the tree structure, we will maintain weight distributions at each node of the tree $\mathcal{T}$, enabling a simulated $D^2$-Sampling process by traversing from root to the leaves. However, since the internal nodes may have $O(n)$ children, sampling could require traversing $O(n)$ branches, resulting in $\Omega(n \log(d\Delta))$ running time. To address this issue, a tree conversion algorithm (Algorithm 6 in Appendix A.2) is proposed to convert the tree structure $\mathcal{T}$ into a new tree structure $\mathcal{T}'$ with bounded branches by creating auxiliary nodes. The following lemma shows that this process only adds an $O(\log(d\Delta))$ factor to the tree height while preserving the bounds for the expected pairwise distances.

**Lemma 2.** Let $\mathcal{T}'$ be the modified tree structure of the output for Algorithm 6. Then, each vertex in $\mathcal{T}'$ has at most 2 children, and the height of the tree $\mathcal{T}'$ can be bounded by $O(\log^2(d\Delta))$. Additionally, it holds that $\delta(p, q) \leq \delta_{\mathcal{T}'}(p, q)$ and $E[\delta_{\mathcal{T}'}(p, q)] \leq \tilde{O}(d) \cdot \delta(p, q)$.

Next, we describe how to maintain a data structure $\mathcal{D}$ to approximate the $D^2$-Sampling distributions. Let $C$ be the set of centers dynamically maintained by the local search process. The data structure $\mathcal{D}$ on $\mathcal{T}'$ should satisfy the following properties.

- Each non-auxiliary node $v$ in the tree $\mathcal{T}'$ is assigned a level $le(v)$, with leaf nodes assigned with level 1. Along any path from a leaf to the root, the level increases by 1 at each non-auxiliary node, while auxiliary nodes retain the same level as their parents.

- Each node $v$ is associated with three values $n_v^l$, $n_v^r$, and $n(v)$. For non-leaf nodes, $n_v^l$ and $n_v^r$ denote the number of leaf nodes in the left and right subtrees of $v$, respectively. $n(v)$ represents the number of active leaf nodes (or opened centers) that are descendants of $v$. Each node $v \in \mathcal{T}'$ is also associated with a weight $w(v)$ where $w(v) = \delta_{\mathcal{T}'}(\mathcal{B}_v, C)$.

- Each non-leaf node $v \in \mathcal{T}'$ is associated with a probability list $p(v)$, which represents the proportion of the total weights contributed by each of its children. If $v$ has only one child, then $p(v) = [1, 0]$. If $v$ has two children, $p(v) = [l_v, r_v]$, where $l_v = \frac{\delta_{\mathcal{T}'}(\mathcal{B}_{l(v)}, C)}{\delta_{\mathcal{T}'}(\mathcal{B}_v, C)}$, $r_v = \frac{\delta_{\mathcal{T}'}(\mathcal{B}_{r(v)}, C)}{\delta_{\mathcal{T}'}(\mathcal{B}_v, C)}$, and $l_v + r_v = 1$.

The data structure $\mathcal{D}$ can be initialized via a bottom-up traversal from the leaf nodes to the root (see Appendix A.3 for details). However, center swaps during the local search process may influence the properties of the maintained data structure. To address this issue, we propose two operations: TREE-OPEN and TREE-CLOSE (as detailed in Appendix A.3). Specifically, TREE-OPEN marks a leaf node as active and updates $\mathcal{D}$ along the path from the leaf to the root, while TREE-CLOSE deactivates a marked leaf node and also updates the structure along the path from the leaf to the root. The following lemma shows that both operations can achieve sublinear update time.

**Lemma 3.** The TREE-OPEN and TREE-CLOSE operations can run in time $O(\log^2(d\Delta))$ to update the tree structure. Let $C'$ be the set of centers after a swap. Then, each non-leaf node $v \in \mathcal{T}'$ holds a probability list $p(v) = [l_v, r_v]$, where $l_v = \frac{\delta_{\mathcal{T}'}(\mathcal{B}_{l(v)}, C')}{\delta_{\mathcal{T}'}(\mathcal{B}_v, C')}$ and $r_v = \frac{\delta_{\mathcal{T}'}(\mathcal{B}_{r(v)}, C')}{\delta_{\mathcal{T}'}(\mathcal{B}_v, C')}$.

---

[3] $\Delta = \frac{\max_{p,q \in P} \delta(p,q)}{\min_{p',q' \in P: p' \neq q'} \delta(p',q')}$ and can be compressed to $\text{poly}(n, d)$ in time $\tilde{O}(nd)$ (Draganov et al. [18]).

Using the TREE-OPEN and TREE-CLOSE operations, we can always maintain a tree structure $\mathcal{T}'$ satisfying the properties of data structure $\mathcal{D}$. The Adaptive-Sampling algorithm (Algorithm 10 in Appendix A.4) then performs a sampling process on $\mathcal{T}'$ to approximate the $D^2$-Sampling distribution: it initializes an empty set $\mathcal{S}$ and repeatedly performs sampling on the tree $\mathcal{T}'$ until a point is accepted. Each sampling step starts at the root and proceeds down the tree, choosing children based on the probability lists. Upon reaching a leaf, the corresponding point is selected using a rejection sampling rule. The following lemma shows that $D^2$-Sampling can be approximated within sublinear time.

**Lemma 4.** *The Adaptive-Sampling Algorithm takes expected $\tilde{O}(k)$ time to sample a data point $x$ with probability $\Omega(\frac{\delta^2(x,C)}{\sqrt{n}\delta^2(P,C)})$.*

### 3.2 Bandit-Based Center Swap

In this subsection, we present the bandit-based center swap method, which aims to accelerate the best swap pair selection process. Specifically, we reformulate the computation for clustering costs as a best-arm identification problem, where each swap pair corresponds to an arm in a multi-armed bandit framework. In this setting, we propose a grouping strategy to estimate the clustering costs. Due to space limits, we leave the intuitive ideas here and leave the detailed analysis in Appendix B.

The proposed algorithm is given in Algorithm 2, which dynamically maintains confidence intervals for the clustering costs of swap pairs via successive sampling (steps 7-8). As the sample sizes increase, suboptimal swap pairs can gradually be eliminated. To achieve sublinear sample sizes, prior bandit methods require strict Gaussian distribution assumptions on each swap pair [6, 34, 33] with constant mean and deviations (see the proofs in Appendix 1 of [6] for details). To relax this, we propose to include virtual swap pairs with zero cost changes and group them with other swap pairs. This enables the elimination of suboptimal swap pairs after sampling $\tilde{O}(k^2)$ points, under a weaker assumption that the swap pairs follow a sub-Gaussian distribution with bounded deviations.

---

**Algorithm 2** BanditCS$(P, k, d, C, \mathcal{S}, \eta, \sigma)$

**Input:** A $k$-means clustering instance $(P, k, d)$, a set $C \subset \mathbb{R}^d$ of clustering centers, a swap group $\mathcal{S}$, a probability parameter $0 < \eta < 1$, and an $\sigma > 0$ representing the prior knowledge on deviations.
**Output:** A group $\mathcal{S}_{\text{swap}}$ of swap pairs.
1: $\mathcal{S}_{\text{swap}} = \mathcal{S}$, $\mathcal{S}_{\text{ref}} = P$, $n_{\text{ref}} = 0$, $\mathcal{B} = \emptyset$.
2: Set $\hat{\mu}_s = 0$ for $s = (u, v) \in \mathcal{S}$.
3: **while** $n_{\text{ref}} < \tilde{O}(k^2)$ and $|\mathcal{S}_{\text{swap}}| > 1$ **do**
4:     Sample a data point $b$ randomly and uniformly from $P$, and set $\mathcal{B} = \mathcal{B} \cup \{b\}$.
5:     For each $s = (u, v) \in \mathcal{S}_{\text{swap}}$, let $G_s(b) = \delta^2(b, C \backslash \{v\} \cup \{u\}) - \delta^2(b, C)$.
6:     **for** $s \in \mathcal{S}_{\text{swap}}$ **do**
7:         $\hat{\mu}_s \leftarrow \frac{n_{\text{ref}} \cdot \hat{\mu}_s + G_s(b)}{n_{\text{ref}} + 1}$.
8:         $\mathcal{F}_s \leftarrow \sigma \cdot \sqrt{\frac{2 \ln(1/\eta)}{n_{\text{ref}} + 1}}$.
9:     $\mathcal{S}_{\text{swap}} = \{s \in \mathcal{S}_{\text{swap}} : \hat{\mu}_s - \mathcal{F}_s \leq \min_{y \in \mathcal{S}_{\text{swap}}} (\hat{\mu}_y + \mathcal{F}_y)\}$, $n_{\text{ref}} \leftarrow n_{\text{ref}} + 1$.
10: **return** $\mathcal{S}_{\text{swap}}$.

---

Let $C$ be the set of the centers opened. Given a data point $b$ randomly and uniformly sampled from the dataset $P$ and a swap pair $s = (u', v')$ ($u' \in P$, $v' \in C$), the clustering cost changes induced by $(u', v')$ on $b$ is defined as $G_s(b) = \delta^2(b, C \backslash \{v'\} \cup \{u'\}) - \delta^2(b, C)$. Under the following assumptions on swap pairs with sub-Gaussian distribution and bounded deviations, with high probability, Algorithm 2 can return a swap pair with minimum clustering cost using $\tilde{O}(k^2)$ samples. Putting all these together, Theorem 1 can be proved (the proofs is given in Appendix B).

**Assumption 1.** *Let $C$ be the set of centers before executing each step 10 of Algorithm 1. For each swap pair $s \in \mathcal{S}_1 \cup \mathcal{S}_2$, it is assumed that $G_s(b)$ is $\sigma_s^2$-sub-Gaussian for a randomly sampled point $b \in P$, with a known bound $\sigma^2 = O((\delta^2(P,C)/|P|)^2)$ and $\sigma^2 \geq \sigma_s^2$.*

**Lemma 5.** *Let $\mathcal{S}$ be a set of swap pairs as input for Algorithm 2, which contains at least one swap pair with a $(1 - 1/100k)$ fraction of cost reduction. For $\eta = O(n^{-4})$, Algorithm 2 can remove non-promising swap pairs in $\mathcal{S}$ with high probability using $\tilde{O}(k^2)$ samples.*

In practice (also in our experiments), since $\sigma$ is usually unknown, we estimate $\sigma_s$ for each swap pair $s$ as the standard deviation $\sigma_s = \text{STD}_{y \in \mathcal{B}}[G_s(y)]$, which can be computed from the set of sampled points in $\mathcal{B}$ accordingly.

## 4 A More Practical Algorithm for $k$-means Clustering

In this section, we propose a more practical local search algorithm for large-scale clustering scenarios. The proposed algorithm is presented in Algorithm 3, where the intuitive idea behind is to combine our bandit-based strategy with Metropolis-Hastings technique to accelerate the sampling process. Due to space limit, we present the high-level idea here and leave the analysis in Appendix C.

---

**Algorithm 3** BanditFastLS$(P, k, d, C, \eta, \sigma)$

---

**Input:** A $k$-means clustering instance $(P, k, d)$, a set $C \subset \mathbb{R}^d$ of random initial clustering centers, a parameter $\eta \in (0, 1)$, a parameter $\sigma$ representing the prior knowledge on deviations.
**Output:** A set $C \subset \mathbb{R}^d$ of clustering centers.
 1: **for** $i = 1$ to $\tilde{O}(k)$ **do**
 2:     Randomly sample a data point $x \in P$.
 3:     **for** $i = 1$ to $\tilde{O}(n/k^2)$ **do**
 4:         Randomly sample a data point $y \in P$.
 5:         Set $x \leftarrow y$ with probability $\min\left\{1, \frac{\delta^2(y,C)}{\delta^2(x,C)}\right\}$.
 6:     $s_x = \arg\min_{q \in C} \delta(q, x)$.
 7:     Randomly sample a point $q'$ from $C \backslash \{s_x\}$.
 8:     $\mathcal{S}_1 = \{(x, s_x), (s_x, s_x)\}$, $\mathcal{S}_2 = \{(x, q'), (q', q')\}$.
 9:     $o_1 = \text{BanditCS}(P, k, d, C, \mathcal{S}_1, \eta, \sigma)$, $o_2 = \text{BanditCS}(P, k, d, C, \mathcal{S}_2, \eta, \sigma)$.
10:     **if** $|o_1| > 1$ and $|o_2| > 1$ **then**
11:         Continue.
12:     **else**
13:         Let $\mathcal{O} = \{o_i : |o_i| = 1, i \in \{1, 2\}\}$, and randomly choose a swap pair $o'$ from $\mathcal{O}$.
14:     $C = C \backslash \{v\} \cup \{u\}$, where $(u, v) \in o'$.
15: **return** $C$.

---

Starting with a random initialization, the BanditFastLS algorithm consists of the following two stages: (1) approximate sampling (steps 2-6); (2) bandit-based swap pair identification (steps 7-14). In the approximate sampling stage, a Metropolis-Hastings strategy is used to approximate the $D^2$-Sampling distribution, where a Markov Chain of bounded length $\tilde{O}(n/k^2)$ is constructed. This reduces the running time for $D^2$-Sampling by eliminating the need to maintain distances between data points and their closest centers. In the bandit-based swap pair identification stage, the bandit-based method proposed in Section 3.2 is adapted to accelerate the swap pair selection process.

Under mild assumptions on optimal cluster sizes and tailed behavior of optimal clusters, we show that the $D^2$-Sampling distribution can be well approximated within $\tilde{O}(n/k^2)$ rounds of sample transitions, where an improved running time can be achieved for swap pair construction. Putting all these together, Theorem 2 can be proved (the proofs are given in Appendix C).

**Assumption 2.** *Assume that the given dataset $P$ is average where each optimal cluster has size $\Omega(k^2)$. For each optimal cluster $P_h^*$, let $c_h^*$ be the optimal center for $P_h^*$. We assume that each $P_h^*$ follows a distribution $F$ over $\mathbb{R}^d$ with exponential tails, i.e., $\exists c, f$ such that $P_r[\delta^2(x, \mu) > a] \le ce^{-fa}$ holds for $x \in P_h^*$, where $c, f$ are constants and $\mu$ is the mean of the distribution.*

**Lemma 6.** *After $\tilde{O}(n/k^2)$ Metropolis-Hastings sampling steps (steps 2-5 of Algorithm 3), if the given clustering instance satisfies the properties in Assumption 2, we can sample a data point $x \in P$ with probability at least $0.5\delta^2(x, C)/\delta^2(P, C)$.*

## 5 Extension to the $k$-median Objective

Our adaptive sampling and bandit-based methods naturally extend to the $k$-median objective. To eliminate the assumptions and sub-Gaussian prior from the bandit process, we propose a weighted

sampling method that can estimate the $k$-median cost of each swap pair using a small number of samples taken from the dynamic tree structure (see Algorithm 12 in Appendix D for details). This method achieves a $(1 \pm \epsilon)$ estimation error with only $\tilde{O}(\sqrt{n}/\epsilon)$ samples. By adapting key ideas from bandit methods, we show that a constant-factor approximation can be obtained in $\tilde{O}(nd + \sqrt{n}k^3)$ time for $k$-median, without relying on any distributional assumptions. Due to space limit, we leave the detailed algorithmic description and analysis in Appendix D. Putting all these together, Theorem 3 can be proved (the proofs are given in Appendix D).

## 6 Experiments

In this section, we compare our proposed BanditFastLS algorithm (the proposed practical algorithm in Section 4) with other state-of-the-art local search methods and coresets methods. For hardware, all the experiments are conducted on a machine with 100 Intel Xeon Gold 6348 CPUs and 1TB memory.

**Datasets.** We evaluate our algorithm on both small and large-scale datasets from prior $k$-means studies [21, 25]. Large-scale datasets include SYN (1M × 2), USC_1990 (2.45M × 68), SUSY (5M × 17) and HIGGS (11M × 27) from UCI Machine Learning Repository [4]. We also include a dataset SIFT (100M × 128) [30]. Small datasets are with sizes ranging from 150 to 50,000 used in [21].

**Algorithms.** We mainly compare our algorithm with the following: LSDS++ [19] (the fastest single-swap local search with constant approximation), LS++ [25], MLS [21] (a fast version of multi-swap local search), and BanditPAM++ [33] (a near-linear time bandit-based algorithm). For our algorithm (BanditFastLS), we set $\eta = 1/(2n^4)$ to satisfy theoretical guarantees. The number of Metropolis-Hastings steps is fixed at 20, and we limit the maximum bandit samples to 50,000.

**Experimental Setup.** We evaluate our BanditFastLS algorithm against other local search algorithms, following the setup in [25, 19, 21]. For fair comparison, the initial centers are randomly selected instead of using a seeding method. For datasets with fewer than 50,000 points, we set $k$ ranging from 3 to 10. For larger datasets, we set $k$ ranging from 10 to 100. Each algorithm is executed for 10 times, and we report the average results with deviations and average runtime. To ensure fairness, all algorithms perform 400 local search steps. Following [25], we apply 10 iterations of a faster Lloyd's algorithm (mini-batch Lloyd's algorithm [31]) after each local search algorithm to finalize the centers. For large-scale datasets (such as HIGGS and SIFT), since LS++'s nearest-neighbor index is memory-prohibitive, we implement it via direct pairwise distance computations.

**Results.** Table 2 and Table 3 (Appendix E) present comparison results on large-scale datasets with more than 1 million points. On the SIFT dataset with 100 million points, even with moderate number of clusters to be opened (i.e., $k = 10$), only MLS, LSDS++, and BanditFastLS algorithms can return feasible solutions within 24 hours. As the parameter $k$ increases, it takes over 24-48 hours for MLS, LSDS++, and LS++ algorithms to return a feasible solution. In contrast, our BanditFastLS algorithm scales efficiently, producing comparable (or even better) clustering results within minutes. Specifically, on dataset SIFT with $k = 10$, BanditFastLS is over 200× faster than competing methods. With $k = 100$, it achieves up to 1000× speedup. By calculating the average results across all large-scale datasets, BanditFastLS is approximately 210× faster than other local search methods, with only a 2% increase in clustering costs. When comparing with the baseline algorithm mini-batch $k$-means++, it also achieves better clustering quality and runtime on dataset SIFT for different choices of parameter $k$. Table 4 (Appendix E) presents results on smaller datasets (datasets with fewer than 50,000 points), where all local search methods perform well, and MLS and LS++ slightly outperform others in clustering quality. However, as data sizes and $k$ increase, BanditFastLS consistently delivers faster runtimes while maintaining comparable clustering performance.

We also evaluate the impact of varying parameters and compare our method with coresets in Appendix E. The input parameters for the BanditFastLS algorithm mainly include the probability parameter $\eta$ and the number of Metropolis-Hastings steps used to approximate the $D^2$-Sampling distribution. Table 5 presents the results for datasets SYN, USC_1990, and SUSY, with a fixed number of clusters $k = 50$ and a maximum sample size of 50,000, while the Metropolis-Hastings steps vary from 20 to 100. The results indicate that increasing the number of Metropolis-Hastings steps can improve the overall clustering quality, with only a slight increase in the running time. However, the overall clustering performance remains consistent, demonstrating that the algorithm is robust to variations in

---

[4]https://archive.ics.uci.edu/ml/index.php

Table 2: Results on the HIGGS (11 million) and SIFT (100 million) datasets with varying $k$, where BanditPAM is excluded as it failed to return a solution within 24 hours in all cases.

| Method | Dataset | $k$ | Cost | Time(s) | Dataset | $k$ | Cost | Time(s) |
|---|---|---|---|---|---|---|---|---|
| $k$-means++ | | | 1.6619E+08±1.3E+06 | 6.51 | | | 1.1160E+13±2.14E+10 | 181.03 |
| MLS | | | 1.5651E+08±5.6E+06 | 451.91 | | | 1.0707E+13±2.26E+10 | 24286.19 |
| LSDS++ | HIGGS | 10 | 1.5618E+08±2.9E+05 | 1162.77 | SIFT | 10 | 1.0770E+13±1.03E+10 | 19994.26 |
| LS++ | | | 1.5704E+08±1.4E+06 | 6902.73 | | | - | >24h |
| Ours | | | **1.5601E+08±5.4E+05** | **5.73** | | | **1.0541E+13±3.88E+10** | **99.81** |
| $k$-means++ | | | 1.5143E+08±1.3E+06 | 9.25 | | | 1.0094E+13±1.76E+10 | 190.74 |
| MLS | | | 1.4273E+08±1.2E+06 | 756.35 | | | - | >24h |
| LSDS++ | HIGGS | 20 | **1.4247E+08±7.2E+05** | 1432.91 | SIFT | 20 | - | >8h |
| LS++ | | | 1.4321E+08±7.2E+05 | 18845.10 | | | - | >24h |
| Ours | | | 1.4335E+08±5.2E+05 | **5.73** | | | **9.6759E+12±1.67E+10** | **101.33** |
| $k$-means++ | | | 1.4296E+08±1.1E+06 | 13.18 | | | 9.6697E+12±5.2E+09 | 276.36 |
| MLS | | | **1.3501E+08±4.0E+04** | 1136.02 | | | - | >48h |
| LSDS++ | HIGGS | 30 | 1.3506E+08±1.6E+05 | 1883.43 | SIFT | 30 | - | >24h |
| LS++ | | | 1.3501E+08±1.0E+05 | 35030.13 | | | - | >48h |
| Ours | | | 1.3527E+08±4.7E+05 | **6.92** | | | **9.2386E+12±1.47E+10** | **105.85** |
| $k$-means++ | | | 1.3119E+08±1.1E+05 | 20.58 | | | 9.1171E+12±1.12E+09 | 381.74 |
| MLS | | | 1.2658E+08±2.1E+05 | 1658.98 | | | - | >48h |
| LSDS++ | HIGGS | 50 | 1.2638E+08±2.9E+05 | 2227.33 | SIFT | 50 | - | >48h |
| LS++ | | | - | >24h | | | - | >48h |
| Ours | | | **1.2636E+08±3.3E+05** | **7.88** | | | **8.7442E+12±1.08E+10** | **125.89** |
| $k$-means++ | | | 1.1998E+08±6.9E+05 | 40.98 | | | 8.4774E+12±5.4E+09 | 677.93 |
| MLS | | | 1.1447E+08±1.0E+05 | 3181.97 | | | - | >48h |
| LSDS++ | HIGGS | 100 | 1.1438E+08±1.6E+05 | 3676.71 | SIFT | 100 | - | >48h |
| LS++ | | | - | >48h | | | - | >48h |
| Ours | | | **1.1434E+08±1.3E+05** | **9.87** | | | **8.1305E+12±4.98E+09** | **163.82** |

the number of Metropolis-Hastings steps. This indicates that fixing the number of transition steps at 20 is sufficient to achieve high-quality clustering results with fast running time.

Tables 7 and 8 present a comparison between coreset construction time and the final clustering time for our algorithm on large-scale datasets. The results show that our algorithm outperforms coreset methods, achieving a 20x speedup on average compared to state-of-the-art coreset construction techniques. In general, the experimental results show that our algorithm is robust across different parameter settings and consistently outperforms coreset-based methods.

## 7    Conclusion, Broader Impact Discussion and Limitations

In this paper, we propose fast and practical clustering algorithms leveraging adaptive sampling and bandit methods. Under mild statistical assumptions on data distributions, the proposed algorithms can achieve better scalability while maintaining comparable clustering quality. Experiments show that, compared with the state-of-the-art local search methods, our proposed algorithm can handle datasets with sizes over 100 million while achieving up to 1000x speedup.

The main purpose of our work is to provide algorithmic insights to accelerate large-scale clustering tasks, with no foreseeable negative societal impacts. One potential limitation is the statistical assumptions. Such assumptions are difficult to validate in practice, as the optimal clusters are usually unknown. Hence, designing fast local search algorithms without any data distribution assumptions is an interesting future direction that deserves further studies. Another potential limitation is the trade-off between clustering quality and efficiency. While our proposed method reduces the combined influence of parameter $k$ and data size $n$ on the runtime complexities, it may not scale well for a sufficiently large $k$ (i.e., $k = \Omega(n)$). This motivates future work on local search algorithms design with strong approximation guarantees and running time independent of the number of clusters $k$.

### Acknowledgments and Disclosure of Funding

This work was supported by National Natural Science Foundation of China (62502545, 62432016), the Science and Technology Innovation Program of Hunan Province (2025RC3207), Open Project of Xiangjiang Laboratory (25XJ03009), and Central South University Research Program of Advanced Interdisciplinary Studies (2023QYJC023). This work was also carried out in part using computing resources at the High Performance Computing Center of Central South University.

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

# Technical Appendices and Supplementary Material

## A   Tree Construction and Rejection Sampling

### A.1   Tree Embedding

The formal description for the tree embedding method is presented in Algorithm 4. Given a $k$-means clustering instance $(P, k, d)$, the algorithm first scales the data points such that the minimum pairwise distance of data points in $P$ is 1 (step 1 of Algorithm 4). Then, it calculates an upper bound $d'_{\max}$ for the maximum pairwise distance between data points in $P$ using a greedy strategy (step 3 of Algorithm 4), which takes $O(nd)$ time. Then, a random shift $0 \leq s \leq d'_{\max}$ (step 4 of Algorithm 4) is applied to each coordinate of the dataset $P$. The root node $r$ of the tree represents an axis-aligned hypercube with side length $2d'_{\max}$ that contains the entire dataset (step 5 of Algorithm 4). Then, the root node $r$ is iteratively decomposed using a tree decomposition algorithm (step 6 of Algorithm 4) to form a tree structure. First, the root node $r$ is decomposed into $2^d$ axis-aligned subcubes (step 3 of Algorithm 5), where the obtained subcubes are associated with side length $d'_{\max}$ and each data point is assigned to the subcube that fully contains its coordinates. For every non-empty subcube, we create a node and add it as a child of the root (step 5 of Algorithm 5). This process is recursively repeated for each node until the side length of the subcube is smaller than $\frac{1}{\sqrt{d}}$ (step 1 of Algorithm 5). The resulting tree structure $\mathcal{T}$ has leaves at the same height, with a maximum tree height of $H = O(\log(d\Delta))$ ($\Delta$ is the aspect ratio of the given clustering instance), and at most $n$ nodes in each layer.

---

**Algorithm 4** Tree-Construction$(P, k, d)$

---

**Input:** A $k$-means instance $(P, k, d)$.
**Output:** An embedded tree structure $\mathcal{T}$.

1: Rescale the data points in $P$ such that the minimum pairwise distance between points in $P$ is 1.
2: Initialize an empty tree structure $\mathcal{T}$.
3: Randomly pick a data point $p$ from $P$, and set $d'_{\max} = 2 \max_{q \in P} \delta(q, p)$.
4: Add a random shift $0 \leq s \leq d'_{\max}$ to each coordinate of each data point in dataset $P$.
5: Create a hypercube $r$ centered at $p$ with side length $2d'_{\max}$, and set $r$ as the root node for $\mathcal{T}$.
6: Call the Tree-Decomposition$(\mathcal{T}, r, 2d'_{\max})$ algorithm to recursively decompose the root node.
7: Assign a level $le(v)$ to each node $v \in \mathcal{T}$, where leaf nodes have level 1 with levels increasing progressively toward the root.
8: **return** $\mathcal{T}$.

---

---

**Algorithm 5** Tree-Decomposition$(\mathcal{T}, \mathcal{B}, l_{\mathcal{B}})$

---

**Input:** A tree structure $\mathcal{T}$, a tree node $\mathcal{B} \in \mathcal{T}$ with side length $l_{\mathcal{B}}$.
**Output:** A decomposed tree structure $\mathcal{T}$.

1: **if** $l_{\mathcal{B}} < 1/\sqrt{d}$ **then**
2:    **STOP**.
3: Decompose $\mathcal{B}$ into $2^d$ axis-aligned subcubes with side length $l_{\mathcal{B}}/2$.
4: Assign data points to the subcubes that fully contain them.
5: Set the non-empty subcubes as children of $\mathcal{B}$.
6: **for** each non-empty child node $\mathcal{B}'$ of $\mathcal{B}$ **do**
7:    Execute Tree-Decomposition$(\mathcal{T}, \mathcal{B}', l_{\mathcal{B}}/2)$.

---

### A.2   Binary Tree Construction

Our objective is to maintain a weight distribution for each node in the tree structure $\mathcal{T}$ to simulate a $D^2$-Sampling process by traversing from the root node to the leaves. However, since non-leaf nodes can have up to $O(n)$ children for the tree structure $\mathcal{T}$, the sampling process on the tree structure may involve $O(n)$ branches, leading to a running time of $\Omega(n \log(d\Delta))$. To address this issue, we propose an algorithm to transform the tree structure $\mathcal{T}$ into a binary tree $\mathcal{T}'$ while preserving the pairwise distance approximation on the tree metric.

The tree conversion algorithm is presented in Algorithm 6. For each node in the tree structure $\mathcal{T}$ with more than 2 children, the tree conversion algorithm creates auxiliary nodes to connect the children and their parent to construct the binary tree structure (steps 3-5 of Algorithm 6). To maintain the expected approximation of pairwise distances, auxiliary nodes are assigned the same level as their parent nodes (step 7 of Algorithm 6). The following lemma shows that this process only adds an $O(\log(d\Delta))$ factor to the tree height while preserving the expected pairwise distances.

**Lemma 2.** *Let $\mathcal{T}'$ be the modified tree structure of the output for Algorithm 6. Then, each vertex in $\mathcal{T}'$ has at most 2 children, and the height of the tree $\mathcal{T}'$ can be bounded by $O(\log^2(d\Delta))$. Additionally, it holds that $\delta(p,q) \leq \delta_{\mathcal{T}'}(p,q)$ and $E[\delta_{\mathcal{T}'}(p,q)] \leq \tilde{O}(d) \cdot \delta(p,q)$.*

*Proof.* Consider an arbitrary non-leaf node $v \in \mathcal{T}$. Let $v_{c_1}, v_{c_2}, ..., v_{c_s}$ be the children of $v$. In Algorithm 6, it adds $s$ auxiliary vertices $v'_{c_1}, v'_{c_2}, ..., v'_{c_s}$. According to the connection rule stated in steps 4-5 of Algorithm 6, $v$ has only one child $v'_{c_1}$. By renaming $v_{c_1}, ..., v_{c_s}$ as $v'_{c_{s+1}}, v'_{c_{s+2}}, ..., v'_{c_{2s}}$, each auxiliary vertex $v'_{c_i}$ for some $i \in [2, s]$ can have at most 2 children $v'_{c_{2i}}$ and $v'_{c_{2i+1}}$. Therefore, each vertex in $\mathcal{T}'$ has at most two children. Since each node has at most $n$ children in $\mathcal{T}$ (where $n$ is the data size), adding $n$ auxiliary nodes is enough to connect the children with the parent and auxiliary nodes. This adds a tree depth of at most $O(\log n)$ for each node. Therefore, the overall height of the tree $\mathcal{T}'$ can be bounded by $O(\log^2(n))$.

For the pairwise distances approximation, we consider any two point $p, q \in P$. Let $v_{p,q}$ be the node representing the lowest common ancestor of $p$ and $q$ in $\mathcal{T}$. Denote $v'_{p,q}$ as the lowest common ancestor of $p$ and $q$ in $\mathcal{T}'$. Since auxiliary nodes maintain the same tree level as their parent nodes, it follows directly that $le(v_{p,q}) = le(v'_{p,q})$. Consequently, the properties of the approximate pairwise distance stated in Lemma 1 also holds for the transformed tree structure $\mathcal{T}'$. □

---

**Algorithm 6** Tree-Conversion($\mathcal{T}$)

---

**Input:** A tree structure $\mathcal{T}$.
**Output:** A new tree structure $\mathcal{T}'$.
  1: Add all nodes of $\mathcal{T}$ into $\mathcal{T}'$, and keep their original levels.
  2: **for** each $v \in \mathcal{T}$ **do**
  3:    Consider each non-leaf node $v \in \mathcal{T}$ with more than 2 children, where its children are denoted as $v_{c_1}, v_{c_2}, ..., v_{c_s}$. Create $s$ auxiliary nodes $v'_{c_1}, v'_{c_2}, ..., v'_{c_s}$.
  4:    For each $i \in [s]$, set the parent of $v_{c_i}$ in $\mathcal{T}'$ as $v'_{c_{\lfloor (i+s)/2 \rfloor}}$.
  5:    For each $i \in [2, s]$, set the parent of $v'_{c_i}$ in $\mathcal{T}'$ as $v'_{c_{\lfloor i/2 \rfloor}}$.
  6:    Set the parent node of $v'_{c_1}$ as $v$.
  7:    Assign each auxiliary node a level that is the same as its parent node.
  8: **return** $\mathcal{T}'$.

---

### A.3  Maintaining a Dynamic Data Structure $\mathcal{D}$

Once the tree structure is constructed, clustering centers are opened by marking the corresponding leaf nodes as active. However, during the local search process, center swaps can significantly influence the tree structure constructed. Hence, to overcome this issue, we propose TREE-INIT, TREE-OPEN and TREE-CLOSE operations to maintain a dynamic tree structure during the local search process while maintaining the pairwise distance approximation. In general, the TREE-INIT operation sets up the necessary components for the dynamic structure $\mathcal{D}$ by traversing from the leaf nodes to the root. TREE-OPEN marks a leaf node as an active center and updates $\mathcal{D}$ along the path from the leaf to the root, while TREE-CLOSE deactivates a marked leaf node and also updates the structure along the path from the leaf to the root. In the following, for better consistency, we also list the properties for the data structure as presented in the main context. Let $C$ be the set of centers dynamically adjusted by the local search process. The data structure $\mathcal{D}$ on $\mathcal{T}'$ should satisfy the following properties.

- Each non-auxiliary node $v$ in the tree $\mathcal{T}'$ is assigned a level $le(v)$, with leaf nodes assigned with level 1. Along any path from a leaf to the root, the level increases by 1 at each non-auxiliary node, while auxiliary nodes retain the same level as their parents.

- Each node $v$ is associated with three values $n_v^l$, $n_v^r$, and $n(v)$. For non-leaf nodes, $n_v^l$ and $n_v^r$ denote the number of leaf nodes in the left and right subtrees of $v$, respectively. $n(v)$ represents the number of active leaf nodes (or opened centers) that are descendants of $v$. Each node $v \in \mathcal{T}'$ is also associated with a weight $w(v)$ where $w(v) = \delta_{\mathcal{T}'}(\mathcal{B}_v, C)$.
- Each non-leaf node $v \in \mathcal{T}'$ is associated with a probability list $p(v)$, which represents the proportion of the total weights contributed by each of its children. If $v$ has only one child, then $p(v) = [1, 0]$. If $v$ has two children, $p(v) = [l_v, r_v]$, where $l_v = \frac{\delta_{\mathcal{T}'}(\mathcal{B}_{l(v)}, C)}{\delta_{\mathcal{T}'}(\mathcal{B}_v, C)}$, $r_v = \frac{\delta_{\mathcal{T}'}(\mathcal{B}_{r(v)}, C)}{\delta_{\mathcal{T}'}(\mathcal{B}_v, C)}$, and $l_v + r_v = 1$.

We first show that the above data structure can be initialized in time $O(nd \log^2(d\Delta))$ by traversing the tree $\mathcal{T}'$ in a bottom up manner, which we call a TREE-INIT operation.

---

**Algorithm 7** TREE-INIT($\mathcal{T}'$)

---

**Input:** A tree structure $\mathcal{T}'$.
**Output:** A tree structure $\mathcal{T}'$ initialized with data structure $\mathcal{D}$.
1: For each leaf node $v \in \mathcal{T}'$, initialize $n_v^l = 1$, $n_v^r = 0$, $n(v) = 0$, $w(v) = d'_{\max}$, where $d'_{\max}$ is the estimation for the maximum pairwise distance during Tree-Construction (Algorithm 4).
2: Enumerate in a bottom-up manner in $\mathcal{T}'$ such that the children of a node is always visited earlier than the parent node.
3: **for** $v \in \mathcal{T}'$ **do**
4:     **if** $v$ is a non-leaf node **then**
5:         **if** $v$ has only one child $v'$ **then**
6:             $n_v^l = n_{v'}^l + n_{v'}^r$, $n_v^r = 0$, $n(v) = 0$, $w(v) = w(v')$, $p(v) = [1, 0]$.
7:         **else**
8:             Let $v_1'$ and $v_2'$ be the left and right child node for $v$.
9:             $n_v^l = n_{v_1'}^l + n_{v_1'}^r$, $n_v^r = n_{v_2'}^l + n_{v_2'}^r$, $n(v) = 0$, $w(v) = w(v_1') + w(v_2')$.
10:             $p(v) = [\frac{n_v^l}{n_v^l + n_v^r}, \frac{n_v^r}{n_v^l + n_v^r}]$.

---

Specifically, for each $v \in \mathcal{T}'$, the level $le(v)$ has been set when performing the Tree-Construction and Tree-Conversion algorithms. By traversing from the leaf nodes to the root, $n_v^l$, $n_v^r$ can also be initialized accordingly (step 6 and steps 8-9 of Algorithm 7). For each node $v \in \mathcal{T}'$, $n(v)$ is set as 0 since there are no active centers (centers that are opened) during the data structure initialization. For each node $v$ with a single child, the probability list $p(v)$ is initialized as $p(v) = [1, 0]$ (i.e, $v$ has only a left child node). On the other hand, for each node $v$ with two children, the list $p(v)$ is initialized as $p(v) = [\frac{n_v^l}{n_v^l + n_v^r}, \frac{n_v^r}{n_v^l + n_v^r}]$. For each leaf node $v \in \mathcal{T}'$, its weights $w(v)$ is set as $w(v) = d'_{\max}$. The weights of the internal nodes are updated by taking a summation of the weights of its children.

Next, we introduce the TREE-OPEN and TREE-CLOSE operations. Given a data point $p \in P$, the TREE-OPEN operation marks the leaf node $v_p$ associated with $p$ as active and then traverses along the path from $v_p$ to the root node for updating the probability list and weights for the nodes. The formal description for the TREE-OPEN process is described in Algorithm 8.

The TREE-OPEN algorithm updates the tree structure $\mathcal{T}'$ when a clustering center is opened. It begins by marking the leaf node $v_p$ corresponding to the opened center as active and setting its weight to zero (step 1 of Algorithm 8). The algorithm then traverses up the tree to the root, updating weights at each node based on whether its child nodes contain opened centers. Depending on whether the child node is a left or right child node, subtree weights are adjusted accordingly (steps 4-15 of Algorithm 8). The total weight of each node is updated as the weight sum of its left and right subtree, and the probability list is updated accordingly (step 15 of Algorithm 8). This process dynamically adapts the tree structure while maintaining the data structure $\mathcal{D}$, ensuring that the $D^2$-Sampling can be approximated without requiring a full tree reconstruction.

Given a node $v$ in a tree structure $\mathcal{T}'$, recall that we use $l(v)$ and $r(v)$ to denote its left child node and right child node, respectively. For the ease of analysis, if $v$ has only one child node $v'$, then we regard $v'$ as the left child for $v$. Let $\mathcal{B}_v$ denote the set of points corresponding to the leaf nodes that are descendants of node $v$. In the following, we use $C$ to denote the set of centers that have been opened before executing a single local search step. Then, a data point $p \in P$ is ready to be inserted to $C$ as a newly selected center. For each center $q \in C \cup \{p\}$, denote $v_q$ as the leaf

---

**Algorithm 8** TREE-OPEN($\mathcal{T}'$, $p$)

---

**Input:** A tree structure $\mathcal{T}'$, a center $p$ to be opened.
**Output:** A tree structure $\mathcal{T}'$ with marked leaves, updated weight, and probability distribution.

1: Let $v_p$ be the leaf node associated with $p$, mark $v_p$ as active and set $w(v_p) = 0$, $n(v_p) = 1$.
2: Traverse the tree towards the root forming a path $v_0, v_1, ..., v_t$, where $v_0 = v_p$ and $v_t = v_r$.
3: **for** each $i \in [t]$ **do**
4:     **if** $v_{i-1}$ is the left child of $v_i$ **then**
5:         **if** $n(v_i) = 0$ **then**
6:             $w_{v_i}^r = 2\sqrt{d} \cdot (2^{le(v_i)} - 2) \cdot n_{v_i}^r$, $w_{v_i}^l = w(v_{i-1})$.
7:         **else**
8:             $w_{v_i}^r = w(r(v_i))$, $w_{v_i}^l = w(v_{i-1})$.
9:     **else**
10:       **if** $n(v_i) = 0$ **then**
11:           $w_{v_i}^l = 2\sqrt{d} \cdot (2^{le(v_i)} - 2) \cdot n_{v_i}^l$, $w_{v_i}^r = w(v_{i-1})$.
12:       **else**
13:           $w_{v_i}^l = w(l(v_i))$, $w_{v_i}^r = w(v_{i-1})$.
14:     $n(v_i) = n(v_i) + 1$, $w(v_i) = w_{v_i}^l + w_{v_i}^r$.
15:     Let $p(v_i) = [l_{v_i}, r_{v_i}]$ be the probability list associated with node $v_i$, and set $l_{v_i} = \frac{w(l(v_i))}{w(l(v_i)) + w(r(v_i))}$ and $r_{v_i} = \frac{w(r(v_i))}{w(l(v_i)) + w(r(v_i))}$.
16: **return** $\mathcal{T}'$.

---

node in $\mathcal{T}'$ that is associated with $q$. Define $\mathcal{L}(q) = \{v_0, v_1, ..., v_t\}$ as the set of the tree nodes representing the path from the leaf node $v_q$ to the root node $v_r$, where $v_0 = v_q$ and $v_t = v_r$. Recall that $\delta_{\mathcal{T}'}(\mathcal{B}_v, C) = \sum_{v' \in \mathcal{B}_v} \delta_{\mathcal{T}'}(v', C)$ is the sum of the tree distances (defined on $\mathcal{T}'$) between the leaf nodes in $\mathcal{B}_v$ to the centers in $C$. The following lemma shows that the probability list maintained for each tree node can well represent the $D^2$-Sampling distribution.

**Lemma A.1.** *For each $q \in C \cup \{p\}$ and a tree node $v_i \in \mathcal{L}(q)$, it holds that* $l_{v_i} = \frac{\delta_{\mathcal{T}'}(\mathcal{B}_{l(v_i)}, C \cup \{p\})}{\delta_{\mathcal{T}'}(\mathcal{B}_{v_i}, C \cup \{p\})}$, $r_{v_i} = \frac{\delta_{\mathcal{T}'}(\mathcal{B}_{r(v_i)}, C \cup \{p\})}{\delta_{\mathcal{T}'}(\mathcal{B}_{v_i}, C \cup \{p\})}$ *and* $w(v_i) = \delta_{\mathcal{T}'}(\mathcal{B}_{v_i}, C \cup \{p\})$.

*Proof.* The proof is based on an induction.

For each center $q \in C \cup \{p\}$, denote $v_q$ as the active leaf node associated with $q$. Consider the base case for $C = \{s\}$ where $s$ is the first center added to $C$. We will show that the properties stated in Lemma A.1 holds for $v_s$. Note that through TREE-INIT, each non-leaf node $v \in \mathcal{T}'$ is assigned a probability list $p(v) = [l_v, r_v]$, where $l_v = \frac{n_v^l}{n_v^l + n_v^r}$, $r_v = \frac{n_v^r}{n_v^l + n_v^r}$. Each leaf node is also assigned a weight $w(v) = d'_{\max}$. For ease of analysis, without loss of generality, we can assume that $v_i \in \mathcal{L}(s)$ is the left child of $v_{i+1} \in \mathcal{L}(s)$. The proof for the base case also uses an induction. Firstly, consider the node $v_1 \in \mathcal{L}(s)$. Since $v_0 = v_s$ is marked as active, the weight for $v_0$ is set as $w(v_0) = 0$ (step 1 of Algorithm 8). Then, for the node $v_1$, we have $\mathcal{B}_{l(v_1)} = \{v_0\}$ and $w_{v_1}^l$ is set as $w_{v_1}^l = w(v_0) = 0$ (step 6 of Algorithm 8), which corresponds to the sum of the distances of data points in $\mathcal{B}_{l(v_1)}$ to the opened center $s$. Then, since $n(v_1)$ is set as 0 through TREE-INIT, we have $w_{v_1}^r = 2\sqrt{d} \cdot (2^2 - 2) \cdot n_{v_1}^r$ (step 6 of Algorithm 8). Since all the paths from leaf nodes to the root node have the same length, it holds that $\delta_{\mathcal{T}'}(a, s) = 4\sqrt{d}$ for each $a \in \mathcal{B}_{r(v_1)}$ and $\delta_{\mathcal{T}'}(\mathcal{B}_{r(v_1)}, s) = 4\sqrt{d} \cdot n_{v_1}^r = w_{v_1}^r$. By updating $w(v_1)$ as $w_{v_1}^l + w_{v_1}^r$, we have $w(v_1) = \delta_{\mathcal{T}'}(\mathcal{B}_{v_1}, s)$.

Now, assume for each $i \in [j-1]$ and $p(v_i) = [l_{v_i}, r_{v_i}]$, it holds that $l_{v_i} = \frac{\delta_{\mathcal{T}'}(\mathcal{B}_{l(v_i)}, s)}{\delta_{\mathcal{T}'}(\mathcal{B}_{v_i}, s)}$, $r_{v_i} = \frac{\delta_{\mathcal{T}'}(\mathcal{B}_{r(v_i)}, s)}{\delta_{\mathcal{T}'}(\mathcal{B}_{v_i}, s)}$ and $w(v_i) = \delta_{\mathcal{T}'}(\mathcal{B}_{v_i}, s)$. Then, we consider the node $v_j$. Since $n(v_j)$ is set as zero through TREE-INIT, we have $w_{v_j}^l = w(v_{j-1}) = \delta_{\mathcal{T}'}(\mathcal{B}_{v_{j-1}}, s)$ and $w_{v_j}^r = 2\sqrt{d} \cdot (2^{le(v_j)} - 2) \cdot n_{v_j}^r$ (step 6 of Algorithm 8). Since all the leaf nodes are at the same height and the auxiliary nodes do not increase the node level, it holds that $\delta_{\mathcal{T}'}(a, s) = 2\sqrt{d} \cdot (2^{le(v_j)} - 2)$ for each $a \in \mathcal{B}_{r(v_j)}$. Hence, we can get that $\delta_{\mathcal{T}'}(\mathcal{B}_{r(v_j)}, s) = 2\sqrt{d} \cdot (2^{le(v_j)} - 2) \cdot n_{v_j}^r$. Therefore, we have $w_{v_j}^l = \delta_{\mathcal{T}'}(\mathcal{B}_{l(v_j)}, s)$ and $w_{v_j}^r = \delta_{\mathcal{T}'}(\mathcal{B}_{r(v_j)}, s)$. Then, it holds that $w(v_j) = w_{v_j}^l + w_{v_j}^r = \delta_{\mathcal{T}'}(\mathcal{B}_{v_j}, s)$ (step 14 of Algorithm 8),

which proves the case for $v_j$. Thus, we can conclude that $C = \{s\}$, where $s$ is the first center added to $C$, holds for Lemma A.1.

Next, given a set $C'$ of opened centers, we assume that for each $q \in C'$ and a tree node $v_i \in \mathcal{L}(q)$, it holds that $l_{v_i} = \frac{\delta_{\mathcal{T}'}(\mathcal{B}_{l(v_i)}, C')}{\delta_{\mathcal{T}'}(\mathcal{B}_{v_i}, C')}$, $r_{v_i} = \frac{\delta_{\mathcal{T}'}(\mathcal{B}_{r(v_i)}, C')}{\delta_{\mathcal{T}'}(\mathcal{B}_{v_i}, C')}$ and $w(v_i) = \delta_{\mathcal{T}'}(\mathcal{B}_{v_i}, C')$. Let $g$ be a center that will be added to $C'$, we will show the case holds for $C' \cup \{g\}$.

The TREE-OPEN operation traverses from the leaf node $v_g$ to the root node $v_r$ to update the weights and probability distributions. For ease of analysis, without loss of generality, we can also assume that $v_i \in \mathcal{L}(g)$ is the left child of $v_{i+1} \in \mathcal{L}(g)$. The proof for $C' \cup \{g\}$ is also based on an induction. We first consider the node $v_1$. Since $v_0 = v_g$ is marked as active, the weight for $v_0$ is set as $w(v_0) = 0$ (step 1 of Algorithm 8). We have $\mathcal{B}_{l(v_1)} = \{v_0\}$ and $w_{v_1}^l$ is set as 0 (step 6 or step 8 of Algorithm 8), which corresponds to $\delta_{\mathcal{T}'}(\mathcal{B}_{l(v_1)}, C' \cup \{g\})$. Then, for analyzing the value of $w_{v_1}^r$, there are two subcases that may happen: (1) $n(v_1) > 0$; (2) $n(v_1) = 0$. If subcase (1) happens, $v_1$ must have been visited before $g$ is added to $C'$ by the TREE-OPEN operation, and $v_1$ belongs to $\mathcal{L}(q')$ for some $q' \in C$. According to the induction assumption, we have $w_{v_1}^r = w(r(v_1)) = \delta_{\mathcal{T}'}(\mathcal{B}_{r(v_1)}, C')$ (step 8 of Algorithm 8). Note that since $g$ does not belong to $\mathcal{B}_{r(v_1)}$, adding $g$ to $C'$ will not induce distance change (no lower common ancestors are added) for the nodes in $\mathcal{B}_{r(v_1)}$. Therefore, $w_{v_1}^r = \delta_{\mathcal{T}'}(\mathcal{B}_{r(v_1)}, C') = \delta_{\mathcal{T}'}(\mathcal{B}_{r(v_1)}, C' \cup \{g\})$. If subcase (2) happens, according to step 6 of Algorithm 8, we also have $w_{v_1}^r = \delta_{\mathcal{T}'}(\mathcal{B}_{r(v_1)}, g)$. Note that since $n(v_1) = 0$, it holds trivially that $w_{v_1}^r = \delta_{\mathcal{T}'}(\mathcal{B}_{r(v_1)}, \{g\}) = \delta_{\mathcal{T}'}(\mathcal{B}_{r(v_1)}, C \cup \{g\})$. For both cases, by updating $w(v_1)$ as $w_{v_1}^l + w_{v_1}^r$, we have $w(v_1) = \delta_{\mathcal{T}'}(\mathcal{B}_{v_1}, C' \cup \{g\})$.

Assume for $i \in [j-1]$ and $p(v) = [l_{v_i}, r_{v_i}]$, it holds that $l_{v_i} = \frac{\delta_{\mathcal{T}'}(\mathcal{B}_{l(v_i)}, C' \cup \{g\})}{\delta_{\mathcal{T}'}(\mathcal{B}_{v_i}, C' \cup \{g\})}$, $r_{v_i} = \frac{\delta_{\mathcal{T}'}(\mathcal{B}_{r(v_i)}, C' \cup \{g\})}{\delta_{\mathcal{T}'}(\mathcal{B}_{v_i}, C' \cup \{g\})}$ and $w(v_i) = \delta_{\mathcal{T}'}(\mathcal{B}_{v_i}, C' \cup \{g\})$. We then consider the node $v_j$. We have $w_{v_j}^l = w(v_{j-1}) = \delta_{\mathcal{T}'}(\mathcal{B}_{v_{j-1}}, C' \cup \{g\})$ according to the induction assumption (step 6 or step 8 of Algorithm 8). Then, there are also two subcases that may happen: (1) $n(v_j) > 0$; (2) $n(v_j) = 0$. If subcase (1) happens, $v_j$ must have been visited by the TREE-OPEN operation and $v_j$ belongs to $\mathcal{L}(q')$ for some $q' \in C'$. According to the induction assumption, we have $w_{v_j}^r = w(r(v_j)) = \delta_{\mathcal{T}'}(\mathcal{B}_{r(v_j)}, C')$. Note that since $g$ does not belong to $\mathcal{B}_{r(v_j)}$, adding $g$ to $C'$ will not induce distance change for the leaf nodes in $\mathcal{B}_{r(v_j)}$ and hence $w_{v_j}^r = \delta_{\mathcal{T}'}(\mathcal{B}_{r(v_j)}, C') = \delta_{\mathcal{T}'}(\mathcal{B}_{r(v_j)}, C' \cup \{g\})$. If subcase (2) happens, we also have $w_{v_j}^r = \delta_{\mathcal{T}'}(\mathcal{B}_{r(v_j)}, g)$. Note that since $n(v_j) = 0$, it holds trivially that $w_{v_j}^r = \delta_{\mathcal{T}'}(\mathcal{B}_{r(v_j)}, g) = \delta_{\mathcal{T}'}(\mathcal{B}_{r(v_j)}, C' \cup \{g\})$. By updating $w(v_j)$ as $w_{v_j}^l + w_{v_j}^r$, we have $w(v_j) = \delta_{\mathcal{T}'}(\mathcal{B}_{v_j}, C' \cup \{g\})$.

Finally, consider the tree nodes in $\mathcal{L}(q')$ for some $q' \in C'$ that do not belong to $\mathcal{L}(g)$. For each such node $v$, a key observation is that $g$ does not belong to $\mathcal{B}_v$. Hence, for each node $v' \in \mathcal{B}_v$, the tree distance between $v'$ to its closest opened center on the tree remains the same, where we have $l_v = \frac{\delta_{\mathcal{T}'}(\mathcal{B}_{l(v)}, C \cup \{g\})}{\delta_{\mathcal{T}'}(\mathcal{B}_v, C \cup \{g\})}$ and $r_v = \frac{\delta_{\mathcal{T}'}(\mathcal{B}_{r(v)}, C \cup \{g\})}{\delta_{\mathcal{T}'}(\mathcal{B}_v, C \cup \{g\})}$, and $w(v) = \delta_{\mathcal{T}'}(\mathcal{B}_v, C \cup \{g\})$.

Putting all these together and using the induction, Lemma A.1 can be proved. $\qquad \square$

**Lemma A.2.** *Assume that the current set of active centers (or opened centers) on the tree $\mathcal{T}'$ is $C$. Given a data point $p \in P$, TREE-OPEN operation runs in time $O(\log^2(d\Delta))$ to update the data structure $\mathcal{D}$ such that each non-leaf node $v \in \mathcal{T}'$ holds a probability list $p(v) = [l_v, r_v]$, where $l_v = \frac{\delta_{\mathcal{T}'}(\mathcal{B}_{l(v)}, C \cup \{p\})}{\delta_{\mathcal{T}'}(\mathcal{B}_v, C \cup \{p\})}$ and $r_v = \frac{\delta_{\mathcal{T}'}(\mathcal{B}_{r(v)}, C \cup \{p\})}{\delta_{\mathcal{T}'}(\mathcal{B}_v, C \cup \{p\})}$.*

*Proof.* According to Lemma A.1, the properties stated in Lemma A.2 hold for each node $v \in \mathcal{L}(q)$ where $q \in C \cup \{p\}$. Then, we only need to consider the tree nodes that do not belong to $\mathcal{L}(q)$ for any $q \in C \cup \{p\}$. For each such node $v$, observe that $v$ is never visited during the TREE-OPEN operations, and hence $\mathcal{B}_v$ does not contain any leaf node that are associated with the centers in $C$. For each such node $v$, since no leaf nodes that are descendants of $v$ have been marked as active nodes, the tree distances between any leaf node $v' \in \mathcal{B}_v$ to the opened centers in $C$ are the same. According to TREE-INIT, we have $l_v = \frac{n_v^l}{n_v^l + n_v^r}$ and $r_v = \frac{n_v^r}{n_v^l + n_v^r}$. Then, it holds that $l_v = \frac{n_v^l}{n_v^l + n_v^r} = \frac{\delta_{\mathcal{T}'}(\mathcal{B}_{l(v)}, C \cup \{p\})}{\delta_{\mathcal{T}'}(\mathcal{B}_v, C \cup \{p\})}$ and $r_v = \frac{n_v^r}{n_v^l + n_v^r} = \frac{\delta_{\mathcal{T}'}(\mathcal{B}_{r(v)}, C \cup \{p\})}{\delta_{\mathcal{T}'}(\mathcal{B}_v, C \cup \{p\})}$.

As for the running time, the TREE-OPEN process traverses from the leaf node associated with $p$ to the root node. Since each node along the path has at most 2 children and the tree structure $\mathcal{T}'$ has height $O(\log^2(d\Delta))$, the running time can be bounded by $O(\log^2(d\Delta))$. $\qquad\square$

Then, we can present the TREE-CLOSE operation. Based on a set $C$ of centers that have been opened, the TREE-CLOSE operation aims to delete a center $p \in C$. Specifically, the TREE-CLOSE operation deactivates the leaf node $v_p$ associated with $p$ and then traverses along the path from $v_p$ to the root node to update the probability list and weights for the nodes. The formal description for the TREE-CLOSE process is described in Algorithm 9. The TREE-CLOSE algorithm updates the tree structure $\mathcal{T}'$ when a clustering center is removed, ensuring the data structure $\mathcal{D}$ remains consistent. It begins by deactivating the leaf node $v_p$ associated with the $p$ (step 2 of Algorithm 9). Then, it identifies the first ancestor $v'$ that still has at least two remaining active leaf node in its descendants (step 3 of Algorithm 9). If $v'$ does not exist, it assigns a maximum weight $d'_{\max}$ for $v_p$ (step 5 of Algorithm 9). Otherwise, it updates the weight based on the tree level of $v'$ (step 7 of Algorithm 9). The algorithm then traverses from $v_p$ to the root, adjusting weights and probability distributions at each node to reflect the center closure. During the updates, the weights are recalculated to maintain the hierarchical structure (steps 9-25 of Algorithm 9). The probability list is updated accordingly to ensure that the tree can still well approximate the $D^2$-Sampling process. By dynamically modifying the weight distributions and probability values, TREE-CLOSE ensures that center closures are properly handled without requiring a full tree reconstruction. The following lemma shows that the probability list maintained for each tree node can still well represent the $D^2$-Sampling distribution based on tree distances between data points. Since the proof for Lemma A.3 is similar to those for Lemma A.2 and Lemma A.1, we omit the proof here.

---

**Algorithm 9** TREE-CLOSE($\mathcal{T}', p$)

---

**Input:** A tree structure $\mathcal{T}'$.
**Output:** A tree structure $\mathcal{T}'$ with updated leaves, weights and probability distributions.
1: Let $v_0, v_1, ..., v_t$ be a path along the leaf node $v_p$ associated with $p$ to the root node $v_r$, where $v_0 = v_p$ and $v_t = v_r$.
2: Deactivate $v_p$, and set $n(v_p) = n(v_p) - 1$.
3: Traverse along the nodes $v_0, ..., v_t$ to find the first node $v'$ such that $n(v') > 1$.
4: **if** $v'$ does not exist **then**
5: $\quad$ Set $w(v_p) = d'_{\max}$.
6: **else**
7: $\quad$ $w(v_p) = 2\sqrt{d} \cdot (2^{le(v')} - 2)$.
8: **for** each $i \in [t]$ **do**
9: $\quad$ **if** $v_{i-1}$ is the left child of $v_i$ **then**
10: $\quad\quad$ **if** $n(v_i) - 1 = 0$ **then**
11: $\quad\quad\quad$ **if** $v'$ does not exist **then**
12: $\quad\quad\quad\quad$ $w^r_{v_i} = d'_{\max} \cdot n^r_{v_i}, w^l_{v_i} = d'_{\max} \cdot n^l_{v_i}$.
13: $\quad\quad\quad$ **else**
14: $\quad\quad\quad\quad$ $w^r_{v_i} = 2\sqrt{d} \cdot (2^{le(v')} - 2) \cdot n^r_{v_i}, w^l_{v_i} = 2\sqrt{d} \cdot (2^{le(v')} - 2) \cdot n^l_{v_i}$.
15: $\quad\quad$ **else**
16: $\quad\quad\quad$ $w^r_{v_i} = w(r(v_i)), w^l_{v_i} = w(v_{i-1})$.
17: $\quad$ **else**
18: $\quad\quad$ **if** $n(v_i) - 1 = 0$ **then**
19: $\quad\quad\quad$ **if** $v'$ does not exist **then**
20: $\quad\quad\quad\quad$ $w^r_{v_i} = d'_{\max} \cdot n^r_{v_i}, w^l_{v_i} = d'_{\max} \cdot n^l_{v_i}$.
21: $\quad\quad\quad$ **else**
22: $\quad\quad\quad\quad$ $w^r_{v_i} = 2\sqrt{d} \cdot (2^{le(v')} - 2) \cdot n^r_{v_i}, w^l_{v_i} = 2\sqrt{d} \cdot (2^{le(v')} - 2) \cdot n^l_{v_i}$.
23: $\quad\quad$ **else**
24: $\quad\quad\quad$ $w^l_{v_i} = w(l(v_i)), w^r_{v_i} = w(v_{i-1})$.
25: $\quad$ $n(v_i) = n(v_i) - 1, w(v_i) = w^l_{v_i} + w^r_{v_i}$
26: $\quad$ Let $p(v_i) = [l_{v_i}, r_{v_i}]$ be the probability list associated with node $v_i$, and set $l_{v_i} = \frac{w(l(v_i))}{w(l(v_i)) + w(r(v_i))}$ and $r_{v_i} = \frac{w(r(v_i))}{w(l(v_i)) + w(r(v_i))}$.
27: **return** $\mathcal{T}'$.

---

**Lemma A.3.** *Assume that the current set of centers opened is $C$, where $C$ also corresponds to the set of active leaf nodes in $\mathcal{T}'$. Given a center $c \in C$, TREE-CLOSE operation runs in time $O(\log^2(d\Delta))$ to update the tree structure such that each non-leaf node $v \in \mathcal{T}'$ holds a probability list $p(v) = [l_v, r_v]$, where $l_v = \frac{\delta_{\mathcal{T}'}(\mathcal{B}_{l(v)}, C\backslash\{p\})}{\delta_{\mathcal{T}'}(\mathcal{B}_v, C\backslash\{p\})}$ and $r_v = \frac{\delta_{\mathcal{T}'}(\mathcal{B}_{r(v)}, C\backslash\{p\})}{\delta_{\mathcal{T}'}(\mathcal{B}_v, C\backslash\{p\})}$.*

Putting all these together, Lemma 3 can be proved.

## A.4 The Adaptive Sampling Strategy

Based on the TREE-OPEN and TREE-CLOSE operations, during the whole local search process, we can always maintain a tree $\mathcal{T}'$ such that each non-leaf node $v \in \mathcal{T}'$ holds a probability list $p(v) = [l_v, r_v]$, where $l_v = \frac{\delta_{\mathcal{T}'}(\mathcal{B}_{l(v)}, C)}{\delta_{\mathcal{T}'}(\mathcal{B}_v, C)}$ and $r_v = \frac{\delta_{\mathcal{T}'}(\mathcal{B}_{r(v)}, C)}{\delta_{\mathcal{T}'}(\mathcal{B}_v, C)}$. However, since the tree structure only guarantees the approximation loss in expectation (see Lemma 1 and Lemma 2 for details), directly performing a sampling on the tree structure may lead to unbounded success probability. Partly inspired by previous work [15], we propose an adaptive sampling algorithm to overcome this issue, which coverts the expected distance-based sampling distribution into the true distance-based sampling distribution. The formal adaptive sampling algorithm is presented in Algorithm 10.

---

**Algorithm 10** Adaptive-Sampling($\mathcal{T}', C$)

---

**Input:** A tree structure $\mathcal{T}'$, a set $C$ of clustering centers.
**Output:** A sampled point $x$.
 1: $\mathcal{S} = \emptyset$.
 2: **while** $\mathcal{S} = \emptyset$ **do**
 3:     Let $v$ be the root node of the tree $\mathcal{T}'$.
 4:     **while** $v$ is not a leaf node **do**
 5:         Let $p(v) = [l_v, r_v]$ be the probability list for $v$.            ▷ $p(v)$ is detailed in Section 3.1
 6:         Update $v$ to be its left child with probability $l_v$ and to be its right child with probability $r_v$.
 7:     Let $x$ be the point associated with the leaf node $v$.
 8:     Add $x$ to $\mathcal{S}$ with probability $\min\left\{1, \frac{\delta(x,C)}{\delta_{\mathcal{T}'}(x,C)}\right\}$, where $\delta_{\mathcal{T}'}(x, C)$ is the tree distance between point $x$ to $C$ in the tree structure $\mathcal{T}'$.            ▷ Tree distance is detailed in Section 3.1
 9: **return** $x \in \mathcal{S}$.

---

The Adaptive-Sampling algorithm samples a data point $x$ by leveraging the tree structure $\mathcal{T}'$ to align with the $D^2$-Sampling distribution. The process begins by initializing an empty set $\mathcal{S}$ and repeatedly drawing samples until a valid point is accepted (steps 4-8 of Algorithm 10). The sampling process starts from the root node and traverses the tree downward, selecting the left or right child based on the probability list associated with each node (steps 5-6 of Algorithm 10). Once a leaf node $v$ is reached, it identifies the corresponding data point $x$ and applies a rejection sampling step (step 8 of Algorithm 10), where $x$ is accepted with probability $\min\left\{1, \frac{\delta(x,C)}{\delta_{\mathcal{T}'}(x,C)}\right\}$. The process repeats until a point is successfully accepted, where the sampled point is then returned by the algorithm.

The following lemmas show that the rejection sampling process can well approximate the $D^2$-Sampling distribution with bounded expected running time.

**Lemma A.4.** *The probability of inserting a point $x$ to the set $\mathcal{S}$ in Adaptive-Sampling algorithm is $\frac{\delta(x,C)}{\sum_{p\in P} \delta(p,C)}$.*

*Proof.* According to Lemma 2, we have $\delta_{\mathcal{T}'}(x, C) \geq \delta(x, C)$, and thus $\frac{\delta(x,C)}{\delta_{\mathcal{T}'}(x,C)} \leq 1$. Based on Lemma A.2 and Lemma A.3, during the local search process, the tree $\mathcal{T}'$ always maintain a probability list $p(v) = [l_v, r_v]$ for each non-leaf node $v \in \mathcal{T}'$ where $l_v = \frac{\delta_{\mathcal{T}'}(\mathcal{B}_{l(v)}, C)}{\delta_{\mathcal{T}'}(\mathcal{B}_v, C)}$ and $r_v = \frac{\delta_{\mathcal{T}'}(\mathcal{B}_{r(v)}, C)}{\delta_{\mathcal{T}'}(\mathcal{B}_v, C)}$. Hence, the probability of sampling a data point $x \in P$ becomes

$$\frac{\delta_{\mathcal{T}'}(x, C)}{\sum_{p\in P} \delta_{\mathcal{T}'}(p, C)} \cdot \frac{\delta(x, C)}{\delta_{\mathcal{T}'}(x, C)},$$

which is independent of the term $\delta_{\mathcal{T}'}(x,C)$. Since the term $\sum_{p\in P}\delta_{\mathcal{T}'}(p,C)$ is the same for each $x \in P$ and does not depend on $x$, the probability of adding any point $x$ to $\mathcal{S}$ is $\frac{\delta(x,C)}{\sum_{p\in P}\delta(x,C)}$ according to the rejection sampling rule. □

**Lemma A.5.** *The expected number of iterations for Adaptive-Sampling to return a sampled point* $x \in P$ *can be bounded by* $\tilde{O}(d)$.

*Proof.* Consider a single step for sampling a data point $x \in P$ using Adaptive-Sampling. According to Lemma A.4, the probability of adding a data point $x \in P$ to $\mathcal{S}$ is $\frac{\delta(x,C)}{\sum_{p\in P}\delta_{\mathcal{T}'}(p,C)}$. Denote the random variable $N$ as the number of steps executed when at least one data point $x \in P$ is added to $\mathcal{S}$. Let $q$ be the probability that a data point $x \in P$ is added to $\mathcal{S}$ within a single sampling step (i.e., steps 3-8 of Algorithm 10). Then, we have

$$q = \frac{\sum_{x\in P}\delta(x,C)}{\sum_{p\in P}\delta_{\mathcal{T}'}(p,C)}.$$

This indicates that $q$ data points are added to $\mathcal{S}$ in expectation within a single sampling step. We can get that

$$E[N] = \sum_{i=1}^{+\infty}(1-q)^{i-1}\cdot q \cdot i$$
$$= 1/q.$$

By combining with Lemma 2, we have $E[N] \leq \frac{\sum_{p\in P}\delta_{\mathcal{T}'}(p,C)}{\sum_{x\in P}\delta(x,C)}$ for a fixed tree $\mathcal{T}'$. By taking the expectation over the randomness of tree-embedding, we have $E[N] \leq \frac{\sum_{p\in P}\delta_{\mathcal{T}'}(p,C)}{\sum_{x\in P}\delta(x,C)} \leq \tilde{O}(d)$, which proves the lemma. □

**Lemma A.6.** *The Adaptive-Sampling takes expected time* $\tilde{O}(k)$ *to sample a data point* $x \in P$ *with probability at least* $\frac{\delta^2(x,C)}{\sqrt{n}\delta^2(P,C)}$.

*Proof.* According to Lemma A.4, we can get that the probability of sampling a data point $x \in P$ is $\frac{\delta(x,C)}{\sum_{p\in P}\delta(p,C)}$. By using Caushy Inequality, we have

$$\left(\sum_{p\in P}\delta^2(p,C)\right)\left(\sum_{p\in P}1^2\right) \geq \left(\sum_{p\in P}\delta(p,C)\right)^2,$$

which implies that

$$\delta(P,C) \leq \sqrt{n}\cdot\sqrt{\delta^2(P,C)}.$$

Then, we can get that each data point $x \in P$ can be sampled with probability at least $\frac{\delta(x,C)}{\delta(P,C)} \geq \frac{\delta(x,C)}{\sqrt{n}\cdot\sqrt{\delta^2(P,C)}} = \frac{1}{\sqrt{n}}\cdot\sqrt{\frac{\delta^2(x,C)}{\delta^2(P,C)}} \geq \frac{\delta^2(x,C)}{\sqrt{n}\delta^2(P,C)}$, where the last step follows from $\frac{\delta^2(x,C)}{\delta^2(P,C)} \leq 1$.

As for the running time, TREE-CLOSE and TREE-OPEN operations take time $O(\log^2(d\Delta))$. Each sampling step traversing from the root node of the tree to the leaf node of the tree takes time $O(\log^2(d\Delta))$ (steps 3-7 of Algorithm 10). Querying the true distance between the sampled data point $x$ to its closest center in $C$ takes $O(kd)$ time (step 8 of Algorithm 10). According to Lemma A.5, the expected number of iterations for sampling a data point $x$ is $\tilde{O}(d)$. Finally, with $\tilde{O}(nd)$ preprocessing time before tree construction, the aspect ratio $\Delta$ and the dimension $d$ can be reduced to $\text{poly}(n,d)$ and $O(\log n)$, respectively. The overall expected running time for sampling a data point $x \in P$ using Adaptive-Sampling is $\tilde{O}(k)$. □

# B  Bandit-Based Center Swap

In this subsection, we present the bandit-based center swap method. For the existing local search methods, even with sampling-based swap pair construction, a clustering cost estimation process is required to pick the best swap pair among the candidate swap pairs. However, in the worst case, this process requires scanning the entire dataset for distance and data point reassignment calculations, which leads to a running time of $\Omega(nd)$ for each center swap step.

To further accelerate the best swap pair selection process, our proposed bandit-based method (Algorithm 2) can identify the swap pair inducing the best clustering cost reduction with sub-linear runtime assuming sub-Gaussian prior on swap pairs. The high-level idea behind is to reformulate the deterministic clustering cost estimation process as a statistical bandit problem. By regarding each swap pair as an arm in the bandit framework, selecting the swap pair that yields the best clustering cost reduction can be modeled as a best-arm identification problem. To give the detailed analysis, we first introduce the following folklores for statistics.

**Lemma B.1.** *If $a \leq X \leq b$ holds with probability $1$ for a random variable $X$, then $X$ is a $(\frac{b-a}{2})^2$-sub-Gaussian[5], where $(\frac{b-a}{2})^2$ is the variance proxy.*

**Lemma B.2.** (**Hoeffding's Inequality**) *Let $X_1, ..., X_n$ be $n$ i.i.d. sub-Gaussian random variables with variance proxy $\sigma^2$. Then, for any $\epsilon > 0$, it holds that $P_r\left(\frac{1}{n}\sum_{i=1}^{n} X_i - E[X] \geq \epsilon\right) \leq e^{-\frac{n\epsilon^2}{2\sigma^2}}$.*

We begin the formal analysis of the proposed bandit-based center swap method by first presenting a reduction of the best swap pair selection problem to a bandit problem. Given a group $\mathcal{S}_i$ ($i = 1, 2$) containing several swap pairs as the input for Algorithm 2, let $\mathcal{S}_{\text{swap}} = \mathcal{S}_i$ and $\mathcal{S}_{\text{ref}} = P$ be the sets of the reference swap pairs and reference points, respectively. In the modeled bandit problem, each swap pair $s \in \mathcal{S}_{\text{ref}}$ is regarded as an arm and the goal is to identify the swap pair $(u, v) \in \mathcal{S}_{\text{swap}}$ with the maximum clustering cost reduction, i.e., $(u, v) = \arg\min_{(u', v') \in \mathcal{S}_{\text{swap}}} \delta^2(P, C \cup \{u'\}\backslash\{v'\})$. Given a data point $b$ randomly sampled from the dataset $P$, for each swap pair $s = (u', v') \in \mathcal{S}_{\text{swap}}$, the clustering cost change induced by $(u', v')$ on $b \in \mathcal{S}_{\text{ref}}$ is defined as $G_s(b) = \delta^2(b, C\backslash\{v'\} \cup \{u'\}) - \delta^2(b, C)$. Given a set $\mathcal{H}$ of data points, let $G_s(\mathcal{H}) = \sum_{b \in \mathcal{H}} G_s(b)$. Define $\mu_s = G_s(P)/|\mathcal{S}_{\text{ref}}|$. The best swap pair $s$ also has the smallest value of $G_s(P)/|\mathcal{S}_{\text{ref}}|$.

The bandit-based center swap algorithm is outlined in Algorithm 2 (denoted as the BanditCS algorithm). BanditCS algorithm iteratively samples data points from the referenced points to estimate the average changes in clustering cost for the swap pairs, where a confidence interval is maintained for each candidate swap pair $s \in \mathcal{S}_{\text{swap}}$. In each iteration, the algorithm updates both the mean estimate $\hat{\mu}_s$ of the average cost reduction and the interval length $\mathcal{F}_s$ (step 7 of Algorithm 2) for each $s \in \mathcal{S}_{\text{swap}}$ based on the sampled points. From these updates, a confidence interval $[\hat{\mu}_s - \mathcal{F}_s, \hat{\mu}_s + \mathcal{F}_s]$ is computed (step 8 of Algorithm 2). The confidence interval captures the uncertainty around the estimated average clustering cost reduction based on the current samples. As more samples are collected, the intervals are narrowed to reflect the true average clustering reduction for the swap pairs. The algorithm then eliminates swap pairs with less promising estimates (step 9 of Algorithm 2). This process continues until only one swap pair remains or a predefined number of sample sizes is reached. We show that under the sub-Gaussian prior on swap pairs with significant clustering cost reduction, the bandit-based algorithm (Algorithm 2) can always return the best swap pair with high probability.

**Assumption 1.** *Let $C$ be the set of centers before executing each step 10 of Algorithm 1. For each swap pair $s \in \mathcal{S}_1 \cup \mathcal{S}_2$, it is assumed that $G_s(b)$ is $\sigma_s^2$-sub-Gaussian for a randomly sampled point $b \in P$, with a known bound $\sigma^2 = O((\delta^2(P, C)/|P|)^2)$ and $\sigma^2 \geq \sigma_s^2$.*

**Lemma 5.** *Let $\mathcal{S}$ be a set of swap pairs as input for Algorithm 2, which contains at least one swap pair with a $(1 - 1/100k)$ fraction of cost reduction. For $\eta = O(n^{-4})$, Algorithm 2 can remove non-promising swap pairs in $\mathcal{S}$ with high probability using $\tilde{O}(k^2)$ samples.*

*Proof.* We first prove that the best swap pair will never be eliminated during the whole bandit process with high probability. The proof begins by showing that with probability at least $1 - 1/n^2$, all confidence intervals maintained by Algorithm 2 can well approximate the real mean of the clustering cost change for each swap pair $s \in \mathcal{S}_{\text{swap}}$. Let $\mathcal{B}$ denote the set of the sampled points obtained after

---

[5]A random variable $X$ is a $\sigma^2$-sub-Gaussian if $P_r[X > t] \leq 2e^{-\frac{t^2}{\sigma^2}}$

executing step 4 of Algorithm 2. Then, we consider a fixed swap pair $s \in \mathcal{S}_{\text{swap}}$. Let $\hat{\mu}_s$ be the average of the $|\mathcal{B}|$ i.i.d. samples in $\mathcal{B}$ drawn from $\mathcal{S}_{\text{ref}}$ where $\mathcal{S}_{\text{ref}} = P$. Since each $G_s(b)$ has a bounded value, random variable $X_s = G_s(\mathcal{B})$ can be regarded as a $\sigma_s^2$-sub-Gaussian variable for some $\sigma_s^2 > 0$. By using the Hoeffding's Inequality (Lemma B.2), we can get that

$$P_r(|\mu_s - \hat{\mu}_s| > \mathcal{F}_s) \le 2e^{-\frac{|\mathcal{B}| \cdot \mathcal{F}_s^2}{2\sigma_s^2}}.$$

According to the definition of $\mathcal{F}_s$ (step 8 of Algorithm 2), we have $\mathcal{F}_s = \sigma \cdot \sqrt{\frac{2\ln(1/\eta)}{n_{\text{ref}}+1}}$, where $\eta$ is the input parameter for Algorithm 2 to control the success probability, $n_{\text{ref}}$ is the number of samples that have been taken, $\sigma$ is provided as input to Algorithm 2 as prior knowledge (serving as an upper bound on each $\sigma_s$). Then, we have

$$2e^{-\frac{|\mathcal{B}| \cdot \mathcal{F}_s^2}{2\sigma_s^2}} = 2e^{-\frac{|\mathcal{B}| \cdot \sigma^2 \cdot 2\ln(1/\eta)}{2\sigma_s^2 \cdot (n_{\text{ref}}+1)}} \le 2e^{-\ln(1/\eta)},$$

where the last step follows from $n_{\text{ref}} + 1 = |\mathcal{B}|$ and $\sigma^2 \ge \sigma_s^2$ according to Assumption 1.

Let $\eta = \frac{1}{2n^4}$. We have $P_r(|\mu_s - \hat{\mu}_s| > \mathcal{F}_s) \le 1/n^4$. Then, $\mu_s \in [\hat{\mu}_s - \mathcal{F}_s, \hat{\mu}_s + \mathcal{F}_s]$ holds for each $s \in \mathcal{S}_{\text{swap}}$ with probability at least $1 - \frac{1}{n^3}$ by taking a union bound over the success probability of all the swap pairs in $\mathcal{S}_{\text{swap}}$. Since there are at most $|P|$ sampling steps, we can get that with probability at least $1 - \frac{1}{n^2}$, the intervals maintained can always reflect the true mean of the clustering cost reduction, i.e., $\mu_s \in [\hat{\mu}_s - \mathcal{F}_s, \hat{\mu}_s + \mathcal{F}_s]$ holds for each $s \in \mathcal{S}_{\text{swap}}$ during the whole bandit process.

Let $s^* = \arg\min_{s \in \mathcal{S}_{\text{swap}}} G_s(P)$ be the optimal target swap pair. Since $\hat{\mu}_{s^*} - \mathcal{F}_{s^*} \le \mu_{s^*}$, we have $\hat{\mu}_{s^*} - \mathcal{F}_{s^*} \le \mu_{s'} \le \hat{\mu}_{s'} + \mathcal{F}_{s'}$ holds for any $s' \in \mathcal{S}_{\text{swap}}$ such that $s' \ne s^*$. This indicates that the optimal swap pair with maximum clustering cost reduction will never be removed.

According to the grouping strategy, there are two cases that may happen (see Lemma B.3 for details): (1) either $\mathcal{S}_1$ or $\mathcal{S}_2$ contains a swap pair $(u, v)$ such that $\delta^2(P, C\backslash\{v\} \cup \{u\}) \le (1 - \frac{1}{100k})\delta^2(P, C)$; (2) $\mathcal{S}_1$ and $\mathcal{S}_2$ do not contain any swap pair $(u, v)$ such that $\delta^2(P, C\backslash\{v\}\cup\{u\}) \le (1-\frac{1}{100k})\delta^2(P, C)$. If case (2) happens, since each swap group contains a virtual swap pair with zero clustering cost change, the optimal swap pair will not be removed. Hence, if the set of swap pair returned by Algorithm 2 has size 1, the swap pair will not induce clustering cost increase. Then, we consider that case (1) happens. For a swap pair $s \in \mathcal{S}_{\text{swap}}$, let $\zeta_s = \mu_s - \mu_s^*$. For $n_{\text{ref}} = \frac{200\sigma^2}{\zeta_s^2}\ln n$, we have

$$\begin{aligned}
2(\mathcal{F}_s + \mathcal{F}_{s^*}) &= 4\sigma\sqrt{2\ln(n^4)/(n_{\text{ref}}+1)} \\
&\le \zeta_s \\
&= \mu_s - \mu_{s^*}.
\end{aligned}$$

Then, we can get that

$$\begin{aligned}
\hat{\mu}_s - \mathcal{F}_s &\ge \mu_s - 2\mathcal{F}_s \\
&= \mu_{s^*} + \zeta_s - 2\mathcal{F}_s \\
&\ge \mu_{s^*} + 2\mathcal{F}_s \\
&> \hat{\mu}_{s^*} + \mathcal{F}_s.
\end{aligned}$$

This implies that after randomly and independently sampling $\frac{200\sigma^2}{\zeta_s^2}\ln n$ points from $\mathcal{S}_{\text{ref}}$, the swap pair $s \in \mathcal{S}_{\text{swap}}$ can be removed from $\mathcal{S}_{\text{swap}}$ because its confidence interval is entirely outside the confidence interval of $s^*$.

Furthermore, in case (1), the given swap pair set $\mathcal{S}_{\text{swap}}$ contains a swap pair $s^* = (u, v)$ satisfying $\delta^2(P, C \cup \{u\}\backslash\{v\}) \le (1 - \frac{1}{100k})\delta^2(P, C)$ and a virtual swap pair $s = (u, u)$ with zero clustering cost change. Then, it holds that $s^* = (u, v)$ and $\mu_{s^*} \le -\frac{\delta^2(P,C)}{|\mathcal{S}_{\text{ref}}|100k}$. Since the virtual swap pair can only induce zero clustering cost change, we have $\zeta_{s^*} \ge \frac{\delta^2(P,C)}{|\mathcal{S}_{\text{ref}}|100k}$. Then, observe that $\sigma^2 \le O((\delta^2(P,C)/|\mathcal{S}_{\text{ref}}|)^2)$ according to Assumption 1, we can get that

$$n_{\text{ref}} = \frac{200\sigma^2 \cdot \ln n}{\zeta_s^2} \le O(\log n) \cdot O\left(\left(\frac{\delta^2(P,C)}{|\mathcal{S}_{\text{ref}}|}\right)^2\right) \cdot \frac{(100k|\mathcal{S}_{\text{ref}}|)^2}{(\delta^2(P,C))^2} = O(k^2 \log n),$$

which implies that by randomly and independently sampling $\tilde{O}(k^2)$ points from reference point set $\mathcal{S}_{\mathrm{ref}}$, the best swap pair in $\mathcal{S}$ can be returned by Algorithm 2. $\qquad \square$

During the swap pair construction process, the BanditLS algorithm (Algorithm 1) constructs swap pairs by finding the center $s_x \in C$ closest to $x$ and a randomly selected center $q' \in C \setminus \{s_x\}$ (steps 8-9 of Algorithm 1), where $\mathcal{A} = \{(x, s_x), (x, q')\}$ serves as the candidate swap pairs. The following lemma shows that if $x$ is sampled with probability $\delta^2(x, C)/\delta^2(P, C)$, with constant probability, there exists at least one swap pair that can induce certain fraction of clustering cost reduction.

**Lemma B.3.** (Fan et al. [19]) *Let $x$ be a data point sampled with probability $\delta^2(x, C)/\delta^2(P, C)$ from $P$. If $\delta^2(P, C) > 2000OPT$, then either with constant probability, there exists at least one swap pair $(u, v) \in \mathcal{A}$ such that $\delta^2(P, C \cup \{u\} \setminus \{v\}) \leq (1 - \frac{1}{100k})\delta^2(P, C)$, or with probability $\Omega(\lambda/k)$, there exists at least one swap pair $(u, v) \in \mathcal{A}$ such that $\delta^2(P, C \cup \{u\} \setminus \{v\}) \leq (1 - \frac{1}{100\lambda})\delta^2(P, C)$, where $\lambda$ denotes the number of lonely centers $(1 \leq \lambda \leq k - 1)$.*

According to Lemma 5, for the swap groups $\mathcal{S}_1$ and $\mathcal{S}_2$ provided as input to Algorithm 2, the algorithm can return a set $o'_1$ for $\mathcal{S}_1$ and a set $o'_2$ for $\mathcal{S}_2$. Algorithm 2 guarantees that if $|o'_i| = 1$, then $o'_i$ contains the best swap pair of $\mathcal{S}_i$ for each $i \in \{1, 2\}$. By taking a union bound success probability over the entire local search process, Algorithm 2 can always return the best swap pair with constant probability for each input swap pair group $\mathcal{S}_i$. According to Lemma B.3, with certain probability, there exists one swap pair that can induce at least a $(1 - 1/100k)$ fraction of reduction on clustering cost in each local search step. Then, with high probability, Algorithm 2 can return a set $o'_i$ of swap pairs with $|o'_i| = 1$. Since the final swap pair is randomly picked from $o'_i$ with $|o'_i| = 1$, we can get that with certain probability, the clustering cost can be reduced for each local search step.

**Lemma B.4.** *After $O(\sqrt{n}k \log(n\Delta))$ rounds of local search steps, Algorithm 1 can return a constant approximate solution in expectation.*

*Proof.* According to Lemma 5 and Lemma B.3, if sampling with distribution $\delta^2(x, C)/\delta^2(P, C)$ (where $x \in P$), then there are two cases that may happen: (1) with constant probability, the clustering cost can be reduced by a factor of at least $(1 - \frac{1}{100k})$ for a single local search step; (2) with probability $\Omega(\lambda/k)$, the clustering cost can be reduced by a factor of $(1 - \frac{1}{100\lambda})$. Based on Lemma A.6, each data point can be sampled with probability at least $\frac{\delta^2(x,C)}{\sqrt{n} \cdot \delta^2(P,C)}$ (where $x \in P$) using Adaptive-Sampling strategy. Hence, either the clustering cost can be reduced by a factor of at least $(1 - \frac{1}{100k})$ with probability $\Omega(\frac{1}{\sqrt{n}})$, or the clustering cost can be reduced by a factor of at least $(1 - \frac{1}{100\lambda})$ with probability $\Omega(\frac{\lambda}{k\sqrt{n}})$. Let $R = 2\alpha k\sqrt{n} \log(n\Delta)$, where $\alpha$ is a sufficiently large constant. We begin by examining case (1) of the local search process. Following the formulation in [25], we define a random process $X$ that starts from an initial clustering cost $\delta^2(P, C')$, where $C'$ is an $O(n\Delta)$-approximation. During $R/2$ iterations, this process reduces $\delta^2(P, C')$ by a multiplicative factor of $(1 - \frac{1}{100k})$ with probability $\eta = \frac{1}{\beta\sqrt{n}}$, while adding an additive term of $2000OPT$ after $R/2$ iterations for some constant $\beta$. From this, we can derive that

$$E[X] = 2000OPT + \delta^2(P, C') \sum_{i=1}^{R/2} \binom{R/2}{i} (\eta)^i (1 - \eta)^{R/2 - i} \left(1 - \frac{1}{100k}\right)^i$$

$$= \delta^2(P, C') \left(1 - \frac{\eta}{100k}\right)^{R/2} + 2000OPT$$

$$\leq \frac{\delta^2(P, C')}{n\Delta} + 2000OPT.$$

Next, we consider case (2) and define a new random process $X'$. Over $R/2$ iterations of sampling and swaps, the process decreases $\delta^2(P, C')$ by a multiplicative factor of $(1 - \frac{1}{100\lambda})$ with probability $\eta' = \frac{\lambda}{\beta\sqrt{n}k}$, and increases the final cost $\delta^2(P, C')$ by an additive term of $2000OPT$ for some constant $\beta$. In this case, we can similarly derive that

$$E[X'] = 2000OPT + \delta^2(P, C') \sum_{i=1}^{R/2} \binom{R/2}{i} (\eta')^i (1 - \eta')^{R/2-i} \left(1 - \frac{1}{100\lambda}\right)^i$$

$$= \delta^2(P, C') \left(1 - \frac{\eta'}{100\lambda}\right)^{R/2} + 2000OPT$$

$$\leq \frac{\delta^2(P, C')}{n\Delta} + 2000OPT,$$

For $R = 2\alpha k \sqrt{n} \log(n\Delta)$ iterations, there are at least $\alpha k \sqrt{n} \log(n\Delta)$ iterations such that either case (1) or case (2) happens. It is easy to see that the random process $X$ or $X'$ can dominate the local search process and hence $E[\delta^2(P, C)] < E[X']$ and $E[\delta^2(P, C)] < E[X]$ hold, where $C$ is the set of the final clustering centers returned by the local search algorithm. Then, we can get that

$$E[\delta^2(P, C)] \leq E[X](\text{or } E[\delta^2(P, C)] \leq E[X'])$$

$$\leq \sum_{C'} E[\delta^2(P, C')]P_r(C')$$

$$\leq \sum_{C'} P_r(C')(\frac{\delta^2(P, C')}{n\Delta} + 2000OPT)$$

$$\leq \frac{E[\delta^2(P, C')]}{n\Delta} + 2000OPT.$$

Since the set $C'$ is an $O(n\Delta)$-approximate initial solution, we have $E[\delta^2(P, C)] \leq O(1)OPT$ by setting $\alpha$ as a large enough constant. $\square$

Putting all these together, Theorem 1 can be proved.

**Theorem 1.** *With sub-Gaussian prior on cost changes of swap pairs, there exists an algorithm for k-means that can achieve constant approximation within expected running time $\tilde{O}(nd + \sqrt{n}k^4)$.*

*Proof.* The approximation guarantees on clustering quality follows from Lemma B.4. Then, we analyze the running time. The tree initialization (steps 1-3 of Algorithm 1) takes time $O(nd \log(n\Delta))$. Note that $\Delta$ and the dimension $d$ can be compressed to $\Delta = \text{poly}(n, d)$ and $O(\log n)$, respectively, using the methods proposed in [18, 28]. Hence, the running time for initialization becomes $\tilde{O}(nd)$. During the local search steps, there are at most $\tilde{O}(\sqrt{n}k)$ iterations. The Adaptive-Sampling takes time $\tilde{O}(k)$ for each local search step according to Lemma A.6. For the bandit-based center swap process, finding the closest center in $C$ for the sampled data point $x$ takes time $\tilde{O}(k)$ in step 6 of Algorithm 1. For each swap pair constructed, the bandit-based clustering cost estimation takes $\tilde{O}(k^2)$ samples. The algorithm needs to calculate the true distances between the samples to their nearest centers, which takes $\tilde{O}(k^3)$ time. Hence, the bandit process takes an overall running time of $\tilde{O}(k^3)$ (steps 9-13 of Algorithm 1). Finally, updating the tree structure through TREE-OPEN and TREE-CLOSE operations take time $\tilde{O}(1)$ for each local search step. Since there are at most $\tilde{O}(\sqrt{n}k)$ local search steps, the total running time is $\tilde{O}(nd + \sqrt{n}k^4)$. $\square$

## C  A More Practical Algorithm with Metropolis Hastings

The Metropolis-Hastings strategy is widely used in statistics for sampling from complex probability distributions, where several initialization schemes with sub-linear (or linear) running time were proposed [3, 4]. In the following, we show that under mild assumptions, the Metropolis-Hastings strategy can be used to approximate the $D^2$-Sampling distribution within $\tilde{O}(n/k^2)$ rounds of sample transitions during the local search process. The following lemma is a folklore for Metropolis-Hastings.

**Lemma C.1.** (Chib and Greenberg [10]) *Let $\pi$ denote the target distribution defined on $P$ that one wish to approximate using Metropolis-Hastings process. Given a parameter $\beta > 0$, if the*

proposal distribution $q$ satisfies that $q(x) \geq \beta\pi(x)$ for all $x \in P$, then after $T = O(1/\beta)$ steps, the distribution of the Metropolis-Hastings chain dominates $\pi/2$.

The following lemma shows that under a tailed behavior of distributions on data points, the ratio between the maximum distances to the optimal clustering center and the average clustering cost can be bounded by a function that is sublinear in the data size $n$.

**Lemma C.2.** (Bachem et al. [3]) *Let F be a probability distribution over $\mathbb{R}^d$ with finite variance that has at most exponential tails, i.e., $\exists c, t$ such that $P_r[\delta^2(x, \mu) > a] \leq ce^{-at}$, where $c, t$ are constants. Let $\mathcal{X}$ be a set of $n$ points independently sampled from F. Then, with high probability, for sufficiently large n, $\alpha = \frac{\max_{x \in \mathcal{X}} \delta^2(x, \mu(\mathcal{X}))}{\frac{1}{n}\sum_{x' \in \mathcal{X}} \delta^2(x', \mu(\mathcal{X}))^2} = O(\log^2 n)$.*

Based on Lemma C.1, Lemma C.2 and Assumption 2, Lemma 6 can be proved.

**Assumption 2.** *Assume that the given dataset $P$ is average where each optimal cluster has size $\Omega(k^2)$. For each optimal cluster $P_h^*$, let $c_h^*$ be the optimal center for $P_h^*$. We assume that each $P_h^*$ follows a distribution F over $\mathbb{R}^d$ with exponential tails, i.e., $\exists c, f$ such that $P_r[\delta^2(x, \mu) > a] \leq ce^{-fa}$ holds for $x \in P_h^*$, where $c, f$ are constants and $\mu$ is the mean of the distribution.*

**Lemma 6.** *After $\tilde{O}(n/k^2)$ Metropolis-Hastings sampling steps (steps 2-5 of Algorithm 3), if the given clustering instance satisfies the properties in Assumption 2, we can sample a data point $x \in P$ with probability at least $0.5\delta^2(x, C)/\delta^2(P, C)$.*

*Proof.* For any data point $p \in P$ and a given set $C$ of clustering centers opened, let $s_p = \arg\min_{c' \in C} \delta(p, c')$ be its closest center in $C$. We first show that if the given clustering instance satisfies the property of average sizes (i.e., each optimal cluster has size $\Omega(k^2)$), the distance between $p$ and $s_p$ can be bounded. Without loss of generality, we can assume that $p$ lies in some optimal cluster $P_j^*$ where $c_j^*$ is its optimal clustering center. Let $OPT_j$ be the clustering cost of $P_j^*$ with respect to $c_j^*$, i.e., $OPT_j = \delta^2(P_j^*, c_j^*)$. Denote $s_{c_j^*}$ as the nearest data point in $C$ to $c_j^*$. Then, we can get that $\delta^2(c_j^*, s_{c_j^*}) \leq \min_{q \in C_j^*} \delta^2(c_j^*, s_q) \leq \sum_{q \in C_j^*} \frac{2\delta^2(c_j^*, q) + 2\delta^2(q, s_q)}{|C_j^*|} \leq \frac{2OPT_j + 2\delta^2(C_j^*, C)}{|C_j^*|} \leq O\left(\frac{OPT_j + \delta^2(C_j^*, C)}{k^2}\right)$, where the second step follows from the triangle inequality, and the last step follows from the assumption that $|C_j^*| = \Omega(k^2)$ (Assumption 2) holds for each $j \in [k]$. Denote $d_j^{\max} = \max_{x \in P_j^*} \delta^2(x, c_j^*)$. Then, we have

$$\begin{aligned}
\delta^2(p, s_p) &\leq \delta^2(p, s_{c_j^*}) \\
&\leq 2\delta^2(p, c_j^*) + 2\delta^2(c_j^*, s_{c_j^*}) \\
&\leq 2d_j^{\max} + O\left(\delta^2(C_j^*, C)/k^2 + OPT_j/k^2\right).
\end{aligned}$$

We now prove that $p(x) \geq \pi(x)/2$, where $p(x)$ is the probability of sampling $x$ using $\tilde{O}(n/k^2)$ steps of Metropolis Hastings (with a uniform proposal distribution $1/n$) and $\pi(x) = \delta^2(x, C)/\delta^2(P, C)$ is the target $D^2$-Sampling distribution. Let $\beta = \frac{k^2}{\tau n \log^2 n}$ for some large enough constant $\tau$. For each optimal cluster $P_j^*$, according to Assumption 2, we assume that $P_j^*$ follows a distribution $F$ with exponential tails. Then, according to Lemma C.2, we have $\alpha_j = \frac{|C_j^*| \cdot d_j^{\max}}{\delta^2(P_j^*, c_j^*)} = O(\log^2 n)$. This implies that

$$\delta^2(p, s_p) = O\left(\frac{\delta^2(C_j^*, C) + OPT_j}{k^2}\right) \cdot \log^2 n.$$

For each $x \in P$, let $\phi(x)$ be the index of the optimal cluster that $x$ belongs to, i.e., $x \in P^*_{\phi(x)}$. Then, it holds that

$$
\begin{aligned}
\beta\pi(x) &\leq \frac{k^2}{\tau n \log^2 n} \cdot \frac{\delta^2(x, C)}{\delta^2(P, C)} \\
&\leq \frac{k^2}{\tau n \log^2 n} \cdot \frac{\delta^2(x, s_x)}{\delta^2(P, C)} \\
&\leq \frac{k^2}{\tau n \log^2 n} \cdot \frac{O\left(\frac{\delta^2(C^*_{\phi(x)}, C) + OPT_{\phi(x)}}{k^2}\right) \cdot \log^2 n}{\delta^2(P, C)} \\
&\leq \frac{1}{n},
\end{aligned}
$$

where the last step follows from the fact that $O\left(\frac{\delta^2(C^*_j, C) + OPT_j}{k^2}\right) \leq O\left(\delta^2(P, C)/k^2\right)$ holds for each $C^*_j$, since $C^*_j \subseteq P$ and $\delta^2(P, C) \geq OPT$. The proof can be concluded using Lemma C.1. $\quad\square$

Putting all these together, Theorem 2 can be proved.

**Theorem 2.** *With sub-Gaussian prior on cost changes of swap pairs and mild assumptions on optimal clusters, there is a constant approximation algorithm for $k$-means with expected runtime $\tilde{O}(nd + k^4)$.*

*Proof.* We first prove the approximation guarantees on clustering quality. According to Lemma 6, with $\tilde{O}(n/k^2)$ Metropolis-Hastings steps, we can sample a data point $x \in P$ with probability at least $0.5\delta^2(x, C)/\delta^2(P, C)$ for each data point $x \in P$. Then, similar to the analysis for Lemma B.4, after $\tilde{O}(k)$ local search steps, we can obtain a constant approximation in expectation.

As for the running time, each approximate $D^2$-Sampling step based on Metropolis-Hastings strategy takes $\tilde{O}(n/k^2)$ transitions, where each transition needs to calculate the distance between the data point sampled to its nearest center in $C$. Hence, the overall running time for the approximate sampling process in a single local search step is $\tilde{O}(nd/k)$. According to Theorem 1, each bandit step takes time $\tilde{O}(k^3 d)$. Finally, since $\tilde{O}(k)$ rounds of local search steps are required to return a constant approximate solution, the total running time is $\tilde{O}(nd + k^4)$ using dimension reduction techniques. $\quad\square$

# D  Extension to the $k$-median Objective

In this section, we show how to extend our proposed adaptive sampling method to the $k$-median objective without relying on any data distribution assumptions. The formal algorithm is presented in Algorithm 11, where the high level idea behind is to replace the bandit process with a weighted sampling strategy (Algorithm 12) to remove the assumptions on sub-Gaussian prior for swap pair.

Similar to the idea for the $k$-means objective, the algorithm begins by embedding the given $k$-median instance into a tree structure $\mathcal{T}'$, where the pairwise distances can be well approximated on the tree metric (steps 1-3 of Algorithm 11). Then, during each local search step, an adaptive sampling process is used to approximate the $D$-Sampling (which refers to the sampling distribution of $\delta(p, C)/\delta(P, C)$ for a given set $C$ of centers opened) process. Instead of modeling the best swap pair identification task as a bandit problem, the algorithm adapts a weighted sampling strategy to estimate the clustering cost induced by each swap pair (Algorithm 12). For each swap pair to be estimated, the Weighted-Sampling algorithm first performs the swaps on the tree structure $\mathcal{T}'$ based on TREE-OPEN and TREE-CLOSE operations (steps 1-3 of Algorithm 12). Then, it independently takes $\tilde{O}(\sqrt{n}/\epsilon)$ samples using the adaptive sampling method (step 8 of Algorithm 12). The intuitive idea is to leverage the properties by adaptive sampling method such that each data point $p \in P$ can be sampled with probability linearly related to the distance of $p$ to its nearest centers opened. This would give a good approximation to the inverse value of the clustering cost for $k$-median objective using weighted summation of the samples drawn. With this technique, the time complexity for clustering cost estimation can be reduced from $O(nk)$ to be sub-linear in the data size $n$.

---

**Algorithm 11** LS-KMedian$(P, k, d, C)$

---

**Input:** A $k$-median clustering instance $(P, k, d)$, a set $C \subset \mathbb{R}^d$ of random clustering centers.
**Output:** A set $C \subset \mathbb{R}^d$ of clustering centers.
1: $\mathcal{T} = $ Tree-Construction$(P, k, d)$. $\triangleright$ Tree-Construction is detailed in Algorithm 4 in Appendix A
2: $\mathcal{T}' = $ Tree-Conversion$(\mathcal{T})$. $\triangleright$ Tree-Conversion is detailed in Algorithm 6 in Appendix A
3: Call TREE-INIT$(\mathcal{T}')$ to initialize the dynamic data structure. $\triangleright$ Algorithm 7 in Appendix A
4: **for** $c \in C$ **do**
5: $\quad$ Call the TREE-OPEN$(\mathcal{T}', c)$ algorithm to mark the leaf node in $\mathcal{T}'$ associated with $c$ as active
$\quad$ and update the tree structure $\mathcal{T}'$. $\triangleright$ TREE-OPEN is detailed in Algorithm 8 in Appendix A
6: **for** $i = 1$ to $\tilde{O}(k)$ **do**
7: $\quad$ $x = $ Adaptive-Sampling$(\mathcal{T}', C)$ $\triangleright$ Adaptive-Sampling is in Algorithm 10 in Appendix A
8: $\quad$ Set $s_x = \arg\min_{c \in C} \delta(x, c)$ and randomly sample a center $q' \in C \backslash \{s_x\}$.
9: $\quad$ $o = \{(x, s_x), (s_x, s_x), (x, q'), (q', q')\}$.
10: $\quad$ **for** $o_i \in o$ **do**
11: $\quad\quad$ $EST(o_i) = $Weighted-Sampling$(P, k, d, \mathcal{T}', o_i, 1/400k)$.
12: $\quad$ $g_1 = \{(x, s_x), (s_x, s_x)\}, g_2 = \{(x, q'), (q', q')\}$.
13: $\quad$ $g_{m_1} = \arg\min_{g \in g_1} EST(g), g_{m_2} = \arg\min_{g \in g_2} EST(g)$.
14: $\quad$ **for** $g \in g_1$ **do**
15: $\quad\quad$ **if** $EST(g) \cdot \frac{400k-1}{400k+1} > EST(g_{m_1})$ **then**
16: $\quad\quad\quad$ remove $g$ from $g_1$.
17: $\quad$ **for** $g \in g_2$ **do**
18: $\quad\quad$ **if** $EST(g) \cdot \frac{400k-1}{400k+1} > EST(g_{m_2})$ **then**
19: $\quad\quad\quad$ remove $g$ from $g_2$.
20: $\quad$ Let $\mathcal{O} = \{g_i : |g_i| = 1, i \in \{1, 2\}\}$, and randomly choose a swap pair $o'$ from $\mathcal{O}$.
21: $\quad$ $C = C \backslash \{v\} \cup \{u\}$, where $(u, v) \in o'$.
22: $\quad$ Call the TREE-CLOSE$(\mathcal{T}', v)$ algorithm to deactivate the leaf node in $\mathcal{T}'$ associated with $v$
$\quad$ and update the tree structure $\mathcal{T}'$. $\triangleright$ TREE-CLOSE is detailed in Algorithm 9 in Appendix A
23: $\quad$ Call the TREE-OPEN$(\mathcal{T}', u)$ algorithm to mark the leaf node in $\mathcal{T}'$ associated with $u$ as active
$\quad$ and update the tree structure $\mathcal{T}'$.
24: **return** $C$.

---

In the following, we give the detailed analysis for the proposed algorithm. The following lemma can be directly obtained via Lemma A.6, which is an extension from the $k$-means objective to the $k$-median objective.

**Lemma D.1.** *The Adaptive-Sampling takes expected time $\tilde{O}(k)$ to sample a data point $x \in P$ with probability $\frac{\delta(x,C)}{\delta(P,C)}$.*

According to Lemma D.1, each data point is sampled with probability proportional to the distance to its nearest centers opened. In the following, we will show that by modeling the clustering cost estimation as a sum estimation problem, the clustering cost can be well approximated within a factor of $(1 \pm \epsilon)$ using $\tilde{O}(\sqrt{n}/\epsilon)$ samples.

For the clustering problem, given a swap pair $(u, v)$, the clustering cost of $\delta(P, C \backslash \{v\} \cup \{u\})$ can be decomposed into $\delta(P, C \backslash \{v\} \cup \{u\}) = \sum_{p \in P} \delta(p, C \backslash \{v\} \cup \{u\})$, which can further be modeled as a sum estimation task. Our main objective is to estimate the sum $W = \sum_{p \in P} \delta(p, C \backslash \{v\} \cup \{u\})$ using as small number of samples as possible. The following lemma shows that if proportional sampling is allowed (i.e., the $D$-Sampling), one can estimate the sum using $\tilde{O}(\sqrt{n}/\epsilon)$ samples.

**Lemma D.2.** (Beretta and Tĕtek [8]) *Given a set $U$ where each item $a \in U$ has a weight $w(a)$, let $W = \sum_{a \in U} w(a)$ be the summation of the weights in $U$. Let $S = \{a_1, ..., a_m\}$ be a set of elements sampled $m_n = O(\sqrt{|U|}/\epsilon)$ times from $U$ such that each $a_i \in S$ is sampled with probabilities proportional to their weights. For each $s \in S$, define $c_s$ to be the number of times $s$ is sampled. Then,*
$$W' = \binom{m_n}{2} \cdot \left( \sum_{s \in \mathcal{S}} \frac{\binom{c_s}{2}}{w(s)} \right)^{-1} \text{ gives an estimation for } W \text{ that induces multiplicative error of } (1 \pm \epsilon)$$
*with probability at least $2/3$.*

---

**Algorithm 12** Weighted-Sampling$(P, k, d, \mathcal{T}', s, \epsilon)$

---

**Input:** A $k$-median clustering instance $(P, k, d)$, a tree structure $\mathcal{T}'$, a swap pair $s$, and a parameter $0 < \epsilon < 1$.

**Output:** The estimation of the clustering cost induced by the swap pair $s$.

1: Let $s = (u, v)$ be the swap pair to be estimated.
2: Call the TREE-CLOSE$(\mathcal{T}', v)$ algorithm to deactivate the leaf node associated with $v$ and update the tree structure $\mathcal{T}'$.           ▷ TREE-CLOSE is detailed in Algorithm 9 in Appendix A
3: Call the TREE-OPEN$(\mathcal{T}', u)$ algorithm to mark the leaf node associated with $u$ as active and update the tree structure $\mathcal{T}'$.           ▷ TREE-OPEN is detailed in Algorithm 8 in Appendix A
4: Initialize $\mathcal{W} = \emptyset$, $C' = C \backslash \{v\} \cup \{u\}$.
5: **for** $m = 1$ to $O(\log n)$ **do**
6:    Initialize an empty set $\mathcal{M} = \emptyset$, and set $m_n = 0$.
7:    **for** $i = 1$ to $O(\sqrt{n}/\epsilon)$ **do**
8:       $s =$Adaptive-Sampling$(\mathcal{T}', C')$, $m_n = m_n + 1$.
9:       **if** $s \in \mathcal{M}$ **then**
10:          $c_s = c_s + 1$.
11:       **else**
12:          $c_s = 1$, $\mathcal{M} = \mathcal{M} \cup \{s\}$.
13:    $W' = \binom{m_n}{2} \cdot \left( \sum_{s \in \mathcal{M}} \frac{\binom{c_s}{2}}{\delta(s, C')} \right)^{-1}$, $\mathcal{W} = \mathcal{W} \cup \{W'\}$.
14: Sort the values in $\mathcal{W}$ with increasing order and let $W$ be the median value.
15: **return** $W$.

---

Lemma D.2 indicates that through $D$-Sampling (sampling with probabilities proportional to the weights), the weighted sum can be closely estimated using $O(\sqrt{n}/\epsilon)$ samples. Since Lemma D.1 guarantees that the Adaptive-Sampling process can sample each data point $a \in P$ with probability $\delta(a, C \backslash \{v\} \cup \{u\})/\delta(P, C \backslash \{v\} \cup \{u\})$, the following lemma can be directly obtained.

**Lemma D.3.** *Given a swap pair $(u, v)$ and a set $C$ of centers opened as the input for Algorithm 12, let $C' = C \backslash \{v\} \cup \{v\}$ be the set of centers obtained after performing the swap. For the $m$-th iteration of Algorithm 12 (steps 6-13), an estimation $W'$ can be obtained in step 13 of Algorithm 12 such that $W' \in [(1 - \epsilon)\delta(P, C'), (1 + \epsilon)\delta(P, C')]$ with probability at least $2/3$.*

However, since there are $\tilde{O}(k)$ local search steps, we need to boost the success probability to guarantee a union bound of success probability over all the local search steps. According to the analysis in weighted sampling process [8], $1/W'$ is an unbiased estimation for $1/W$ and the success probability is established by using Chebyshev inequality. Hence, based on a standard Median of Means trick [26] (step 5 and step 14 of Algorithm 12), the success probability for finding an accurate estimation can be boosted to $1 - 1/n^2$ by constructing $O(\log n)$ groups and taking the median value. Hence, we have the following result.

**Lemma D.4.** *Given a swap pair $(u, v)$ and a set $C$ of centers opened as the input for Algorithm 12, let $C' = C \backslash \{v\} \cup \{v\}$ be the set of centers obtained after performing the swap. An estimation $W$ can be returned by Algorithm 12 such that $W \in [(1 - \epsilon)\delta(P, C'), (1 + \epsilon)\delta(P, C')]$ with high probability of at least $1 - 1/n^2$.*

Next, we will show that by carefully adjusting the parameter $\epsilon$ and introducing a judging criteria, the clustering cost can be reduced by at least a factor of $(1 - 1/100k)$ with certain probability for each local search step. The basic idea behind is to add a virtual swap pair $(x, x)$ for some $x \in C$ to serve as baseline, where swap pairs that fail to induce clustering cost reductions can be filtered by the virtual swap pair. The following lemma shows that using the virtual swap pair as a baseline, the optimal swap pair among the swap pair set that induces the maximum reduction on clustering cost will never be removed (steps 12-19 of Algorithm 11) .

**Lemma D.5.** *Let $g_i$ $(i \in \{1, 2\})$ be the swap groups constructed in step 12 of Algorithm 11. For each swap group $g_i$, let $t_i = \arg\min_{(u,v) \in g_i} \delta^2(P, C \backslash \{v\} \cup \{u\})$ be the swap pair in $g_i$ with minimum clustering cost. Then, $t_i$ will never be removed during the subsequent steps 14-19 of Algorithm 11*

*Proof.* Given a swap pair $a \in o$ (where $o$ is the set of swap pairs constructed in step 9 of Algorithm 11), we use $\text{cost}(a) = \delta(P, C \backslash \{v\} \cup \{u\})$ to denote the true clustering cost induced by the swap pair $a$, where $a = (u, v)$. According to Lemma D.4, by taking a union bound over the success probability, we can assume without loss of generality that the clustering cost estimation $EST(o_i)$ for each swap pair $o_i \in o$ should satisfy $EST(o_i) \in [(1 - 1/400k)\text{cost}(o_i), (1 + 1/400k)\text{cost}(o_i)]$ by setting $\epsilon = 1/400k$. Consider an arbitrary swap group $g_i$ for some $i \in \{1, 2\}$. Let $g_{m_i}$ be the swap pair in $g_i$ with minimum clustering cost estimations (step 13 of Algorithm 11). According to the judging criteria (step 15 and step 18 of Algorithm 11), a swap pair $g \in g_i$ is removed only if its cost estimation $EST(g)$ satisfies $EST(g) \cdot \frac{400k-1}{400k+1} > EST(g_{m_i})$. For the best swap pair $t_i$, it hods that $EST(t_i) \cdot \frac{400k-1}{400k+1} \leq \text{cost}(t_i) \cdot (1 + 1/400k) \cdot \frac{400k-1}{400k+1} \leq \frac{400k-1}{400k}\text{cost}(g_{m_i}) \leq EST(g_{m_i})$, where the first and third inequalities follow from Lemma D.4, the second inequality follows from that $t_i$ is the swap pair with minimum clustering cost. This indicates that the optimal swap pair will never be removed from the set $g_i$ of candidate swap pairs. $\square$

Lemma D.5 implies that the optimal swap pair in each swap group $g_i$ ($i \in \{1, 2\}$) will never be removed. Then, we will demonstrate that the clustering cost can be reduced significantly with certain probability for each local search step. According to Lemma B.3, with certain probability, for a single local search step, there exists at least one swap pair $(u, v) \in o$ such that the clustering cost can be reduced by a factor of at least $(1 - 1/100k)$. The following lemma shows that if such swap pair exists, the virtual swap pair and swap pairs with clustering cost larger than the virtual swap pair must be removed.

**Lemma D.6.** *If there exists at least one swap pair $(u, v) \in g_i$ such that $\delta(P, C \backslash \{v\} \cup \{u\}) \leq (1 - 1/100k)\delta(P, C)$, then swap pairs in $g_i$ with clustering cost larger than $\delta(P, C)$ will be removed.*

*Proof.* Given a swap pair $t_i = (u, v)$ such that $\delta(P, C \backslash \{v\} \cup \{u\}) \leq (1 - 1/100k)\delta(P, C)$, it holds that

$$
\begin{aligned}
EST(t_i) &\leq (1 + 1/400k)\text{cost}(t_i) \\
&\leq (1 + 1/400k)(1 - 1/100k)\delta(P, C) \\
&\leq (1 - 3/400k)\delta(P, C),
\end{aligned}
$$

where the first inequality follows from Lemma D.4, and the second inequality follows from the definition of $t_i$. For any swap pair $g \in g_i$ with clustering cost $\text{cost}(g) \geq \delta(P, C)$, it holds that

$$
\begin{aligned}
EST(g) \cdot \frac{400k-1}{400k+1} &\geq \text{cost}(g) \cdot (1 - 1/400k) \cdot \frac{400k-1}{400k+1} \\
&\geq \delta(P, C) \cdot \frac{(400k-1)^2}{400k(400k+1)} \\
&> \delta(P, C) \cdot (1 - 3/400k),
\end{aligned}
$$

where the second inequality follows from the definition for $g$. This indicates that $g$ will be removed from $g_i$ according to the judging criteria. $\square$

By combining Lemma D.5 and Lemma D.6, the clustering cost will not increase during each local search step. Then, based on Lemma B.3, in each local search step, with certain probability, there exists at least one swap pair $(u, v) \in o$ such that the clustering cost can be reduced by a factor of at least $(1 - 1/100k)$. By grouping such $(u, v)$ with the virtual swap pair, the virtual swap pair will be removed from $g_i$ for each $i \in \{1, 2\}$. Since the final swap pair is randomly picked from $g_i$ with $|g_i| = 1$, we can get that with certain probability, the clustering cost can be reduced by a factor of at least $(1 - \frac{1}{100k})$ for each local search step.

**Corollary D.4.** *For the LS-KMedian algorithm, either with constant probability, there exists at least one swap pair such that the clustering cost can be reduced by a factor of $(1 - 1/100k)$, or with probability $\Omega(\lambda/k)$, there exists at least one swap pair such that the clustering cost can be reduced by a factor of $(1 - 1/100\lambda)$ where $\lambda$ denotes the number of lonely centers $(1 \leq \lambda \leq k - 1)$.*

Putting all these together, Theorem 3 can be proved.

**Theorem 3.** *For the $k$-median problem, there exists an algorithm that can output a constant approximate solution in expected $\tilde{O}(nd + \sqrt{n}k^3)$ running time without any data assumptions.*

*Proof.* We first prove the approximation guarantees on clustering quality. According to Corollary D.4, with certain probability, the clustering cost of $P$ with respect to $C$ can be reduced by a certain fraction for each local search step. Then, similar to the analysis for Lemma B.4, after $\tilde{O}(k)$ local search steps, we can obtain a constant approximation in expectation.

As for the running time, since $\Delta$ can be compressed to $\Delta = \mathrm{poly}(n, d)$ using the method proposed in [18], the running time for initialization becomes $\tilde{O}(nd)$. Each weighted sampling step requires to take $\tilde{O}(\sqrt{n}/\epsilon)$ samples for $\epsilon = 1/400k$. During this process, the Adaptive-Sampling algorithm is used to take samples, where each sample can be picked within time $\tilde{O}(k)$. The weighted sum calculations (step 13 of Algorithm 12) involves distance calculation between the sampled point to the centers after swaps, which takes $\tilde{O}(k)$ time. Hence, the overall running time for weighted sampling during each local search step can be bounded by $\tilde{O}(\sqrt{n}k^2)$ since $d$ can be compressed to $O(\log n)$ with $\tilde{O}(nd)$ preprocessing time. Finally, since $\tilde{O}(k)$ rounds of local search steps are required to return a constant approximate solution, the total running time is $\tilde{O}(nd + \sqrt{n}k^3)$. □

# E    Complementary Experiments

In this section, we present complementary experiments on both large-scale and small datasets. We also evaluate the robustness of the BanditFastLS algorithm by testing it with varying parameters.

## E.1    Comparison results on other large-scale datasets

Table 3 presents the comparison results with other local search algorithms (with mini-batch $k$-means++ as baseline) on datasets SYN, USC_1990, and SUSY, with $k$ varying from 10 to 100. The results demonstrate that our algorithm still outperforms other local search methods in terms of running time, while achieving much better clustering quality than the mini-batch $k$-means++ seeding method.

Table 3: Comparison results on datasets SYN, USC_1990 and SUSY with varying $k$, where algorithms fail to return a feasible solution within 24 hours are not included

| Method | Dataset | $k$ | Cost | Time(s) | Dataset | $k$ | Cost | Time(s) | Dataset | $k$ | Cost | Time(s) |
|---|---|---|---|---|---|---|---|---|---|---|---|---|
| $k$-means++ | | | 3.3500E+07±1.0E+05 | 3.83 | | | 2.6026E+08±7.2E+06 | 2.30 | | | 6.0714E+05±2.2E+04 | 0.59 |
| MLS | | | 3.0388E+07±2.7E+05 | 172.23 | | | 2.3916E+08±2.6E+06 | 129.83 | | | 5.6302E+05±5.3E+02 | 27.64 |
| LSDS++ | SUSY | 10 | 3.0169E+07±1.9E+05 | 446.25 | USC_1990 | 10 | 2.4114E+08±3.0E+06 | 308.94 | SYN | 10 | 5.6706E+05±2.0E+03 | 56.19 |
| LS++ | | | 3.0320E+07±2.8E+05 | 849.06 | | | **2.3658E+08±2.9E+04** | 519.16 | | | **5.6226E+05±9.9E+02** | 42.86 |
| Ours | | | **3.0147E+07±2.3E+05** | **3.67** | | | 2.4089E+08±4.1E+06 | **1.12** | | | 6.7609E+05±1.0E+04 | **0.41** |
| $k$-means++ | | | 2.7035E+07±2.9E+05 | 7.49 | | | 1.9725E+08±1.7E+06 | 2.97 | | | 1.7295E+05±8.5E+03 | 0.93 |
| MLS | | | 2.5095E+07±5.1E+04 | 299.55 | | | 1.8890E+08±8.5E+05 | 220.42 | | | 1.7812E+05±1.4E+04 | 43.72 |
| LSDS++ | SUSY | 20 | 2.5039E+07±2.5E+05 | 616.36 | USC_1990 | 20 | 1.8491E+08±1.1E+06 | 372.19 | SYN | 20 | 1.6041E+05±1.7E+03 | 81.44 |
| LS++ | | | **2.4979E+07±1.6E+05** | 1308.89 | | | **1.8364E+08±1.2E+06** | 552.81 | | | **1.5655E+05±3.9E+02** | 59.69 |
| Ours | | | 2.5031E+07±2.2E+05 | **2.65** | | | 1.8621E+08±1.3E+06 | **1.24** | | | 1.6223E+05±4.9E+03 | **0.52** |
| $k$-means++ | | | 2.4013E+07±7.4E+04 | 8.06 | | | 1.6686E+08±5.3E+06 | 4.01 | | | 8.1212E+04±2.3E+03 | 2.06 |
| MLS | | | **2.2299E+07±3.7E+03** | 435.32 | | | 1.5849E+08±7.1E+05 | 294.56 | | | 8.0732E+04±3.0E+03 | 62.41 |
| LSDS++ | SUSY | 30 | 2.2384E+07±8.0E+04 | 719.61 | USC_1990 | 30 | 1.5466E+08±1.1E+06 | 450.36 | SYN | 30 | 7.5435E+04±4.3E+02 | 93.99 |
| LS++ | | | 2.2327E+07±2.7E+05 | 1410.85 | | | **1.5389E+08±1.3E+06** | 558.81 | | | **7.3313E+04±1.6E+02** | 118.18 |
| Ours | | | 2.2367E+07±1.5E+05 | **3.42** | | | 1.5797E+08±3.2E+06 | **1.69** | | | 7.7674E+04±1.8E+03 | **0.69** |
| $k$-means++ | | | 2.0866E+07±9.2E+05 | 9.67 | | | 1.3367E+08±1.3E+06 | 5.69 | | | 2.9880E+04±5.1E+02 | 3.46 |
| MLS | | | 1.9545E+07±1.7E+04 | 612.43 | | | 1.3209E+08±8.3E+05 | 493.50 | | | 4.5235E+04±8.4E+03 | 100.63 |
| LSDS++ | SUSY | 50 | 1.9512E+07±3.0E+04 | 956.10 | USC_1990 | 50 | 1.2525E+08±1.3E+06 | 575.61 | SYN | 50 | 2.8270E+04±1.1E+02 | 141.38 |
| LS++ | | | **1.9474E+07±8.8E+04** | 2088.05 | | | **1.2440E+08±5.8E+05** | 592.95 | | | **2.7670E+04±1.1E+02** | 274.49 |
| Ours | | | 1.9527E+07±2.2E+05 | **4.22** | | | 1.2875E+08±1.6E+06 | **2.10** | | | 3.0961E+04±2.1E+03 | **0.95** |
| $k$-means++ | | | 1.7419E+07±4.2E+03 | 21.54 | | | 1.0421E+08±8.0E+05 | 11.36 | | | 7.7096E+03±9.0E+01 | 7.92 |
| MLS | | | 1.6331E+07±2.5E+04 | 1251.13 | | | 1.0100E+08±1.3E+06 | 1151.48 | | | 1.4768E+04±1.1E+03 | 195.48 |
| LSDS++ | SUSY | 100 | 1.6433E+07±2.8E+05 | 1536.94 | USC_1990 | 100 | 9.9693E+07±3.4E+05 | 915.23 | SYN | 100 | 7.3685E+03±7.4E+01 | 271.29 |
| LS++ | | | 1.6403E+07±1.7E+04 | 2819.42 | | | **9.6182E+07±3.2E+05** | 921.09 | | | **7.1562E+03±3.1E+01** | 514.95 |
| Ours | | | **1.6321E+07±2.9E+04** | **6.54** | | | 9.9865E+07±1.5E+06 | **3.58** | | | 1.1788E+04±2.1E+03 | **1.79** |

Table 4 presents the comparison results on small datasets with sizes smaller than 50,000, where the number of clusters $k$ ranges from 3 to 10. The results indicate that all local search algorithms achieve fast running times, with the MLS and LS++ algorithms delivering better clustering quality.

### E.1.1    Experiments with Varying Parameters

In this subsection, we present experiments on the BanditFastLS algorithm with varying parameters. The input parameters for the BanditFastLS algorithm include the probability parameter $\eta$ and the number of Metropolis-Hastings steps used to approximate the $D^2$-Sampling distribution. As outlined in the main text, we fix the number of Metropolis-Hastings steps at 20 and set $\eta = 1/(2n^4)$ to meet the success probability guarantees specified in theoretical analysis. However, since controlling $\eta$ to

Table 4: Comparison results on small datasets

| Method | Dataset | k | Cost | Time(s) | k | Cost | Time(s) | k | Cost | Time(s) |
|---|---|---|---|---|---|---|---|---|---|---|
| k-means++ | | | 82.9406 ± 4.7260 | 0.0466 ± 0.0673 | | 56.5191 ± 9.1071 | 0.0119 ± 0.0124 | | 28.4889 ± 0.4447 | 0.0109 ± 0.0035 |
| MLS | | | 79.7067 ± 0.9330 | **0.0384 ± 0.0108** | | **46.8863 ± 0.0941** | 0.0561 ± 0.0020 | | 26.9977 ± 0.2052 | 0.1363 ± 0.0076 |
| LSDS++ | | | 79.5794 ± 1.4596 | 0.2230 ± 0.0683 | | 48.1309 ± 1.8013 | 0.2050 ± 0.0335 | | 26.7825 ± 0.4441 | 0.1832 ± 0.0027 |
| LS++ | iris(150, 4) | 3 | 80.2948 ± 1.4146 | 0.1811 ± 0.1508 | 5 | 47.9004 ± 2.1091 | 0.1299 ± 0.0322 | 10 | 27.0137 ± 0.2987 | 0.2350 ± 0.1657 |
| BanditPAM++ | | | **79.1369 ± 0.0946** | 0.5786 ± 0.6218 | | 46.9289 ± 0.0497 | 0.3037 ± 0.0514 | | **26.6409 ± 0.1368** | 0.4544 ± 0.1304 |
| Ous | | | 80.9941 ± 1.9905 | 0.2752 ± 0.0218 | | 50.6115 ± 1.8605 | 0.3150 ± 0.0319 | | 31.9050 ± 4.8661 | **0.0011 ± 0.0007** |
| k-means++ | | | 677.4968 ± 102.5994 | **0.0034 ± 0.0004** | | 433.2457 ± 16.6050 | **0.0013 ± 0.0001** | | 236.8584 ± 4.1258 | **0.0035 ± 0.0033** |
| MLS | | | 589.4259 ± 0.7753 | 0.0410 ± 0.0016 | | **391.0053 ± 7.4491** | 0.0677 ± 0.0030 | | 205.3238 ± 3.1704 | 0.1624 ± 0.0062 |
| LSDS++ | | | 589.3379 ± 1.4684 | 0.1900 ± 0.0189 | | 396.0341 ± 12.5617 | 0.1840 ± 0.016 | | 205.9997 ± 4.4902 | 0.2128 ± 0.0459 |
| LS++ | seeds(210, 7) | 3 | **588.3799 ± 0.6264** | 0.3623 ± 0.4091 | 5 | 391.5538 ± 7.7126 | 0.1619 ± 0.1479 | 10 | 207.2282 ± 9.9328 | 0.1386 ± 0.0453 |
| BanditPAM++ | | | 589.3587 ± 0.4321 | 0.3941 ± 0.0856 | | 401.2967 ± 0.5204 | 0.4950 ± 0.0233 | | **204.6278 ± 0.5762** | 0.4719 ± 0.0364 |
| Ous | | | 590.3719 ± 2.1562 | 0.2585 ± 0.0103 | | 412.7108 ± 14.2730 | 0.2919 ± 0.0117 | | 225.5556 ± 10.4282 | 0.4719 ± 0.0364 |
| k-means++ | | | 673.7093 ± 61.0323 | **0.0027 ± 0.0034** | | 480.4887 ± 36.5133 | **0.0032 ± 0.0038** | | 259.2790 ± 19.0167 | **0.0019 ± 2.0887** |
| MLS | | | 591.0264 ± 1.2844 | 0.0413 ± 0.0022 | | 412.2717 ± 11.4333 | 0.0679 ± 0.0025 | | 233.0858 ± 2.3131 | 0.1808 ± 0.0135 |
| LSDS++ | | | 590.5831 ± 0.5444 | 0.1854 ± 0.0050 | | 412.3793 ± 11.0836 | 0.2129 ± 0.0341 | | 232.1317 ± 1.4002 | 0.1901 ± 0.0019 |
| LS++ | glass(214, 9) | 3 | 591.7814 ± 2.1027 | 0.3063 ± 0.3206 | 5 | **409.3808 ± 9.9240** | 0.1185 ± 0.0658 | 10 | **232.0718 ± 1.4021** | 0.1917 ± 0.1221 |
| BanditPAM++ | | | **590.4112 ± 1.1400** | 0.2913 ± 0.0289 | | 472.0507 ± 1.8650 | 1.1673 ± 0.8264 | | 256.4949 ± 9.8140 | 2.5653 ± 0.7060 |
| Ous | | | 624.1871 ± 59.2087 | 0.3070 ± 0.0263 | | 452.6347 ± 36.7350 | 0.3959 ± 0.0546 | | 349.6041 ± 80.9667 | **0.0012 ± 0.0006** |
| k-means++ | | | 9.4490E+10 ± 5.7E+09 | **0.0064 ± 0.0087** | | 5.9574E+10 ± 2.8E+09 | **0.0037 ± 0.0006** | | 3.2918E+10 ± 9.1E+08 | **0.0262 ± 0.0215** |
| MLS | | | **8.0098E+10 ± 4.1E+08** | 0.0651 ± 0.0030 | | 5.3603E+10 ± 3.6E+08 | 0.1029 ± 0.0051 | | **3.1506E+10 ± 4.9E+08** | 0.2661 ± 0.0145 |
| LSDS++ | | | 8.1120E+10 ± 5.6E+08 | 0.0282 ± 0.0382 | | **5.3517E+10 ± 2.4E+08** | 0.2412 ± 0.0212 | | 3.1699E+10 ± 3.0E+07 | 0.2118 ± 0.0030 |
| LS++ | Who(440, 8) | 3 | 8.0694E+10 ± 2.2E+08 | 0.0835 ± 0.0607 | 5 | 5.4099E+10 ± 9.9E+07 | 0.1393 ± 0.1090 | 10 | 3.2729E+10 ± 2.3E+08 | 0.1374 ± 0.0141 |
| BanditPAM++ | | | 8.1643E+10 ± 1.8E+08 | 1.1234 ± 0.2641 | | 5.8113E+10 ± 7.9E+07 | 2.6745 ± 0.6594 | | 3.7597E+10 ± 8.2E+07 | 3.8934 ± 0.4584 |
| Ous | | | 8.2792E+10 ± 2.8E+08 | 0.2589 ± 0.0149 | | 5.8621E+10 ± 3.2E+08 | 0.3268 ± 0.0080 | | 4.0189E+10 ± 1.9E+08 | 0.6269 ± 0.0814 |
| k-means++ | | | 2933536.2344 ± 303207.7251 | **0.0083 ± 0.0043** | | 1991419.0036 ± 143383.5953 | **0.0124 ± 0.0094** | | 1145322.5900 ± 65050.74 | **0.0096 ± 0.0037** |
| MLS | | | **2566187.0550 ± 10524.6421** | 0.0826 ± 0.0029 | | 1790989.3510 ± 7429.1685 | 0.1334 ± 0.0045 | | 1033273.0030 ± 15612.5191 | 0.3511 ± 0.0187 |
| LSDS++ | | | 2585244.4835 ± 50372.9150 | 0.2385 ± 0.0283 | | 1790466.5975 ± 2555.8529 | 0.2212 ± 0.0025 | | 1023832.8293 ± 8422.3927 | 0.2419 ± 0.0324 |
| LS++ | HCV(572, 12) | 3 | 2587609.7490 ± 49288.0561 | 0.2323 ± 0.2572 | 5 | **1789778.9990 ± 2822.7958** | 0.1320 ± 0.0957 | 10 | **1023768.3390 ± 12822.3782** | 0.1701 ± 0.0885 |
| BanditPAM++ | | | 3136455.9693 ± 3678.1983 | 1.9193 ± 0.7104 | | 1934469.8632 ± 59989.9421 | 2.0508 ± 0.3165 | | 1299891.1575 ± 4938.8460 | 4.5757 ± 1.1338 |
| Ous | | | 3449750.6830 ± 153601.3066 | 0.3066 ± 0.0429 | | 2854923.0060 ± 297723.1772 | 0.3989 ± 0.0538 | | 2158395.9250 ± 276279.0021 | 0.9130 ± 0.2162 |
| k-means++ | | | 171489.6701 ± 2650.3350 | **0.0536 ± 0.0081** | | 155222.1022 ± 5454.1228 | **0.0229 ± 0.0320** | | 126569.1958 ± 2963.7521 | **0.0447 ± 0.0350** |
| MLS | | | 167903.1180 ± 1857.3234 | 0.1487 ± 0.0062 | | **141810.7915 ± 467.8543** | 0.2057 ± 0.0087 | | 118420.3319 ± 622.6128 | 0.4891 ± 0.0296 |
| LSDS++ | | | 168369.1680 ± 2452.0328 | 0.6063 ± 0.0978 | | 142304.5431 ± 774.0083 | 0.6214 ± 0.1022 | | 118280.3486 ± 505.2902 | 0.6435 ± 0.0364 |
| LS++ | TRR(5456, 24) | 3 | **166839.3052 ± 1179.3739** | 1.1227 ± 0.6979 | 5 | 142451.1482 ± 855.7568 | 1.9264 ± 0.7808 | 10 | **118149.6283 ± 513.2673** | 1.3803 ± 0.9263 |
| BanditPAM++ | | | 170295.3619 ± 577.6118 | 16.1555 ± 3.0836 | | 143081.7860 ± 41.0930 | 18.3144 ± 2.4909 | | 118273.6474 ± 158.3774 | 66.2000 ± 4.0447 |
| Ous | | | 167123.6863 ± 1672.7972 | 0.2831 ± 0.0037 | | 142957.2886 ± 1337.6564 | 0.3013 ± 0.0157 | | 119295.0631 ± 1353.9847 | 0.3378 ± 0.0136 |
| k-means++ | | | 125974.5199 ± 8535.3156 | **0.0967 ± 0.0743** | | 63808.1555 ± 7709.2704 | **0.1036 ± 0.0412** | | 28715.5645 ± 1627.9409 | **0.3084 ± 0.3052** |
| MLS | | | **115109.9601 ± 496.8671** | 4.5662 ± 0.2834 | | 56324.6142 ± 163.7329 | 11.2501 ± 2.3645 | | 24900.2479 ± 350.4709 | 34.1285 ± 1.9756 |
| LSDS++ | | | 117140.2662 ± 3071.61 | 6.4798 ± 0.6575 | | 56386.2449 ± 204.3293 | 6.8931 ± 0.6771 | | **24780.4470 ± 303.2270** | 7.6977 ± 0.8895 |
| LS++ | UrbanGB(360,177, 2) | 3 | 117413.8981 ± 3560.6927 | 8.4238 ± 2.2973 | 5 | 56341.0427 ± 222.1342 | 17.1557 ± 2.9083 | 10 | 24835.4868 ± 400.0881 | 15.0960 ± 3.4364 |
| BanditPAM++ | | | 119056.4776 ± 60.0795 | 1398.2442 ± 92.5744 | | **56181.3117 ± 11.2802** | 3348.5389 ± 494.1768 | | 25029.3040 ± 21.5183 | 5144.3904 ± 106.0113 |
| Ous | | | 119006.9273 ± 4346.7823 | 0.3242 ± 0.0261 | | 56937.8839 ± 868.1835 | 0.3318 ± 0.0127 | | 25352.2220 ± 855.9602 | 0.4082 ± 0.0082 |

balance the efficiency of the algorithm is impractical, we instead manage the bandit process by fixing the maximum sample size at 50,000.

Table 5 presents the results for datasets SYN, USC_1990, and SUSY, with a fixed number of clusters $k = 50$ and a maximum sample size of 50,000, while the Metropolis-Hastings steps vary from 20 to 100. The results indicate that increasing the number of Metropolis-Hastings steps can improve the overall clustering quality, with also a slight increase in the running time. However, the overall clustering performance remains consistent, demonstrating that the algorithm is robust to variations in the number of Metropolis-Hastings steps. This indicates that fixing the number of transition steps at 20 is sufficient to achieve high-quality clustering results with fast running time.

Table 5: Results with varying Metropolis-Hastings steps on datasets SYN, USC_1990, and SUSY, where $k$ is fixed at $50$ and the maximum sample size is fixed at 50,000.

| Method | Datasets | Steps (Transition) | Cost | Time(s) |
|---|---|---|---|---|
| Ours | SYN | 20 | 3.0940E+04±2.1E+03 | 1.42 |
| | | 40 | 3.1489E+04±2.1E+03 | 1.46 |
| | | 60 | 3.1231E+04±1.1E+03 | 1.71 |
| | | 80 | 3.1253E+04±1.1E+03 | 1.73 |
| | | 100 | 3.0834E+04±8.9E+02 | 1.88 |
| Ours | USC_1990 | 20 | 1.2757E+08±1.3E+06 | 3.78 |
| | | 40 | 1.2999E+08±3.5E+06 | 5.47 |
| | | 60 | 1.2771E+08±1.5E+06 | 4.72 |
| | | 80 | 1.2893E+08±3.4E+06 | 4.96 |
| | | 100 | 1.2740E+08±1.6E+06 | 4.35 |
| Ours | SUSY | 20 | 1.9573E+07±2.2E+05 | 3.93 |
| | | 40 | 1.9526E+07±7.4E+04 | 3.75 |
| | | 60 | 1.9615E+07±2.8E+05 | 3.95 |
| | | 80 | 1.9621E+07±2.4E+05 | 3.91 |
| | | 100 | 1.9466E+07±1.0E+05 | 4.26 |

Table 6 presents the results for datasets SYN, USC_1990, and SUSY with a fixed number of clusters $k = 50$ and 20 Metropolis-Hastings steps, while the maximum sample size varies from 50,000 to 250,000 during the bandit process. The results show that increasing the sample size improves the clustering quality, with also a slight increase in the running time. However, the the overall clustering

performance remains stable, indicating that the algorithm is robust to different choices of maximum sample sizes. Fixing the sample size at 50,000 is sufficient to achieve high-quality clustering solutions with fast running time.

Table 6: Results with varying maximum sample size on datasets SYN, USC_1990, and SUSY, where $k$ is fixed at $50$ and the Metropolis-Hastings step is fixed at 20.

| Method | Datasets | Maximum Samples | Cost | Time(s) |
|---|---|---|---|---|
| Ours | SYN | 50,000 | 3.1404E+04±1.1E+03 | 1.38 |
| | | 100,000 | 3.2266E+04±2.4E+03 | 1.48 |
| | | 150,000 | 3.0020E+04±7.9E+02 | 1.79 |
| | | 200,000 | 3.0758E+04±1.9E+03 | 2.18 |
| | | 250,000 | 3.0456E+04±1.3E+03 | 2.17 |
| Ours | USC_1990 | 50,000 | 1.2790E+08±3.2E+06 | 4.55 |
| | | 100,000 | 1.2734E+08±9.8E+05 | 4.14 |
| | | 150,000 | 1.2756E+08±1.6E+06 | 3.98 |
| | | 200,000 | 1.2761E+08±1.4E+06 | 4.73 |
| | | 250,000 | 1.2859E+08±1.8E+06 | 4.23 |
| Ours | SUSY | 50,000 | 1.9563E+08±1.6E+05 | 3.57 |
| | | 100,000 | 1.9576E+08±1.7E+05 | 3.97 |
| | | 150,000 | 1.9500E+08±7.4E+04 | 4.33 |
| | | 200,000 | 1.9713E+08±2.6E+05 | 3.61 |
| | | 250,000 | 1.9479E+08±4.9E+04 | 5.20 |

### E.2 Comparison with Coresets Methods

In this subsection, we give additional experiments by comparing our proposed BanditFastLS algorithm with coresets methods.

**Algorithms.** We compare our BanditFastLS algorithm against 2 different fast coresets construction methods, which are summarized as follows.

- Lightweight Coreset: The lightweight coreset is constructed by sampling data points using a mixture of sampling distributions [5]. The sensitivity is calculated by $s(p) = 1/|P| + \delta^2(p, \mu)/\delta^2(P, \mu)$, where $P$ is the given dataset, $\mu$ is the 1-means solution for $P$. This method runs in $O(nd/\epsilon)$ time but provides a weaker guarantee, where there is an $\epsilon\delta^2(P, \mu)$ term of additive error for some $\epsilon \in (0, 1)$.
- Fast-Coreset: This is the coreset construction method proposed in [18], where aspect ratio and rejection sampling methods are used to construct the coresets. The proposed algorithm can run in $\tilde{O}(nd)$ time for constructing a coreset with $(1 + \epsilon)$-approximation guarantee.

**Parameter Settings.** In all datasets, a preprocessing step is used to remove the data points that are co-located at the same position. Following the settings in [18], we default the coreset size as $40k$ and we only run the dimension-reduction step on datasets with high dimensions ($d \geq 50$).

**Experimental Setup.** For a fair comparison, we report both the time taken to construct the coreset and the overall time required to obtain the final clustering solutions. After constructing the coresets, we use the LS++ algorithm [25], which provides significantly better clustering quality than other local search algorithms, to finalize the clustering centers on the coresets. Each algorithm is executed 10 times, and we report the average clustering costs along with deviations and the average running time.

**Datasts.** We mainly compare our algorithm with coresets methods on large-scale datasets, as local search methods have already demonstrated fast and accurate clustering performance on smaller datasets. Large-scale datasets include SYN (1M × 2), USC_1990 (2.45M × 68), SUSY (5M × 17) and HIGGS (11M × 27) from UCI Machine Learning Repository [6]. We also include a larger dataset SIFT (100M × 128) [7].

---

[6]https://archive.ics.uci.edu/ml/index.php
[7]http://corpus-texmex.irisa.fr/

**Results.** Tables 7 and 8 present a comparison between coreset construction time and the final clustering time for our algorithm on large-scale datasets. The results show that our algorithm outperforms coreset methods, achieving a 20x speedup on average compared to state-of-the-art coreset construction techniques.

Table 7: Comparison of the coreset construction time and the final clustering time of our algorithm on the SYN, USC_1990, and SUSY datasets.

| Method | Dataset | $k$ | Time(s) | Dataset | $k$ | Time(s) | Dataset | $k$ | Time(s) |
|---|---|---|---|---|---|---|---|---|---|
| Fast-Coreset LightWeight Ours | SYN | 10 | 407.41 2.09 **0.41** | USC_1990 | 10 | 1052.54 96.15 **1.12** | SUSY | 10 | 918.94 54.28 **3.67** |
| Fast-Coreset LightWeight Ours | SYN | 20 | 445.09 2.11 **0.52** | USC_1990 | 20 | 1186.93 95.25 **1.24** | SUSY | 20 | 954.93 55.13 **2.65** |
| Fast-Coreset LightWeight Ours | SYN | 30 | 441.29 2.09 **0.69** | USC_1990 | 30 | 1199.86 94.91 **1.69** | SUSY | 30 | 917.64 54.72 **3.42** |
| Fast-Coreset LightWeight Ours | SYN | 50 | 598.33 2.09 **0.95** | USC_1990 | 50 | 1034.49 95.49 **2.10** | SUSY | 50 | 941.23 54.52 **4.22** |
| Fast-Coreset LightWeight Ours | SYN | 100 | 524.78 2.12 **1.79** | USC_1990 | 100 | 1037.78 94.49 **3.58** | SUSY | 100 | 903.07 54.53 **6.54** |

Table 8: Comparison of coreset construction time and final clustering time for our algorithm on the datasets HIGGS and SIFT.

| Method | Dataset | $k$ | Time(s) | Dataset | $k$ | Time(s) |
|---|---|---|---|---|---|---|
| Fast-Coreset LightWeight Ours | HIGGS | 10 | 2311.19 185.56 **5.73** | SIFT | 10 | > 24h 7687.46 **99.81** |
| Fast-Coreset LightWeight Ours | HIGGS | 20 | 2479.21 184.01 **5.73** | SIFT | 20 | > 24h 7031.09 **101.33** |
| Fast-Coreset LightWeight Ours | HIGGS | 30 | 2193.73 183.76 **6.92** | SIFT | 30 | > 24h 7007.87 **105.85** |
| Fast-Coreset LightWeight Ours | HIGGS | 50 | 2126.65 184.81 **7.88** | SIFT | 50 | > 24h 6964.60 **125.89** |
| Fast-Coreset LightWeight Ours | HIGGS | 100 | 2195.06 184.92 **9.87** | SIFT | 100 | > 24h 6965.15 **163.82** |

Tables 9 and 10 present a comparison between coreset-based local search methods and our Bandit-FastLS algorithm on large-scale datasets. The results show that our algorithm not only delivers better clustering quality but also provides at least a 10x speedup on the running time. On average, our algorithm achieves a 5% improvement on clustering quality, while the running time is 50 times faster than coreset-based local search methods.

Table 9: Comparison of the coreset construction methods and our algorithm on SYN, USC_1990, and SUSY datasets using LS++ as the solver on the coresets.

| Method | Dataset | $k$ | Cost | Time(s) | Dataset | $k$ | Cost | Time(s) | Dataset | $k$ | Cost | Time(s) |
|---|---|---|---|---|---|---|---|---|---|---|---|---|
| Fast-Coreset | SYN | 10 | 8.80E+05±9.6E+04 | 413.76 | USC_1990 | 10 | 2.59E+08±5.1E+06 | 1070.35 | SUSY | 10 | 3.21E+07±5.2E+05 | 957.71 |
| LightWeight | | | **5.97E+05±2.0E+04** | 8.53 | | | 2.59E+08±8.3E+06 | 118.93 | | | 3.24E+07±4.4E+05 | 94.28 |
| Ours | | | 6.76E+05±1.0E+04 | **0.41** | | | **2.40E+08±4.1E+06** | **1.12** | | | **3.01E+07±2.3E+05** | **3.67** |
| Fast-Coreset | SYN | 20 | 2.03E+05±9.8E+03 | 451.46 | USC_1990 | 20 | 1.97E+08±2.4E+06 | 1209.99 | SUSY | 20 | 2.67E+07±1.1E+05 | 997.49 |
| LightWeight | | | 1.72E+05±8.4E+03 | 8.72 | | | 1.92E+08±2.5E+06 | 123.08 | | | 2.61E+07±1.8E+05 | 103.83 |
| Ours | | | **1.62E+05±4.9E+03** | **0.52** | | | **1.86E+08±1.3E+06** | **1.24** | | | **2.50E+07±2.2E+05** | **2.65** |
| Fast-Coreset | SYN | 30 | 9.15E+04±5.4E+03 | 448.03 | USC_1990 | 30 | 1.66E+08±7.0E+05 | 1226.02 | SUSY | 30 | 2.37E+07±5.5E+04 | 965.04 |
| LightWeight | | | 8.02E+04±2.4E+03 | 8.94 | | | 1.61E+08±1.1E+06 | 127.43 | | | 2.36E+07±1.2E+05 | 101.54 |
| Ours | | | **7.76E+04±1.8E+03** | **0.69** | | | **1.57E+08±3.2E+06** | **1.69** | | | **2.23E+07±1.5E+05** | **3.42** |
| Fast-Coreset | SYN | 50 | 3.34E+04±9.0E+02 | 636.26 | USC_1990 | 50 | 1.75E+08±1.4E+07 | 1065.87 | SUSY | 50 | 2.07E+07±1.1E+05 | 996.84 |
| LightWeight | | | 3.10E+04±9.8E+02 | 9.56 | | | 1.31E+08±1.8E+06 | 138.59 | | | 2.05E+07±1.1E+05 | 110.29 |
| Ours | | | **3.09E+04±2.1E+03** | **0.95** | | | **1.28E+08±1.6E+06** | **2.10** | | | **1.95E+07±2.2E+05** | **4.22** |
| Fast-Coreset | SYN | 100 | **8.40E+03±7.4E+01** | 534.09 | USC_1990 | 100 | 1.05E+08±4.4E+05 | 1092.78 | SUSY | 100 | 1.73E+07±8.9E+04 | 981.29 |
| LightWeight | | | 8.77E+03±9.1E+02 | 11.12 | | | 1.03E+08±3.5E+06 | 161.66 | | | 1.71E+07±5.5E+04 | 128.02 |
| Ours | | | 1.17E+04±2.1E+03 | **1.79** | | | **9.98E+07±1.5E+06** | **3.58** | | | **1.63E+07±2.9E+04** | **6.54** |

Table 10: Comparison of the coreset construction methods and our algorithm on HIGGS and SIFT datasets using LS++ as the solver on the coresets.

| Method | Dataset | $k$ | Cost | Time(s) | Dataset | $k$ | Cost | Time(s) |
|---|---|---|---|---|---|---|---|---|
| Fast-Coreset | HIGGS | 10 | 1.6521E+08±5.4E+05 | 2422.77 | SIFT | 10 | - | > 24h |
| LightWeight | | | 1.6548E+08±5.9E+05 | 297.14 | | | 1.1044E+13±3.2E+10 | 8853.91 |
| Ours | | | **1.5601E+08±5.4E+05** | **5.73** | | | **1.0541E+13±3.88E+10** | **99.81** |
| Fast-Coreset | HIGGS | 20 | 1.4994E+08±5.4E+05 | 2587.18 | SIFT | 20 | - | > 24h |
| LightWeight | | | 1.4986E+08±8.2E+05 | 282.25 | | | 1.0161E+13±1.4E+10 | 8470.58 |
| Ours | | | **1.4335E+08±5.2E+05** | **5.73** | | | **9.6759E+12±1.67E+10** | **101.33** |
| Fast-Coreset | HIGGS | 30 | 1.4100E+08±6.5E+05 | 2310.14 | SIFT | 30 | - | > 24h |
| LightWeight | | | 1.4117E+08±2.7E+05 | 291.58 | | | 9.6357E+12±6.4E+09 | 8586.20 |
| Ours | | | **1.3527E+08±2.7E+05** | **6.92** | | | **9.2386E+12±1.47E+10** | **105.85** |
| Fast-Coreset | HIGGS | 50 | 1.3131E+08±2.1E+05 | 2263.74 | SIFT | 50 | - | > 24h |
| LightWeight | | | 1.3135E+08±5.1E+05 | 316.89 | | | 9.1203E+12±7.1E+09 | 9267.05 |
| Ours | | | **1.2636E+08±3.3E+05** | **7.88** | | | **8.7442E+12±1.08E+10** | **125.89** |
| Fast-Coreset | HIGGS | 100 | 1.1935E+08±3.2E+05 | 2398.43 | SIFT | 100 | - | > 24h |
| LightWeight | | | 1.1941E+08±1.0E+05 | 373.28 | | | 8.4956E+12±2.8E+09 | 10891.71 |
| Ours | | | **1.1434E+08±1.3E+05** | **9.87** | | | **8.1305E+12±4.98E+09** | **163.82** |

