# OpenReview forum: "Fast Local Search Algorithms for Clustering with Adaptive Sampling and Bandit Strategies"
_NeurIPS.cc/2025/Conference — NeurIPS 2025 poster_

### Official Review · Reviewer_BsVn · 2025-06-29

**Clarity:** 2
**Significance:** 4
**Originality:** 3
**Rating:** 4
**Confidence:** 3

**Summary:**

The paper designs faster algorithms for k-means in $R^d$, tackling the challenge of achieving running time better than $O(ndk)$, which is obviously important when all three parameters $n,d,k$ are large. In essence, the goal is to decouple the product $nk$, and obtain running time like $O(nd + d k^2)$. The paper designs two algorithms, that achieve O(1) approximation for k-means in time $\tilde{O}(nd + \sqrt{n} k^4 d)$ under some assumption, and better time $\tilde{O}(nd + k^4 d)$ under more assumptions. The second result can be extended to k-median without assumptions.

The said assumptions about the data are somewhat indirect, and roughly assert that for the algorithm can evaluate a potential swap operation on the current set of centers using a random point from the data. The general approach is to speed up the well-known local search method (start with random centers and repeatedly swap one center), using two ideas. One idea is to represent the distances approximately using a randomized tree embedding based on [12], because the tree structure can quickly sample proportional to the tree distances, and also quickly processing an update that changes one of the centers. A second idea is to model a swap of one center as a bandit problem, and estimate the benefit of each swap using a sample of $\tilde{O}(k^2)$ points, and here the assumption is crucial.

The empirical evaluation compares the new algorithm to 4 previous algorithms, with real-life datasets of different sizes, and with k ranging between 10 and 100. he new algorithm is faster than all other algorithms, especially as k gets larger, and the accuracy is comparable to the best among the 4 previous ones.

**Questions:**

Your work is important when k is large, but in that case why is it important to have exactly k clusters, and not employ bicriteria approximation, which allows $O(k)$ clusters, or even uniform facility location where the number of clusters is controlled indirectly via a penalty term?

In Lemma 1, can you explain how this bound on the expectation of the tree distance $\delta_T$, but not on the expectation of its square, is useful for k-means, which corresponds to squared distances? In addition, why do you use [12] and not the well-known tree embedding that corresponds to a randomly shifted quadtree, and has stretch bound $O(d\log\Delta)$ instead of $O(d)$ in Lemma 1?

In Lemma 4, what is the intuition for the term $\sqrt n$, I mean where it is coming from? In addition, this lemma shows that each point x is sampled almost with the right probability, up to factor $\sqrt n$; can we use $O(\sqrt n)$ such samples and rejection sampling to obtain one sample with a precisely correct probability? Doesn't this require computing many distances in $R^d$ (not in the tree) which would take too much time?

In assumption 2, what does it mean that "P is average" ?  In addition, this assumption sounds very strong, in the sense of making the problem much easier. For example, sampling the points uniformly at rate $(\log n)/k^2$ will reduce the input size by factor almost $k^2$ and the optimal clusters/centers would probably (I have not verified it) change only a little bit. Why do you think this assumption is reasonable? Has it been used before?

**Ethical Concerns:**

["NO or VERY MINOR ethics concerns only"]

**Final Justification:**

The rebuttal does not change my overall evaluation, but does clarify some technical issues.

**Limitations:**

yes

**Paper Formatting Concerns:**

The tables are in a very small font that is barely readable.

**Quality:**

3

**Strengths And Weaknesses:**

The main strength in my opinion is tackling the challenge of breaking below running time $O(ndk)$, particularly for k-means where a randomized tree embedding is usually not effective (it does not provide good guarantees for squared distances). Another advantage is that the new algorithm is indeed faster, both in theory and in practice, while still achieving high accuracy. The approach seems new, even though it is based on known components, like a randomized tree embedding, and formulating swap pair selection as a bandit problem.

The main weakness of the theoretical results is that the assumptions are rather indirect and not intuitive, and in fact refer to the algorithm itself (e.g. the current centers and the proposed swap).

A possible weakness of the empirical evaluation is that it compares the different algorithms' performance on data that essentially does not cluster well. Table 2 shows two datasets, and in both of them, when the number of clusters k doubles (from 20-30 to 50 and then to 100), the clustering cost goes down by about 10%. I suspect that the good accuracy reported here simply misses the intended application, although I am not sure what is a good or standard way to control for this gap.

Bottom line, I really like the direction and the results sound promising, however I have to be skeptic because the theoretical assumptions seem strong and some pieces of the proof do not fit, as mentioned in my questions for the authors.

---

> ### Author Rebuttal · Authors · 2025-07-31
>
> We thank the reviewer for the positive rating and thoughtful comments. Below, we address the concerns.
>
> **Question 1: Your work is important when k is large, but in that case why is it important to have exactly k clusters, and not employ bicriteria approximation, which allows $O(k)$ clusters, or even uniform facility location where the number of clusters is controlled indirectly via a penalty term?**
>
> Response: We thank the reviewer for raising this insightful question. In the standard $k$-means objective, enforcing exactly $k$ centers is crucial for fair comparisons against prior methods. By fixing $k$, it allows for more meaningful comparisons of theoretical results and clearer assessments of algorithmic effectiveness.
>
> We agree with the reviewer that allowing more than $k$ centers to be opened can reduce the clustering cost and improve approximation ratios. Bicriteria approximation is therefore a practical and effective alternative in real-world scenarios. Extending our  framework to obtain bicriteria solutions and achieve better practical performance is an interesting direction for further studies.
>
> As for the uniform facility location problem, to the best of our knowledge, existing local search algorithms do not achieve linear running time. This is mainly because the opening costs may affect the sampling process and complicate the design of sampling distributions. Developing  local search methods with linear runtime for facility location remains an interesting open problem for future work.
>
> **Question 2: In Lemma 1, can you explain how this bound on the expectation of the tree distance $\delta_T$, but not on the expectation of its square, is useful for k-means, which corresponds to squared distances? In addition, why do you use [12] and not the well-known tree embedding that corresponds to a randomly shifted quadtree, and has stretch bound $O(d\log\Delta)$ instead of $O(d)$ in Lemma 1?**
>
> Response: We appreciate the reviewer’s insightful question.
>
> For the $k$-means problem, our rejection sampling relies on a tree embedding to approximate the squared-distance-based sampling distribution. A key challenge is that directly bounding the expectation of squared tree distances through tree metric leads to large distortion. Prior work (e.g., [1]) shows that the expected squared distance distortion under tree embeddings can be as large as $\Omega(n)$, which adds an $O(n)$ factor loss in sampling probabilities and runtime.
>
> To address this, we instead bound the expected tree distance under the $\ell_1$ objective, as stated in Lemma 1. Then, by carefully relating $\ell_1$ objective to squared distances through the Cauchy–Schwarz Inequality, we reduce the distortion in sampling probabilities from $O(n)$ to $\tilde{O}(\sqrt{n})$.
>
> Regarding the quadtree embedding, the method in [12] is indeed based on the well-known quadtree embedding technique for $\ell_1$ distances, which has a distortion of $O(d \log \Delta)$. We apologize for the typo in Lemma 1, where the distortion should be $\tilde{O}(d)$ instead of $O(d)$ to reflect the hidden logarithmic dependence.
>
> To ensure that the $\log \Delta$ factor does not affect the overall runtime, we include a preprocessing step (lines 198–199 in our paper) that rescales the dataset to reduce the aspect ratio $\Delta$ to $\mathrm{poly}(n, d)$. This step runs in $\tilde{O}(nd)$ time and guarantees that $\log \Delta = \tilde{O}(1)$. As a result, the distortion becomes $\tilde{O}(d)$, which only contributes an $\tilde{O}(1)$ overhead and does not change the overall time complexity of our algorithm. Thus, the quadtree embedding remains compatible with our rejection sampling framework.
>
> We will clarify this in the revised version.
>
> [1] Cohen-Addad et al. Streaming euclidean k-median and k-means with o(log n) space. Proc. FOCS 2023, pp. 883-908.
>
> **Question 3：In Lemma 4, what is the intuition for the term $O(\sqrt{n})$, I mean where it is coming from? In addition, this lemma shows that each point $x$ is sampled almost with the right probability, up to factor $O(\sqrt{n})$; can we use such samples and rejection sampling to obtain one sample with a precisely correct probability? Doesn't this require computing many distances in $R^d$ (not in the tree) which would take too much time?**
>
> Response: We thank the reviewer for the thoughtful question.
>
> The $O(\sqrt{n})$ factor loss arises from approximating squared distances using $\ell_1$ objective in the sampling probability. For our rejection sampling process, to avoid the $O(n)$ factor loss caused by squared distance embedding, the $\ell_1$ distances are used. However, we still need to relate the $\ell_1$-based sampling distribution to the one suitable for squared distances. This introduces an $O(\sqrt{n})$ factor loss through Cauchy–Schwarz inequality.
>
> For the sampling process, the samples drawn from rejection sampling cannot recover the precise sampling probabilities. However, our goal is to ensure that we can sample a “good” point for local search swaps with constant probability. For these points, it is observed that the clustering cost of them account for a certain fraction of the total clustering cost (as shown in Lemma B.3). Since our sampling probabilities are within an $O(1/\sqrt{n})$ factor, by repeating the rejection sampling process for $\tilde{O}(\sqrt{n})$ rounds, it suffices to select a good point with constant probability.
>
> As for computation cost, each round of rejection sampling only requires computing the true squared distance from a single candidate point to its nearest center in $\mathbb{R}^d$. Thus, the total cost over $\tilde{O}(\sqrt{n})$ rounds is $\tilde{O}(\sqrt{n}kd)$, which does not significantly impact the overall runtime.
>
> **Question 4: In assumption 2, what does it mean that $P$ is average"? In addition, this assumption sounds very strong, in the sense of making the problem much easier. For example, sampling the points uniformly at rate $\log(n)/k^2$ will reduce the input size by factor almost $k^2$ and the optimal clusters/centers would probably (I have not verified it) change only a little bit. Why do you think this assumption is reasonable? Has it been used before?**
>
> Response: We thank the reviewer for this thoughtful question. In Assumption 2, $P$ is “average” means that each optimal cluster has size $\Omega(k^2)$, which implies the clustering instance is not heavily imbalanced.
>
> This type of assumption is commonly used in the clustering literature. Prior works on clustering algorithm design (e.g., [2]–[4]) often assume that each optimal cluster has size $\Omega(n/k)$ to ensure meaningful sampling guarantees. Our assumption, which requires each cluster to have size $\Omega(k^2)$, is consistent with these when $k^3 = O(n)$. This is a reasonable setting in practice, as it typically holds that $n$ is much larger than $k$.
>
> Regarding the reviewer’s observation about uniform sampling at rate $\log(n)/k^2$, we agree with the reviewer that this helps to reduce the input size to $\tilde{O}(n/k^2)$ while guaranteeing the theoretical loss (e.g., [2]). However, this approach often leads to worse clustering quality in practice.
>
> To evaluate this strategy, we conducted additional experiments comparing our method with a baseline that uniformly samples $\beta$ points (ranging from 1,000 to 1,000,000) and applies a fast local search algorithm (LS++) to the subsample. Tables 1–3 give the comparison on large-scale datasets including SUSY (5M) and SIFT (100M). The results show that while using small samples (e.g., 1,000–10,000) yields faster runtimes than ours, it leads to at least a 5% increase in clustering cost. To match the clustering quality of ours, it requires to sample over 500,000 points. in which case our algorithm runs much faster than the uniform sampling baseline.
>
> [2] Huang et al. The power of uniform sampling for k-median. Proc. ICML 2023, pp. 13933-13956.
>
> [3] Meyerson et al. A k-median algorithm with running time independent of data size. Machine Learning, 2004, 56(1): 61-87.
>
> [4] Huang et al. Bi-criteria Sublinear Time Algorithms for Clustering with Outliers in High Dimensions. Proc. COCOON 2024, pp. 91-103.
>
> *Table 1: Comparisons with uniform sampling on SUSY*
>
> |**Dataset**|**k**|**Method**|**Cost**|**Time(s)**|
> |---|---|---|---|---|
> |SUSY|10|size=1000|3.13E+07$\pm$4.3E+05|0.49|
> |||size=10000|3.05E+07$\pm$1.6E+05|**0.43**|
> |||size=100000|3.05E+07$\pm$2.1E+05|9.32|
> |||size=500000|3.02E+07$\pm$1.1E+05|47.03|
> |||size=1000000|3.04E+07$\pm$3.0E+05|558.41|
> |||Ours|**3.01E+07$\pm$2.3E+05**|3.67|
> ||50|size=1000|2.18E+07$\pm$1.3E+05|0.89|
> |||size=10000|2.01E+07$\pm$1.4E+05|**0.83**|
> |||size=100000|1.96E+07$\pm$4.1E+04|20.64|
> |||size=500000|1.96E+07$\pm$1.5E+05|83.91|
> |||size=1000000|1.96E+07$\pm$3.1E+05|1052.03|
> |||Ours|**1.95E+07$\pm$2.2E+05**|4.22|
> ||100|size=1000|1.89E+07$\pm$1.2E+05|0.94|
> |||size=10000|1.70E+07$\pm$6.7E+04|**0.81**|
> |||size=100000|1.65E+07$\pm$3.7E+04|20.47|
> |||size=500000|1.64E+07$\pm$4.8E+04|99.38|
> |||size=1000000|1.64E+07$\pm$2.5E+05|1647.13|
> |||Ours|**1.63E+07$\pm$2.9E+04**|6.54|
>
> *Table 2: Comparisons with uniform sampling on SIFT*
>
> |**Dataset**|**k**|**Method**|**Cost**|**Time**|
> |---|---|---|---|---|
> |SIFT|10|size=1000|1.08E+13$\pm$6.8E+10|3.46|
> |||size=10000|1.07E+13$\pm$7.6E+10|**2.89**|
> |||size=100000|1.06E+13$\pm$2.1E+10|31.72|
> |||size=500000|1.06E+13$\pm$1.9E+10|139.13|
> |||size=1000000|1.06E+13$\pm$7.5E+09|2814.43|
> |||Ours|**1.05E+13$\pm$3.9E+10**|99.81|
> ||50|size=1000|9.41E+12$\pm$1.4E+10|**2.74**|
> |||size=10000|8.90E+12$\pm$4.7E+09|3.69|
> |||size=100000|8.89E+12$\pm$4.8E+10|42.26|
> |||size=500000|8.85E+12$\pm$1.3E+10|206.24|
> |||size=1000000|8.80E+12$\pm$1.5E+10|4693.98|
> |||Ours|**8.74E+12$\pm$1.1E+10**|125.89|
> ||100|size=1000|9.17E+12$\pm$1.6E+10|**2.73**|
> |||size=10000|8.35E+12$\pm$2.1E+10|3.59|
> |||size=100000|8.21E+12$\pm$8.0E+09|44.71|
> |||size=500000|8.20E+12$\pm$8.1E+09|232.58|
> |||size=1000000|8.15E+12$\pm$1.9E+09|5537.51|
> |||Ours|**8.13E+12$\pm$5.0E+09**|168.32|

---

> ### Comment · Reviewer_BsVn · 2025-08-03
>
> Thanks for the detailed rebuttal. It does not change my overall evaluation. but I appreciate the correction regarding Lemma 1 (a randomly shifted quadtree has an extra O(log n) factor in the expected distortion, which predates [12]) and explaining what is the source of the $\sqrt{n}$ factor.

---

### Official Review · Reviewer_MdzR · 2025-06-30

**Clarity:** 3
**Significance:** 3
**Originality:** 3
**Rating:** 5
**Confidence:** 3

**Summary:**

This paper introduces two fast local-search algorithms for Euclidean $k$-means and $k$-median that avoid the traditional $\Omega(n d k)$ running time for local search algorithms.

1. BanditLS combines an adaptive $D^2$-sampling surrogate—based on a binary tree that stores distance weights and is updated per swap—with a multi-armed-bandit selection rule that groups swap pairs and prunes low-impact arms. Under a sub-Gaussian prior on swap-pair gains, it achieves a constant-factor approximation for k-means in expected $\tilde O(nd + \sqrt{n} k^4 d)$ time.
1. A second variant replaces the tree walk by a Metropolis–Hastings approximation of D²-sampling, removing the $\sqrt{n}$ factor under some additional assumptions on optimal clusters, yielding $\tilde O(nd + k^4 d)$ time.
1. The same adaptive-sampling idea, plus weighted estimators, is extended to $k$-median, giving expected $\tilde O(nd + \sqrt{n} k^3 d)$ runtime without distributional assumptions

Large-scale experiments show running-time improvements over prior local-search methods and coreset-based techniques while outputting high-quality cluster centers.

**Questions:**

1. How reasonable are the sub-Gaussian assumptions for Theorems 1 and 2?
1. Does the algorithm scale to high-dimensional datasets? The experiments seems to vary $n, k$ but not $d$.

**Ethical Concerns:**

["NO or VERY MINOR ethics concerns only"]

**Final Justification:**

The rebuttal more than addressed my concerns. I believe this is a nice contribution and maintain my current score.

**Limitations:**

yes

**Quality:**

3

**Strengths And Weaknesses:**

### Strengths
1. To the best of my knowledge, this is the first work that breaks the $O(ndk)$ running-time barrier for local-search-based $k$-means and $k$-medians clustering.
1. Empirical results are pretty convincing: the running time beats $k$-means++ while outputting better quality clusters on reasonably large datasets.
1. The dynamic tree-based importance sampling and best-arm identification ideas may extend to other tasks.


---

### Weaknesses
1. The theoretical approximation ratios are quite large (2000+)
2. The running time guarantees depend on some non-standard assumptions, and it is unclear how reasonable these assumptions are.

---

> ### Author Rebuttal · Authors · 2025-07-31
>
> We thank the reviewer for the positive rating and thoughtful comments. Below, we address the concerns.
>
> **Question 1: How reasonable are the sub-Gaussian assumptions for Theorems 1 and 2?**
>
> Response: We thank the reviewer for this important question. For the $k$-means objective, it is based on squared distances. This makes our assumption-free weighted sampling strategy (originally designed for $k$-median) inapplicable, as it can lead to significant approximation errors in estimating swap gains. To address this issue, we formulate the best swap pair selection as a bandit problem. In the bandit setting, the sub-Gaussian assumption on arm gains is a standard and widely adopted condition for establishing theoretical guarantees (e.g., [1], [2] and [6], [30], [31] in our ref list). Thus, the sub-Gaussian assumption used in Theorems 1 and 2 is both reasonable and well supported by prior work.
>
> Moreover, previous assumptions for bandit-based methods are much stronger than ours. In [6], [30], and [31] (see our ref list), it is assumed that each arm’s gain follows a fixed distribution with mean $\gamma$ and variance 1, or that the variance-to-maximum-gain ratio is bounded by a constant. In contrast, we only require the variance to be bounded by a function of the average clustering cost, making our assumption much weaker than previous ones.
>
> To empirically verify the sub-Gaussian assumption, we use the fact that any variable with range $2\sigma$ is $\sigma$-sub-Gaussian. Thus, we assess sub-Gaussianity by computing the ratio between the gain range and the average clustering cost $\delta^2(P, C)/n$, where the gain of a point $p$ under swap pair $(u, v)$ and a center set $C$ is defined as $\delta^2(p, C \setminus u \cup v) - \delta^2(p, C)$. We run our algorithm 5 times with 400 local search steps and calculate the ratios. In Tables 1 and 2, we give the mean, variance, and maximum of these ratios. The results show that the ratio remains below 50 on most datasets except for dataset SYN and HCV_L. This indicates that the sub-Gaussian assumption is both reasonable and empirically justified in practice.
>
> *Table 1: Ratio of gain range to average cost on large datasets*
>
> |**Dataset**|**$k$**|**Mean**|**Variance**|**Max**|
> |---|---|---|---|---|
> |SYN|10|9.75|11.89|77.92|
> ||30|7.97|18.98|211.73|
> ||50|6.54|19.50|185.31|
> ||100|6.23|25.89|370.32|
> |USC1990|10|3.77|7.96|41.67|
> ||30|1.71|2.81|48.99|
> ||50|1.69|1.33|12.31|
> ||100|1.62|1.17|11.00|
> |SUSY|10|4.05|4.83|34.08|
> ||30|2.99|4.51|33.96|
> ||50|2.87|3.74|28.71|
> ||100|2.41|3.74|42.28|
> |HIGGS|10|1.66|1.14|12.82|
> ||30|1.56|1.25|14.94|
> ||50|1.41|1.08|22.56|
> ||100|1.41|2.04|41.22|
>
> *Table 2: Ratio of gain range to average cost on small datasets*
>
> |**Dataset**|**$k$**|**Mean**|**Variance**|**Max**|
> |---|---|---|---|---|
> |iris|3|4.98|3.32|13.61|
> ||5|3.54|3.16|15.42|
> ||10|2.36|1.21|7.02|
> |seeds|3|3.84|2.39|11.40|
> ||5|2.99|1.46|8.41|
> ||10|2.66|1.45|9.81|
> |glass|3|4.31|3.15|16.21|
> ||5|3.65|3.56|14.74|
> ||10|3.01|3.12|15.88|
> |Who_FL|3|5.37|4.74|32.04|
> ||5|6.19|7.74|37.71|
> ||10|4.67|8.42|59.39|
> |HCV_L|3|7.21|13.31|62.19|
> ||5|8.34|19.14|96.74|
> ||10|3.68|8.22|78.61|
> |TRR_FL|3|1.31|0.34|2.53|
> ||5|1.36|0.41|2.75|
> ||10|1.24|0.38|2.73|
> |urbanGB_L|3|4.89|3.11|14.68|
> ||5|5.47|4.22|22.74|
> ||10|5.37|5.16|28.84|
>
> [1] Baharav T Z, Kang R, Sullivan C, et al. Adaptive sampling for efficient softmax approximation[C]//Proceedings of the 38th International Conference on Neural Information Processing Systems. 2024: 117580-117613.
>
> [2] Tiwari M, Kang R, Lee J Y, et al. Faster maximum inner product search in high dimensions[C]//Proceedings of the 41st International Conference on Machine Learning. 2024: 48344-48361.
>
> **Question 2: Does the algorithm scale to high-dimensional datasets? The experiments seems to vary n, k but not d.**
>
> Response: We thank the reviewer for this insightful question. Our algorithm consists of two main components: (1) sampling via the Metropolis-Hastings process, and (2) bandit-based swap pair selection. In the Metropolis-Hastings step, samples are drawn uniformly, and only their distances to the current centers are computed, requiring time linear in the dimension $d$. Similarly, the bandit-based selection evaluates swap gains on a small sample of size $\tilde{O}(k^2)$, where each step also involves distance computations with runtime linear in $d$. Therefore, the overall runtime scales linearly with $d$, making the algorithm suitable for high-dimensional datasets.
>
> To empirically verify this, we conducted additional experiments on several high-dimensional datasets from diverse domains. These include: (1) the Twin Gas Sensor Arrays (TGas) dataset with 40,000 points and 2,000 dimensions; (2) the CIFAR-10 image dataset represented as 10,000 flattened 3,072-dimensional vectors; and (3) the Electricity Load Diagrams (ELD) dataset with 370 instances and 140,256 dimensions. These datasets were selected to cover a wide range of $d$ values and application domains.
>
> Tables 3-5 give the comparison results on these datasets. It can be seen that our algorithm consistently scales well in all cases, achieving an average 10x speedups over baseline local search methods. Moreover, it maintains comparable or even better clustering quality despite the increase in dimensionality. These findings demonstrate that our method is robust not only to data size $n$ and number of clusters $k$, but also to the dimensionality $d$.
>
> We will include these additional experimental results and corresponding runtime analysis in the appendix of the revised manuscript.
>
> *Table 3: Comparison results on dataset TGas*
>
> |**Dataset**|**k**|**Method**|**Cost**|**Time(s)**|
> |---|---|---|---|---|
> |TGas(400000, 2000)|10|k-means++|2.69E+05$\pm$5.0E+02|**26.77**|
> |||MLS|-|>6h|
> |||LSDS++|2.68E+05$\pm$1.1E+03|984.67|
> |||LS++|2.63E+05$\pm$3.1E+02|482.69|
> |||Ours|**2.62E+05$\pm$1.2E+03**|31.13|
> ||50|k-means++|2.61E+05$\pm$5.9E+02|111.96|
> |||MLS|-|>6h|
> |||LSDS++|2.60E+05$\pm$1.1E+03|1047.36|
> |||LS++|**2.57E+05$\pm$1.6E+02**|624.44|
> |||Ours|2.58E+05$\pm$1.4E+02|**32.53**|
> ||100|k-means++|2.58E+05$\pm$4.0E+02|228.96|
> |||MLS|-|>6h|
> |||LSDS++|2.58E+05$\pm$1.0E+03|1054.94|
> |||LS++|**2.57E+05$\pm$2.4E+02**|881.86|
> |||Ours|2.57E+05$\pm$3.5E+02|**42.04**|
>
> *Table 3: Comparison results on dataset CIFAR10*
>
> |**Dataset**|**k**|**Method**|**Cost**|**Time(s)**|
> |---|---|---|---|---|
> |CIFAR10(10000, 3072)|10|k-means++|8.97e+10$\pm$8.2E+09|**0.43**|
> |||MLS|8.58E+10$\pm$1.5E+09|35.34|
> |||LSDS++|8.86E+10$\pm$1.1E+09|38.93|
> |||LS++|8.63E+10$\pm$4.1E+08|5.52|
> |||Ours|**8.58E+10$\pm$2.0E+08**|2.50|
> ||50|k-means++|7.51E+10$\pm$2.4E+08|**1.18**|
> |||MLS|7.28E+10$\pm$6.1E+08|375.17|
> |||LSDS++|7.47E+10$\pm$9.7E+08|43.45|
> |||LS++|7.28E+10$\pm$2.5E+08|6.64|
> |||Ours|**7.28E+10$\pm$2.3E+08**|6.71|
> ||100|k-means++|7.38E+10$\pm$2.5E+08|**2.19**|
> |||MLS|7.00E+10$\pm$3.7E+08|2855.34|
> |||LSDS++|7.14E+10$\pm$5.7E+08|47.68|
> |||LS++|6.93E+10$\pm$2.2E+08|7.24|
> |||Ours|**6.79E+10$\pm$2.3E+08**|12.24|
>
> *Table 3: Comparison results on dataset ELD*
>
> |**Dataset**|**k**|**Method**|**Cost**|**Time**|
> |---|---|---|---|---|
> |ELD(370, 140256)|10|k-means++|3.62E+06$\pm$8.6E+04|**0.20**|
> |||MLS|3.43E+06$\pm$1.6E+04|11.00|
> |||LSDS++|3.53E+06$\pm$7.6E+05|9.33|
> |||LS++|**3.43E+06$\pm$6.5E+03**|16.69|
> |||Ours|3.43E+06$\pm$7.3E+03|0.59|
> ||50|k-means++|2.12E+06$\pm$2.2E+04|**0.58**|
> |||MLS|2.04E+06$\pm$5.9E+03|76.27|
> |||LSDS++|2.09E+06$\pm$1.9E+05|14.00|
> |||LS++|**1.98E+06$\pm$2.8E+03**|22.77|
> |||Ours|2.02E+06$\pm$4.7E+03|1.06|
> ||100|k-means++|1.69E+06$\pm$3.8E+03|**1.09**|
> |||MLS|1.67E+06$\pm$1.8E+03|419.52|
> |||LSDS++|1.66E+06$\pm$8.6E+03|22.13|
> |||LS++|**1.61E+06$\pm$2.7E+03**|30.86|
> |||Ours|1.65E+06$\pm$9.5E+03|1.60|

---

> > ### Comment · Reviewer_MdzR · 2025-08-02
> >
> > Thanks for the detailed rebuttal. I especially appreciate the additional experimental evidence.
> > Although it may be quite obvious for experts, I believe including the discussion above for the sub-Gaussian assumption will strengthen the paper.
> > I think this should be a nice contribution and maintain my score.

---

### Official Review · Reviewer_sbwN · 2025-06-30

**Clarity:** 3
**Significance:** 2
**Originality:** 3
**Rating:** 5
**Confidence:** 5

**Summary:**

This paper proposes a new local search method to accelerate clustering on large datasets. Under a sub-Gaussian prior assumption on "swap pairs", the authors propose an $O(1)$-approximation algorithm with $\tilde{O}(nd+\sqrt{n}k^4d)$-running time for $k$-Means (and $\tilde{O}(nd+\sqrt{n}k^3d)$-time for $k$-Median without assumptions). Additionally, they achieve a better running time $\tilde{O}(nd+k^4d)$ for $k$-Means under more assumptions on the optimal center set. This is faster than existing local search methods.

In the algorithm, they first convert the dataset by a tree structure, and use an adaptive sampling method to approximate $D^2$-Sampling distribution. Then they use a bandit method to accelerate the swap pair identification. Experimental results demonstrate that the proposed algorithm performs better than existing methods on large datasets, achieving a significant speedup.

**Questions:**

Could the authors provide a comparison between their new algorithm and a faster coreset-based approach over one million data points? For instance, a possible method is to apply uniform sampling to obtain a weak coreset and run $k$-Means++ on the coreset. If additional experimental results could demonstrate that the proposed algorithm still outperforms such coreset-based methods, I think it will increase the contribution.

**Ethical Concerns:**

["NO or VERY MINOR ethics concerns only"]

**Final Justification:**

I appreciated the additional experiments in the author's response. It justifies the most concern on the reasonability of the assumption. Although the assumption is still not easy to be checked before running the algorithm, it provides evidences that they could hold in practice, largely increasing the applicability of the algorithm together with its provably guarantee.

**Limitations:**

Yes

**Quality:**

2

**Strengths And Weaknesses:**

Strengths:
* Compared with the prior work, the algorithm breaks the $O(ndk)$ running time using local search method, which is non-trivial.
* The adaptive sampling method is novel and useful. Combined with a tree structure, it avoids maintaining distances to all data points and reduces the running time.
* The experiments are thorough and convincing, conducted on multiple datasets to validate the algorithm's acceleration performance. The proposed algorithm is practical for large datasets.

Weaknesses
* The Sub-Gaussian Prior assumption seems not mild and difficult to verify in practice. Generally, the assumptions of an algorithm are expected to depend on the dataset and be easy to verify. However, this assumption concerns the “swap pairs” generated during the algorithm's execution, making it difficult to determine whether a given dataset satisfies the assumption in practice.
* As mentioned in the introduction of this paper, there exist coreset-based algorithms achieving $\tilde{O}(nd + k^3d)$ running time, which is better than the new algorithm theoretically. This weakens the contribution.
* "For datasets with sizes over 100 million, the constructions for coresets can require several hours of computation". This conclusion may have some problems. By [Huang et al., 2023], a weak coreset can be obtained by uniform sampling under the balanceness assumption, which is very efficient.
* Empirically, the algorithm improves only on very large datasets with over one million data points and has no significant advantages on small datasets.

[Huang et al., 2023] Huang L, Jiang S H C, Lou J. The power of uniform sampling for k-median[C]//International Conference on Machine Learning. PMLR, 2023: 13933-13956.

---

> ### Author Rebuttal · Authors · 2025-07-31
>
> We thank the reviewer for the thoughtful comments and constructive feedbacks. Below, we address the concerns.
>
> **Weakness 1: The Sub-Gaussian Prior assumption seems not mild and difficult to verify in practice. Generally, the assumptions of an algorithm are expected to depend on the dataset and be easy to verify. However, this assumption concerns the “swap pairs” generated during the algorithm's execution, making it difficult to determine whether a given dataset satisfies the assumption in practice.**
>
> Response: We thank the reviewer for raising this important concern. Unlike the $k$-median problem with $\ell_1$-norm objective, our weighted sampling strategy (which requires no assumptions) cannot be applied to $k$-means due to approximation loss for swap gain estimation under squared distances. To overcome this issue, we model the swap pair selection process as a bandit problem. In this setting, the sub-Gaussianity assumption is both necessary for theoretical guarantees and commonly adopted in prior work (e.g., [6], [30], [31] in our ref list).
>
> We refer to our sub-Gaussian Prior assumption as mild because it is much weaker than those used in prior work. Existing bandit-based clustering methods (e.g., [6], [30], [31]) typically assume that the gain of each swap pair follows a fixed distribution with mean $\gamma$ and variance $1$, or that the ratio between the variance and the maximum gain is bounded by a constant. In contrast, we only require that the variance is bounded by a function of the average cost, making it a milder assumption than previous ones.
>
> To verify our assumptions empirically, we conduct the following experiments. We use the fact that any variable with range $2\sigma$ is $\sigma$-sub-Gaussian. Thus, the ratio between the gain range and the average clustering cost can be used to measure whether the sub-Gaussian assumption is satisfied. For each swap pair $(u,v)$ and center set $C$, we define the gain for a point $p$ as $\delta^2(p, C \setminus u \cup v) - \delta^2(p, C)$. Then the ratio can be calculated as the gain divide by $\delta^2(P, C)/n$. For each dataset, we run our algorithm 5 times with 400 local search steps and record the gain-range-to-average-cost ratio for each swap pair. Tables 1-2 give the mean, variance, and maximum of these ratios. The results show that the maximum ratio is typically bounded by a constant (less than 50) across most datasets, supporting the validity of the sub-Gaussian assumption. Even on datasets like SYN, where the assumption may not strictly hold, our algorithm still achieves over 20× speedup with competitive clustering quality.
>
> *Table 1: Ratio of gain range to the average cost on large datasets*
>
> |**Dataset**|**$k$**|**Mean**|**Variance**|**Max**|
> |---|---|---|---|---|
> |SYN|10|9.75|11.89|77.92|
> ||30|7.97|18.98|211.73|
> ||50|6.54|19.50|185.31|
> ||100|6.23|25.89|370.32|
> |USC1990|10|3.77|7.96|41.67|
> ||30|1.71|2.81|48.99|
> ||50|1.69|1.33|12.31|
> ||100|1.62|1.17|11.00|
> |SUSY|10|4.05|4.83|34.08|
> ||30|2.99|4.51|33.96|
> ||50|2.87|3.74|28.71|
> ||100|2.41|3.74|42.28|
> |HIGGS|10|1.66|1.14|12.82|
> ||30|1.56|1.25|14.94|
> ||50|1.41|1.08|22.56|
> ||100|1.41|2.04|41.22|
>
> *Table 2: Ratio of gain range to the average cost on small datasets*
>
> |**Dataset**|**$k$**|**Mean**|**Variance**|**Max**|
> |---|---|---|---|---|
> |iris|3|4.98|3.32|13.61|
> ||5|3.54|3.16|15.42|
> ||10|2.36|1.21|7.02|
> |seeds|3|3.84|2.39|11.40|
> ||5|2.99|1.46|8.41|
> ||10|2.66|1.45|9.81|
> |glass|3|4.31|3.15|16.21|
> ||5|3.65|3.56|14.74|
> ||10|3.01|3.12|15.88|
> |Who_FL|3|5.37|4.74|32.04|
> ||5|6.19|7.74|37.71|
> ||10|4.67|8.42|59.39|
> |HCV_L|3|7.21|13.31|62.19|
> ||5|8.34|19.14|96.74|
> ||10|3.68|8.22|78.61|
> |TRR_FL|3|1.31|0.34|2.53|
> ||5|1.36|0.41|2.75|
> ||10|1.24|0.38|2.73|
> |urbanGB_L|3|4.89|3.11|14.68|
> ||5|5.47|4.22|22.74|
> ||10|5.37|5.16|28.84|
>
> **Weakness 2: As mentioned in the introduction of this paper, there exist coreset-based algorithms achieving running time of $\tilde{O}(nd+k^3d)$, which is better than the new algorithm theoretically. This weakens the contribution.**
>
> Response: We thank the reviewer for raising this concern. While it is true that some coreset-based algorithms for $k$-means achieve a theoretical runtime of $\tilde{O}(nd + k^3d)$ under no distribution assumptions, this may overlook the practical overhead involved in coreset construction—particularly for large-scale datasets. Experimental results indicate that there is a sharp increase on the coreset construction time as data sizes grow (see Tables 5 and 6 in our appendix). Moreover, coreset-based methods often produce worse clustering quality than algorithms that directly optimize the original objective.
>
> In contrast, our method bypasses the data compression step required by coreset construction and achieves comparable time complexity with much better empirical performance. As demonstrated in our experiments (see our Appendix), it delivers up to 80× speedup over coreset-based methods on large-scale datasets such as SIFT (100M points), while achieving better clustering quality. Therefore, even in the presence of coreset methods with similar time complexity, our approach contributes a more practical and scalable solution with theoretical guarantees.
>
> **Weakness 3 and Question 1: Could the authors provide a comparison between their new algorithm and a faster coreset-based approach over one million data points? For instance, a possible method is to apply uniform sampling to obtain a weak coreset and run -Means++ on the coreset. If additional experimental results could demonstrate that the proposed algorithm still outperforms such coreset-based methods, I think it will increase the contribution.**
>
> Response: We thank the reviewer for the constructive feedbacks. We agree with the reviewer that an unweighted weak coreset can be constructed efficiently via uniform sampling when the dataset exhibits the balance property. While this approach offers excellent runtime performance, it often results in worse clustering qualities. As it targets a different trade-off between accuracy and efficiency, we did not include it in our original comparison.
>
> To address this concern, we conducted additional experiments on large-scale datasets such as USC1990 (1M), HIGGS (11M), and SIFT (100M), where we apply uniform sampling (with sizes ranging from 1,000 to 1,000,000) to construct the coreset followed by $k$-means++ to obtain the clustering solutions. The results (Tables 3-5) show that even with 1 million samples, our method consistently achieves much better clustering quality than uniform coresets while maintaining comparable runtime. On average, our method reduces the clustering cost by 5% compared to the uniform coreset baseline.
>
> We will include these results in the revised version to provide a more complete comparison.
>
> *Table 3: Comparisons with weak coreset on USC1990*
>
> |**Dataset**|**k**|**Method**|**Cost**|**Time(s)**|
> |---|---|---|---|---|
> |USC1990|10|Uniform+KM(size=1000)|2.68E+08$\pm$1.1E+07|0.17|
> |||Uniform+KM(size=10000)|2.57E+08$\pm$1.2E+07|**0.08**|
> |||Uniform+KM(size=100000)|2.59E+08$\pm$1.2E+07|0.31|
> |||Uniform+KM(size=500000)|2.58E+08$\pm$1.3E+07|1.29|
> |||Uniform+KM(size=1000000)|2.54E+08$\pm$4.9E+06|7.52|
> |||Ours|**2.40E+08$\pm$4.1E+06**|1.12|
> ||50|Uniform+KM(size=1000)|1.46E+08$\pm$1.8E+06|**0.08**|
> |||Uniform+KM(size=10000)|1.34E+08$\pm$1.3E+06|0.13|
> |||Uniform+KM(size=100000)|1.33E+08$\pm$6.0E+05|0.34|
> |||Uniform+KM(size=500000)|1.34E+08$\pm$4.4E+05|1.96|
> |||Uniform+KM(size=1000000)|1.36E+08$\pm$4.2E+05|11.76|
> |||Ours|**1.29E+08$\pm$1.6E+06**|2.10|
> ||100|Uniform+KM(size=1000)|1.17E+08$\pm$9.8E+05|**0.06**||
> |||Uniform+KM(size=10000)|1.05E+08$\pm$4.8E+05|0.25|
> |||Uniform+KM(size=100000)|1.04E+08$\pm$1.3E+05|0.52|
> |||Uniform+KM(size=500000)|1.03E+08$\pm$1.6E+05|3.25|
> |||Uniform+KM(size=1000000)|1.03E+08$\pm$5.0E+04|18.14|
> |||Ours|**9.99E+07$\pm$1.5E+06**|3.58|
>
> *Table 4: Comparisons with weak coreset on HIGGS*
>
> |**Dataset**|**k**|**Method**|**Cost**|**Time**|
> |---|---|---|---|---|
> |HIGGS|10|Uniform+KM(size=1000)|1.67E+08$\pm$1.2E+06|0.71|
> |||Uniform+KM(size=10000)|1.69E+08$\pm$2.2E+06|**0.26**|
> |||Uniform+KM(size=100000)|1.66E+08$\pm$4.5E+05|3.28|
> |||Uniform+KM(size=500000)|1.67E+08$\pm$1.1E+05|7.33|
> |||Uniform+KM(size=1000000)|1.67E+08$\pm$7.7E+05|12.03|
> |||Ours|**1.56E+08$\pm$5.4E+05**|5.73|
> ||50|Uniform+KM(size=1000)|1.36E+08$\pm$4.6E+05|**0.77**|
> |||Uniform+KM(size=10000)|1.33E+08$\pm$3.4E+05|0.79|
> |||Uniform+KM(size=100000)|1.32E+08$\pm$4.1E+05|8.06|
> |||Uniform+KM(size=500000)|1.32E+08$\pm$1.5E+05|10.44|
> |||Uniform+KM(size=1000000)|1.32E+08$\pm$2.6E+05|19.75|
> |||Ours|**1.26E+08$\pm$3.3E+05**|7.88|
> ||100|Uniform+KM(size=1000)|1.27E+08$\pm$5.3E+05|**0.45**||
> |||Uniform+KM(size=10000)|1.20E+08$\pm$2.0E+05|0.84|
> |||Uniform+KM(size=100000)|1.20E+08$\pm$1.8E+05|3.48|
> |||Uniform+KM(size=500000)|1.20E+08$\pm$2.0E+05|4.09|
> |||Uniform+KM(size=1000000)|1.19e+08$\pm$2.6E+05|14.68|
> |||Ours|**1.14E+08$\pm$1.3E+05**|9.87|
>
> *Table 4: Comparisons with weak coreset on SIFT*
>
> |**Dataset**|**k**|**Method**|**Cost**|**Time**|
> |---|---|---|---|---|
> |SIFT|10|Uniform+KM(size=1000)|1.10E+13$\pm$8.6E+10|1.32|
> |||Uniform+KM(size=10000)|1.10E+13$\pm$1.7E+10|**1.10**|
> |||Uniform+KM(size=100000)|1.10E+13$\pm$9.2E+09|8.25|
> |||Uniform+KM(size=500000)|1.10E+13$\pm$2.0E+09|14.76|
> |||Uniform+KM(size=1000000)|1.09E+13$\pm$1.3E+08|33.24|
> |||Ours|**1.05E+13$\pm$3.9E+10**|99.81|
> ||50|Uniform+KM(size=1000)|9.51E+12$\pm$2.7E+10|**0.39**|
> |||Uniform+KM(size=10000)|9.21E+12$\pm$6.6E+09|0.58|
> |||Uniform+KM(size=100000)|9.14E+12$\pm$1.4E+10|1.96|
> |||Uniform+KM(size=500000)|9.17E+12$\pm$7.9E+08|7.50|
> |||Uniform+KM(size=1000000)|9.15E+12$\pm$1.5E+09|72.05|
> |||Ours|**8.74E+12$\pm$1.1E+10**|125.89|
> ||100|Uniform+KM(size=1000)|9.17E+12$\pm$5.4E+09|**0.11**|
> |||Uniform+KM(size=10000)|8.58E+12$\pm$1.2E+10|0.58|
> |||Uniform+KM(size=100000)|8.49E+12$\pm$1.2E+10|2.11|
> |||Uniform+KM(size=500000)|8.49E+12$\pm$1.2E+10|7.02|
> |||Uniform+KM(size=1000000)|8.46E+12$\pm$3.7E+09|96.31|
> |||Ours|**8.13E+12$\pm$5.0E+09**|168.32|

---

> > ### Comment · Reviewer_sbwN · 2025-08-02
> >
> > Thank you for the response. It primarily addresses my questions on the assumption and the comparison with the coreset methods.

---

### Official Review · Reviewer_jiTA · 2025-07-03

**Clarity:** 3
**Significance:** 2
**Originality:** 2
**Rating:** 4
**Confidence:** 3

**Summary:**

The paper proposes faster local search algorithms for k-means and k-median clustering using adaptive sampling and bandit strategies, claiming significant runtime improvements while preserving constant-factor approximation guarantees. The authors introduce tree-based data structures to accelerate sampling and model swap selection as a multi-armed bandit problem.

**Questions:**

"The authors mention that 'to ensure fairness, all algorithms perform 400 local search steps' (line 329). However, for large-scale datasets, such a high number of steps is often unnecessary in practice. Can the authors report the performance of each method with fewer steps, such as 50 or 100, to provide a more realistic comparison of practical efficiency?

**Ethical Concerns:**

["NO or VERY MINOR ethics concerns only"]

**Final Justification:**

The  rebuttal resolving circular reasoning concerns. However, some issue remain: (1) the sub-Gaussian assumption is theoretically elegant but impractical to verify in real applications; (2) no improved runtime bounds are provided despite empirical speedups;  (3) presentation issues persist. At the same time, the theoretical framework and synthetic validation provide some foundation for borderline paper.

**Limitations:**

The claimed theoretical guarantees depend heavily on assumptions such as sub-Gaussian priors on swap pairs and specific distributional behavior of optimal clusters (e.g., exponential tails). These assumptions are not standard or justified empirically

**Quality:**

2

**Strengths And Weaknesses:**

Strengths: The use of tree-based structures and bandit modeling is conceptually interesting.
Weaknesses :
    Over-reliance on strong assumptions:
    The claimed theoretical guarantees depend heavily on assumptions such as sub-Gaussian priors on swap pairs and specific distributional behavior of optimal clusters (e.g., exponential tails). These assumptions are not standard or justified empirically, and no sensitivity analysis is provided to show how the algorithm performs when these assumptions are violated.
Presentation:
    The paper has typos and formatting issues.
    The title itself has a typo: “Adpative” instead of “Adaptive, which is unusual.
Experimental results:
Experimental results show that the clustering quality is slightly weaker compared to MLS, which is expected since multi-swap strategies can naturally achieve better results than single-swap approaches.
All methods evaluated in the experiments show only marginal improvements in clustering quality over k-means++, which is unsurprising given that k-means++ already performs well when the clusters are well-separated. In such cases, local search via center swaps is not particularly effective for further improving the clustering quality

---

> ### Author Rebuttal · Authors · 2025-07-31
>
> We thank the reviewer for the thoughtful comments and constructive feedbacks. Below, we address the concerns.
>
> **Weakness 1: These assumptions are not standard or justified empirically, and no sensitivity analysis is provided to show how the algorithm performs when these assumptions are violated.**
>
> Response: We thank the reviewer for raising this concern. For the $k$-median problem, we obtain theoretical guarantees without assumptions by applying weighted sampling on $\ell_1$-norm objectives. However, this approach cannot be used for $k$-means due to the large approximation loss caused by squared distance definitions. To reduce the dependence on both $n$ and $k$ in the runtime, we apply bandit method for best swap pair selection. As shown in the literature (e.g., [6], [30], [31] in our ref list), to obtain theoretical guarantees in bandit settings, sub-Gaussianity is a standard assumption.
>
> Previous assumptions for bandit methods are much stronger than ours. In [6], [30], and [31], it is assumed that the gain of each arm follows a fixed distribution with mean $\gamma$ and variance 1, or the variance-to-maximum-gain ratio is bounded by a constant. In contrast, we only require the variance to be bounded by a function of the average cost, making our assumption much weaker.
>
> To further improve runtime for $k$-means, we design a practical algorithm (Alg 3) based on the Metropolis-Hastings method. To achieve theoretical guarantees, both exponential tail and non-degeneracy assumptions are required in previous work (e.g., [3] in the ref list). In contrast, our analysis only requires an exponential tail and a cluster size condition, making it much weaker than previous ones.
>
> To validate the assumptions, we conduct the following experiments. For the sub-Gaussianity assumption, we use the fact that a variable with range $2\sigma$ is $\sigma$-sub-Gaussian, and measure the ratio between gain range and average clustering cost $\delta^2(P,C)/n$. The gain for a point $p$ under swap $(u, v)$  and center set $C$ is defined as $\delta^2(p, C \setminus u \cup v) - \delta^2(p, C)$. In Tables 1 and 2, we report the mean, variance, and maximum of the ratios calculated during the local search processes. The results show that this ratio is typically bounded by a constant (< 50) across most datasets, verifying the sub-Gaussianity assumption. Even on datasets like SYN, where the assumption may not hold (with large ratio), our algorithm still achieves over 20× speedup, demonstrating the empirical robustness.
>
> For the exponential tail assumption, we evaluate it using approximate solutions from multi-swap local search, since the optimal clusters are unknown. Similar evaluation based on approximate solutions has also been adopted in prior work such as [1]. We rely on the fact that the log-CCDF (Complementary Cumulative Distribution Function) of an exponential distribution is upper bounded by a linear function of the variable. For each approximate cluster, we: (1) compute the log-CCDF of squared distances, and (2) apply linear regression using slope < 0 and p-value < 0.05 as criteria. As shown in Tables 3-4, 7 out of 12 datasets are likely to satisfy the assumption. All datasets with over 1M points meet the condition, while smaller ones (e.g., Glass) may not. Most datasets also satisfy the cluster size condition, though SYN ($k=100$) may not for its small clusters. However, our algorithm achieves over 20× speedup on datasets violating both sub-Gaussian and cluster size assumptions (e.g., SYN), demonstrating the empirical robustness.
>
> [1] Huang et al. The power of uniform sampling for k-median. Proc. ICML 2023, pp. 13933-13956.
>
> *Table 1: Ratio of gain range to average cost on large datasets*
>
> |**Dataset**|**$k$**|**Mean**|**Variance**|**Max**|
> |---|---|---|---|---|
> |SYN|10|9.75|11.89|77.92|
> ||30|7.97|18.98|211.73|
> ||50|6.54|19.50|185.31|
> ||100|6.23|25.89|370.32|
> |USC1990|10|3.77|7.96|41.67|
> ||30|1.71|2.81|48.99|
> ||50|1.69|1.33|12.31|
> ||100|1.62|1.17|11.00|
> |SUSY|10|4.05|4.83|34.08|
> ||30|2.99|4.51|33.96|
> ||50|2.87|3.74|28.71|
> ||100|2.41|3.74|42.28|
> |HIGGS|10|1.66|1.14|12.82|
> ||30|1.56|1.25|14.94|
> ||50|1.41|1.08|22.56|
> ||100|1.41|2.04|41.22|
>
> *Table 2: Ratio of gain range to average cost on small datasets*
>
> |**Dataset**|**$k$**|**Mean**|**Variance**|**Max**|
> |---|---|---|---|---|
> |iris|3|4.98|3.32|13.61|
> ||5|3.54|3.16|15.42|
> ||10|2.36|1.21|7.02|
> |seeds|3|3.84|2.39|11.40|
> ||5|2.99|1.46|8.41|
> ||10|2.66|1.45|9.81|
> |glass|3|4.31|3.15|16.21|
> ||5|3.65|3.56|14.74|
> ||10|3.01|3.12|15.88|
> |Who_FL|3|5.37|4.74|32.04|
> ||5|6.19|7.74|37.71|
> ||10|4.67|8.42|59.39|
> |HCV_L|3|7.21|13.31|62.19|
> ||5|8.34|19.14|96.74|
> ||10|3.68|8.22|78.61|
> |TRR_FL|3|1.31|0.34|2.53|
> ||5|1.36|0.41|2.75|
> ||10|1.24|0.38|2.73|
> |urbanGB_L|3|4.89|3.11|14.68|
> ||5|5.47|4.22|22.74|
> ||10|5.37|5.16|28.84|
>
> *Table 3: Cluster size and tail tests on large datasets*
>
> |**Dataset**|**k**|**MinClusterSize/$k^2$**|**Num of Exponential Tail Clusters (slope<0, pvalue<0.05)**|
> |---|---|---|---|
> |SYN|10|437.75|10|
> ||30|6.51|30|
> ||50|1.43|50|
> ||100|0.14|100|
> |USC1990|10|889.5|10|
> ||30|13.52|30|
> ||50|2.05|50|
> ||100|0.34|100|
> |SUSY|10|652.66|10|
> ||30|39.61|30|
> ||50|6.31|50|
> ||100|0.78|100|
> |HIGGS|10|8907.19|10|
> ||30|143.87|30|
> ||50|11.49|50|
> ||100|1.82|100|
>
> *Table 4: Cluster size and tail tests on small datasets*
>
> |**Dataset**|**k**|**MinClusterSize/$k^2$**|**Num of Exponential Tail Clusters(slope<0, pvalue<0.05)**|
> |---|---|---|---|
> |iris|3|4.33|3|
> ||5|0.52|5|
> ||10|0.04|9|
> |seeds|3|6.77|3|
> ||5|1.24|5|
> ||10|0.11|10|
> |glass|3|2.44|1|
> ||5|0.08|4|
> ||10|0.02|7|
> |Who_FL|3|5.66|3|
> ||5|0.24|4|
> ||10|0.02|8|
> |HCV_L|3|1.22|3|
> ||5|0.28|5|
> ||10|0.03|9|
> |TRR_FL|3|119.33|3|
> ||5|36.96|5|
> ||10|2.05|10|
> |urbanGB_L|3|2449|3|
> ||5|894.52|5|
> ||10|65.14|10|
>
> **Weakness 2: Regarding the typos**
>
> Response: We apologize for the typos. In the revised version, we will carefully correct them to improve the presentation.
>
> **Weakness 3: The runtime comparison is also somewhat unfair, as MLS performs multi-swap operations while LS++ is based on single swaps. There is no clear reason why MLS should be faster than LS++**
>
> Response: We thank the reviewer for raising this concern.
>
> Although MLS is a multi-swap method and LS++ is based on single-swap, as shown in prior work (e.g., [7], [21] in the ref list), both of them are compared together to evaluate the trade-offs in runtime and clustering quality.
>
> While single-swap methods are generally expected to be faster than multi-swap methods, MLS is faster than LS++ because it employs a heuristic that estimates swap gains using a small sample of size $O(k \log k)$ instead of the full dataset. In contrast,  LS++ evaluates exact gains on the entire dataset, incurring a cost of $O(ndk)$ per local search step. This makes MLS more efficient despite its multi-swap nature.
>
> It can be seen that the sampling step is the key factor behind MLS’s speed advantage. Additional experiments show that when MLS is modified to compute exact gains over the full dataset (denoted as MLSP in Tables 5-7), it becomes much slower than LS++.
>
> We will clarify this in our revised version.
>
> **Question 1: Can the authors report the performance of each method with fewer steps, such as 50 or 100, to provide a more realistic comparison of practical efficiency?**
>
> Response: We thank the reviewer for this insightful suggestion. In our experiments, 400 local search steps is set to ensure that all methods have sufficient iterations to converge. In the following (Tables 5-7), we conducted additional experiments on large datasets SUSY (5M), HIGGS (11M) and SIFT (100M) using 50 local search steps. The results show that our algorithm still outperforms other local search methods with 50x to 300x speedups and comparable clustering quality.
>
> *Table 5: Results on SUSY with 50 LS steps*
>
> |**Dataset**|**$k$**|**Method**|**Cost**|**Time(s)**|
> |---|---|---|---|---|
> |SUSY|10|k-means++|3.35E+07$\pm$1.0E+05|3.83|
> |||LS++|3.03E+07$\pm$4.7E+05|150.51|
> |||LSDS++|3.27E+07$\pm$7.4E+05|135.49|
> |||MLS|3.08E+07$\pm$4.2E+05|33.85|
> |||MLSP|**2.94E+07$\pm$6.3E+05**|6286|
> |||Ours|3.05E+07$\pm$2.5E+05|**1.98**|
> |SUSY|50|k-means++|2.40E+07$\pm$7.4E+05|8.06|
> |||LS++|1.94E+07$\pm$1.8E+05|328.23|
> |||LSDS++|2.03E+07$\pm$2.1E+05|256.04|
> |||MLS|1.95E+07$\pm$4.5E+04|106.09|
> |||MLSP|-|>6h|
> |||Ours|**1.94E+07$\pm$7.4E+04**|**3.83**|
> |SUSY|100|k-means++|1.74E+07$\pm$4.2E+03|21.54|
> |||LS++|**1.63E+07$\pm$1.0E+04**|695.99|
> |||LSDS++|1.71E+07$\pm$9.3E+04|387.79|
> |||MLS|1.64E+07$\pm$2.3E+04|191.41|
> |||MLSP|-|>6h|
> |||Ours|1.64E+07$\pm$3.0E+04|**9.12**|
>
> *Table 6: Results on HIGGS with 50 LS steps*
>
> |**Dataset**|**k**|**Method**|**Cost**|**Time(s)**|
> |---|---|---|---|---|
> |HIGGS|10|k-means++|1.66E+08$\pm$1.3E+06|**6.51**|
> |||LS++|1.58E+08$\pm$2.3E+06|507.53|
> |||LSDS++|1.62E+08$\pm$2.8E+06|304.46|
> |||MLS|1.56E+08$\pm$1.7E+06|65.31|
> |||MLSP|**1.53E+08$\pm$7.8E+05**|10266.83|
> |||Ours|1.56E+08$\pm$3.4E+05|7.39|
> |HIGGS|50|k-means++|1.31E+08$\pm$1.1E+05|20.58|
> |||LS++|**1.26E+08$\pm$1.2E+05**|13234.24|
> |||LSDS++|1.30E+08$\pm$1.2E+06|1007.52|
> |||MLS|1.26E+08$\pm$2.1E+05|209.14|
> |||MLSP|-|>6h|
> |||Ours|1.26E+08$\pm$1.8E+05|**18.23**|
> |HIGGS|100|k-means++|1.20E+08$\pm$6.9E+05|40.98|
> |||LS++|-|>6h|
> |||LSDS++|1.19E+08$\pm$1.5E+06|2137.09|
> |||MLS|1.15E+08$\pm$6.8E+04|637.82|
> |||MLSP|-|>6h|
> |||Ours|**1.14E+08$\pm$1.5E+05**|**25.82**|
>
> *Table 7: Results on SIFT with 50 LS steps*
>
> |**Dataset**|**k**|**Method**|**Cost**|**Time(s)**|
> |---|---|---|---|---|
> |SIFT|10|k-means++|1.12E+13$\pm$2.1E+10|181.03|
> |||LS++|1.05E+13$\pm$2.8E+10|23560.27|
> |||LSDS++|1.65E+13$\pm$3.0E+10|6265.61|
> |||MLS|1.05E+13$\pm$2.5E+10|4691.80|
> |||MLSP|-|>6h|
> |||Ours|**1.05E+13$\pm$2.1E+09**|**70.24**|
> |SIFT|50|k-means++|9.12E+12$\pm$1.1E+09|381.74|
> |||LS++|-|>6h|
> |||LSDS++|-|>6h|
> |||MLS|-|>6h|
> |||MLSP|-|>6h|
> |||Ours|**8.75E+12$\pm$1.1E+09**|**122.51**|
> |SIFT|100|k-means++|8.48E+12$\pm$5.4E+09|677.93|
> |||LS++|-|>6h|
> |||LSDS++|-|>6h|
> |||MLS|-|>6h|
> |||MLSP|-|>6h|
> |||Ours|**8.13E+12$\pm$8.1E+08**|**65.51**|

---

> > ### Comment · Reviewer_jiTA · 2025-08-05
> >
> > Thank you for the detailed rebuttal. My concern is partially resolved. The supplementary experiments with 50 and 100 iterations provide empirical support for the guess  that the cost at 100 iterations is already very close to that at 400 iterations for most of the test instances. However, I still have a remaining concern: the validation of the exponential tail assumption is based on approximate solutions obtained via multi-swap local search. This introduces a form of circular reasoning—using the output of a heuristic algorithm to justify an assumption that is, in turn, necessary for the theoretical analysis of that same algorithm. Such empirical evidence, while suggestive, does not constitute a rigorous justification that the assumption holds for the optimal clusters, which is what the theoretical guarantees ultimately rely on.

---

> ### Author Response · Authors · 2025-08-06
> **Response to the Reviewer**
>
> **Comment: Thank you for the detailed rebuttal. My concern is partially resolved. The supplementary experiments with 50 and 100 iterations provide empirical support for the guess that the cost at 100 iterations is already very close to that at 400 iterations for most of the test instances. However, I still have a remaining concern: the validation of the exponential tail assumption is based on approximate solutions obtained via multi-swap local search. This introduces a form of circular reasoning—using the output of a heuristic algorithm to justify an assumption that is, in turn, necessary for the theoretical analysis of that same algorithm. Such empirical evidence, while suggestive, does not constitute a rigorous justification that the assumption holds for the optimal clusters, which is what the theoretical guarantees ultimately rely on.**
>
> Response: We thank the reviewer for raising this important concern. Since computing the optimal $k$-means solution is NP-hard in Euclidean space (even in the plane), it is generally infeasible to obtain the exact optimum for real-world datasets. This is why we rely on high-quality approximate solutions to evaluate the exponential tail and cluster size conditions. We agree with the reviewer that this approach may introduce a form of circular reasoning.
>
> To address this concern, we construct synthetic datasets where the optimal clustering centers are fixed using the following methods. This eliminates potential circular reasoning and enables a direct evaluation of our algorithm's performance under both assumption-satisfying and assumption-violating scenarios.
>
> **Assumption-Satisfying Datasets (Tail-1 and Tail-2):** Each dataset is generated by selecting cluster centers uniformly at random in range $[-100, 100]^d$ and adding zero-mean Laplace noise with scaling factor fixed as 1 (to ensure exponential tail). To guarantee that the given center is exactly the optimal 1-means center, we shift all points so that their empirical mean matches the predefined center. Additionally, we enforce a minimum separation of 20 units between clusters to guarantee that the constructed partition corresponds to the optimal ones. Tail-1 contains 1,250,000 points with $k=10$, $d=10$ and minimum cluster size $10k^2$. Tail-2 has 125,000 points with the same $k$, $d$, and minimum cluster size $k^2$.
>
> **Assumption-Violating Datasets (Non-Tail-1 and Non-Tail-2):** These datasets are constructed with overlapping clusters, unbalanced sizes (ranging from 50 to 20,000), and random shifts to misalign the true cluster center with the optimal 1-means center. To violate exponential tails, uniform distributions are used to generate the points in the clusters. Non-Tail-1 has around 1,000,000 points and Non-Tail-2 has around 100,000 points, where both datasets are generated with $k=50$ and $d=10$.
>
> Tables 1–4 present the comparison results. It can be seen that our algorithm consistently achieves comparable or better clustering quality across both types of datasets. On the assumption-satisfying Tail-1 and Tail-2 datasets, it matches or outperforms LS++ (the strongest baseline in clustering quality, while achieving at least 10× speedup. On the more challenging Non-Tail-1 and Non-Tail-2 datasets, which violate the theoretical assumptions, our method maintains similar clustering quality to LS++ with at least 7× speedup. These results demonstrate that our algorithm is not only theoretically grounded but also empirically robust, even when the assumptions are not fully satisfied.
>
> *Table 1: Comparisons on assumption-satisfying dataset Tail-1*
>
> |Tail-1(n=1,250,000,k=50,d=10)|Method|Cost|Time(s)|
> |---|---|---|---|
> |OPT Value: 2.2509E+07|k-means++|4.3956E+07$\pm$1.5E+06|4.48|
> ||MLS|2.4441E+07$\pm$2.9E+06|38.55|
> ||LS++|**2.2511E+07$\pm$3.7E+02**|201.60|
> ||LSDS++|4.1684E+07$\pm$1.9E+06|123.88|
> ||Ours|2.2511E+07$\pm$6.8E+02|**3.42**|
>
> *Table 2: Comparisons on assumption-satisfying dataset Tail-2*
>
> |Tail-2(n=125,000, k=50, d=10)|Method|Cost|Time(s)|
> |---|---|---|---|
> |OPT Value: 2.2494E+06|k-means++|9.6880E+06$\pm$1.9E+05|**0.41**|
> ||MLS|2.5235E+06$\pm$4.5E+04|4.32|
> ||LS++|2.2532E+06$\pm$4.2E+03|26.91|
> ||LSDS++|8.9932E+06$\pm$2.9E+04|13.95|
> ||Ours|**2.2503E+06$\pm$1.1E+03**|2.27|
>
> *Table 3: Comparisons on assumption-violating dataset Non-Tail-1*
>
> |Non-Tail-1(n=1,312,474, k=50, d=10)|Method|Cost|Time(s)|
> |---|---|---|---|
> ||k-means++|3.1769E+08$\pm$1.2E+07|4.78|
> ||MLS|2.0784E+08$\pm$4.5E+04|33.59|
> ||LS++|**2.0497E+08$\pm$7.6E+05**|188.81|
> ||LSDS++|2.6426E+08$\pm$4.3E+06|93.37|
> ||Ours|2.0658E+08$\pm$8.9E+05|**2.88**|
>
> *Table 4: Comparisons on assumption-violating dataset Non-Tail-2*
>
> |Non-Tail-2(n=131,228, k=50, d=10)|Method|Cost|Time(s)|
> |---|---|---|---|
> ||k-means++|3.1673E+07$\pm$5.2E+05|**0.47**|
> ||MLS|2.1226E+07$\pm$1.8E+05|4.69|
> ||LS++|2.1026E+07$\pm$2.7E+05|20.94|
> ||LSDS++|2.6397E+07$\pm$5.2E+05|14.22|
> ||Ours|**2.0854E+07$\pm$1.1E+04**|2.60|

---

### Decision · Program_Chairs · 2025-09-17

**Decision:**

Accept (poster)

**Comment:**

The overall sentiment from reviewers is positive. In particular, they appreciated the additional experiments provided in the authors’ response, which addressed most concerns regarding the reasonableness of the assumptions and the comparison with coreset methods. Therefore, I believe this paper is suitable for acceptance.